# Epigenetic-based differentiation therapy for Acute Myeloid Leukemia

Edurne San José-Enériz [1,2,16], Naroa Gimenez-Camino[1,2,16], Obdulia Rabal[3], Leire Garate[1,2], Estibaliz Miranda[1,2], Nahia Gómez-Echarte [1], Fernando García [4], Stella Charalampopoulou[5], Elena Sáez[3], Amaia Vilas-Zornoza[1], Patxi San Martín-Uriz[1], Luis V. Valcárcel [1,2,6], Naroa Barrena [6], Diego Alignani[1,2], Luis Esteban Tamariz-Amador [1,2,7], Ana Pérez-Ruiz [8], Sebastian Hilscher [9,10], Mike Schutkowski[9,10], Ana Alfonso-Pierola [1,2,7], Nicolás Martinez-Calle[1,2,7], María José Larrayoz [11], Bruno Paiva [1,2], María José Calasanz [11], Javier Muñoz [12,13], Marta Isasa[4], José Ignacio Martin-Subero [2,5,14,15], Antonio Pineda-Lucena[3], Julen Oyarzabal[3] ✉, Xabier Agirre [1,2] ✉ & Felipe Prósper [1,2,7] ✉

Despite the development of novel therapies for acute myeloid leukemia, outcomes remain poor for most patients, and therapeutic improvements are an urgent unmet need. Although treatment regimens promoting differentiation have succeeded in the treatment of acute promyelocytic leukemia, their role in other acute myeloid leukemia subtypes needs to be explored. Here we identify and characterize two lysine deacetylase inhibitors, CM-444 and CM-1758, exhibiting the capacity to promote myeloid differentiation in all acute myeloid leukemia subtypes at low non-cytotoxic doses, unlike other commercial histone deacetylase inhibitors. Analyzing the acetylome after CM-444 and CM-1758 treatment reveals modulation of non-histone proteins involved in the enhancer–promoter chromatin regulatory complex, including bromodomain proteins. This acetylation is essential for enhancing the expression of key transcription factors directly involved in the differentiation therapy induced by CM-444/CM-1758 in acute myeloid leukemia. In summary, these compounds may represent effective differentiation-based therapeutic agents across acute myeloid leukemia subtypes with a potential mechanism for the treatment of acute myeloid leukemia.

Acute myeloid leukemia (AML) is a malignant disease characterized by the uncontrolled proliferation, differentiation arrest, and accumulation of immature myeloid progenitors[1]. Although AML treatment is still largely based on standard chemotherapy and the use of allogeneic hematopoietic stem cell transplantation, improvements in understanding of the pathobiology of the disease have resulted in the development of novel therapies that have altered the landscape of AML treatment. Nevertheless, for the majority of AML patients (~80%)

the prognosis remains highly unsatisfactory and failure remains high, especially in elderly patients[2–4]. Thus, there is an urgent need to develop novel curative therapeutic strategies.

Since the 1970s, various studies have demonstrated that the strategy of inducing malignant cells to overcome their blocked differentiation was an elegant alternative to killing cancer cells. The potential of differentiation therapy to improve cure rates in AML is exemplified by the development of all-trans retinoic acid (ATRA) for

the targeted treatment of acute promyelocytic leukemia (APL), which only represents 10% of all AMLs[5–7]. However, other subtypes of AML display resistance to ATRA-based treatment, and efforts to identify new therapeutic targets to overcome myeloid differentiation blockade have been largely unsuccessful in other subtypes of AML. Therefore, the identification of new therapeutic strategies is an unmet challenge.

Epigenetic alterations contribute to the pathogenesis of hematopoietic malignancies, including AML[8–10]. Epigenetic-modifying enzymes, such as DNA methyltransferases (DNMT), histone methyltransferases (HMT), or histone deacetylases (HDAC), are frequently mutated and/or deregulated in AML, which leads to abnormal chromatin remodeling[11,12]. Epigenetic modifications are reversible, so targeting epigenetic-modifying altered enzymes has become an important area in anti-cancer drug development[11,13]. The approval of two types of epigenetic drugs, the DNMT inhibitors Azacitidine and Decitabine and the HDAC inhibitors (HDACi) Vorinostat and Panobinostat, for poor-prognosis hematological tumors have shown promising clinical benefits for patients who are ineligible or refractory to current therapies[14,15]. In addition, novel epigenetic drugs targeting DNMTs, histone lysine methylation, bromodomains, or demethylases are being developed, including inhibitors of the histone methyltransferase G9a[13,16–18]. Actually, epigenetic inhibitors have proven to be some of the most attractive candidates for inducing differentiation in AML. For instance, HDACi Panobinostat treatment of a mouse model of AML bearing the AML1/ETO fusion triggered terminal myeloid differentiation with a remarkable anti-leukemic response[19], and treatment with small-molecule inhibitors against IDH1/2 also induced myeloid differentiation in AML models with mutant IDH1/2[20,21]. Other examples are the methyltransferase inhibitor LSD1 ORY-1001[22], the bromodomain and extra-terminal (BET) inhibitors OTX015 and JQ1[23], or EPZ004777, an inhibitor of the methyltransferase DOT1L[24].

Here, we report the identification and characterization of CM-444 and CM-1758, two potent deacetylase inhibitors (DACi) with high capacity for inducing myeloid differentiation at low non-cytotoxic doses, both in vitro and in vivo, in different AML subtypes. Both compounds show a mechanism of action different from other DACi, which is mediated by the acetylation of non-histone proteins related to the enhancer-promoter chromatin regulatory complex. These compounds represent a promising approach for a differentiation-based therapy for testing in AML patients.

## Results

### Identification of CM-444 and CM-1758 as pan-HDACi with capacity to induce differentiation of AML cells

Based on previous chemical synthesis efforts, we selected 41 proprietary epigenetic inhibitors belonging to various chemical series: (1) selective and potent substrate-competitive dual inhibitors against methyltransferase activity of G9a and DNMTs[18,25,26]; (2) dual inhibitors targeting HDACs and DNMTs[27]; (3) triple inhibitors against HDACs, G9a, and DNMTs[27]; 4) dual compounds targeting G9a and HDACs; (5) HDAC inhibitors, and (6) dual phosphodiesterase-5 (PDE5) and HDAC inhibitors[28]. We only focused on compounds that showed potent biochemical activity against their own specific targets, with half-maximal inhibitory concentration (IC$_{50}$) values in the nanomolar range (Supplementary Data 1). We also incorporated some reference inhibitors of G9a (A-366), DNMTs (Decitabine), HDACs (Panobinostat), PDE5 (Dildenafil) and ATRA. In pursuit of epigenetic inhibitors capable of inducing myeloid differentiation in AML cells, we performed a dual differentiation-apoptosis assay with these epigenetic small molecules in the HL-60 cell line. First, for each epigenetic small-molecule, we calculated its in vitro half-maximal cell proliferation inhibitory concentration (GI$_{50}$) for the HL-60 cell line (Supplementary Data 1), then decided to use 25% of the GI$_{50}$ concentrations of each compound for a dual differentiation-apoptosis assay to determine their differentiation capacity without directly inducing cell death. In this small-molecule

differentiation screening, we explored induction of the myeloid differentiation marker CD11b with the apoptosis marker annexin-V by flow cytometry after daily treatment of the HL-60 cells with 25% of the GI$_{50}$ concentrations of each compound for 2 days. This led to the identification of several epigenetic small molecules with high capacity to promote myeloid cell differentiation in HL-60 cells (>40% of CD11b+ cells) at low non-cytotoxic doses (<5% of annexin-V + cells). Finally, we selected CM-444 and CM-1758 as our lead compounds because they showed higher differentiation capacity with lower apoptosis induction (Fig. 1A and Supplementary Data 1). Design, synthesis, and SAR exploration of both compounds are described in detail by Rabal O. and colleagues[27] and in our patent WO2018229139A1.

To conduct a complete biochemical characterization of CM-444 and CM-1758, we assessed the inhibitory activity of both compounds over a wide range of 95 enzymes implicated in the regulation of epigenetic mechanisms. Both CM-444 and CM-1758 were found to be selective pan-HDACi, exhibiting low nM IC$_{50}$ values for all HDACs tested (Fig. 1B and Supplementary Data 2). Specifically, we found that CM-444 and CM-1758 had IC$_{50}$ values against HDAC1 (HDAC family-I) of 6.55 and 4.3 nM, against HDAC7 (HDAC family-IIA) of 120 nM, against HDAC10 (HDAC family-IIB) of 15 and 29 nM and against HDAC11 activity of 567.5 and 599.8 nM (HDAC family-IV), respectively. Furthermore, molecular docking studies were also performed showing that both compounds could efficiently bind to different HDACs (HDAC1, HDAC6, or HDAC7) active sites (Fig. 1C). CM-444 also manifested biochemical inhibitory activity against DNMTs, and both compounds were able to biochemically inhibit the histone 3 lysine 27 trimethylation (H3K27me3) demethylase UTX albeit IC$_{50}$ values for UTX were in the micromolar range (Fig. 1B and Supplementary Data 2). Considering that the doses used for differentiation assays in this study were in the low nM range, we considered both compounds to be pan-HDACi, at least at the doses tested. We verified this by measuring histone 3 acetylation (H3Ac) and H3K27me3 levels by western blot and the 5 methylcytosine (5mC) levels by dot blot and performing *LINE-1* pyrosequencing after treatment of HL-60 cells with CM-444 and CM-1758. We only detected a significant increase in H3Ac, with no changes in H3K27me3 or 5mC or *LINE-1* DNA methylation levels (Fig. 1D–F). We validated these results in three different AML cell lines, which showed that our lead compounds only induced an increase in H3Ac (Supplementary Fig. 1A–C). To definitely demonstrate that our compounds do not induce changes in DNA methylation, we treated AML cells at long-term (10 days for HL-60 and ML-2 and 5 days for MOLM-13 and MV4-11) with CM-444 and CM-1758 and then analyzed DNA methylation levels of *LINE-1* (Supplementary Fig. 1D). In summary, these results confirmed that CM-444 and CM-1758 were potent pan-HDACi compounds with a high capacity to promote myeloid differentiation in AML cell lines at low non-cytotoxic doses.

### CM-444 and CM-1758 induce cell differentiation of genetically diverse subtypes of AML

Based on our findings demonstrating the CM-444 and CM-1758 capacity of differentiating HL-60 cells, we evaluated the effects of both compounds in a panel of 15 AML cell lines belonging to different AML subtypes (M1 to M7, according to FAB classification). ATRA was used as a reference to induce differentiation of AML cells. After calculating the GI$_{50}$ at 48 h for all cell lines (Supplementary Data 3), we analyzed CD11b expression by flow cytometry after treatment of cells daily with the 25% GI$_{50}$ of each compound for 48 h. We observed that CM-444 and CM-1758 induced cell differentiation in all subtypes of AML, independently of the genetic alterations, mutations or translocations that were present (Fig. 2A). ATRA treatment induced differentiation in the APL cell line (NB-4) but was not able to induce differentiation in other AML cells lines (Fig. 2A). Furthermore, a more complete analysis including more myeloid markers revealed an increase of CD13, CD14 and HLA-DR after CM-444 and CM-1758 treatment (Supplementary Table 1). In four AML cell lines, HL-60, ML-2, MOLM-13 and MV4-11, we analyzed CD11b and

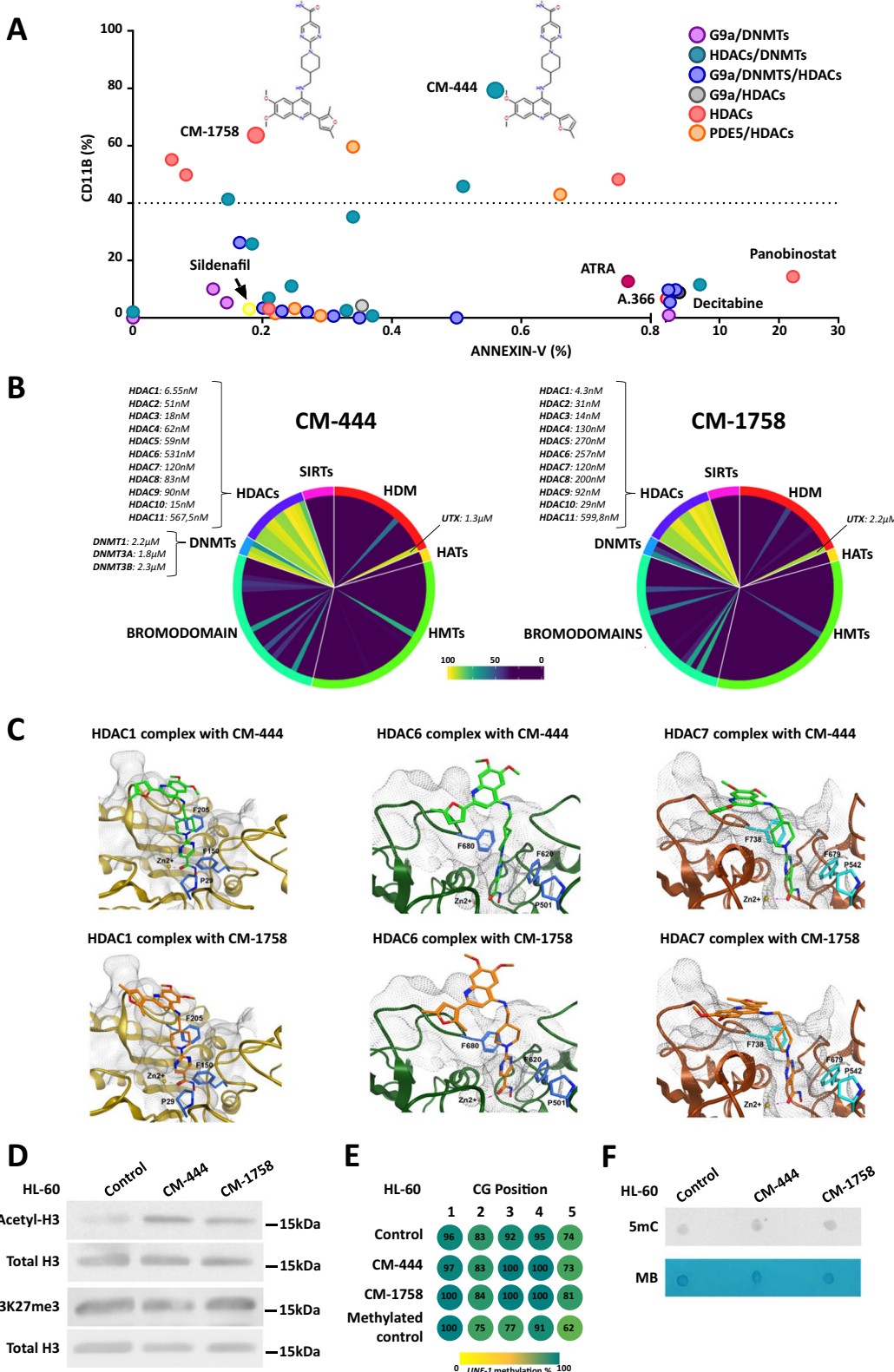

annexin-V for up to 8 days, observing a progressive increase in CD11b expression, achieving differentiation levels close to 90% in all cases after 6–8 days of treatment (Fig. 2B and Supplementary Fig. 2A). Remarkably, CM-444 and CM-1758 also promoted myeloid differentiation of 8 different patient-derived AML myeloid blast cells with distinct genetic translocations and gene mutations (Fig. 2C and Supplementary Table 2).

To analyze the effect of treatment with CM-444 and CM-1758 at the transcriptional level, RNA-seq was performed after treatment of

ML-2 and HL-60 AML cell lines with 25% $GI_{50}$ of our epigenetic inhibitors for 12 h. Principal component analysis (PCA) of RNA-seq data showed the differences between treatments (Supplementary Fig. 2B). After treatment, 136 (3 down-regulated and 133 up-regulated) and 1278 (409 down-regulated and 869 up-regulated) genes were deregulated by CM-444 and CM-1758, respectively in HL-60 cells. In the case of ML-2 cell line, the number of deregulated genes by CM-444 and CM-1758 were 460 (44 down-regulated and 416 up-regulated) and 2520 (1183

**Fig. 1 | Identification and characterization of CM-444 and CM-1758, as a pan-HDACi, with high AML differentiation potencies at low non-cytotoxic doses.**
**A** Epigenetic small-molecule screening was performed in HL-60 cell line treated daily with 25% $GI_{50}$ of each compound for 48 h. CD11b and annexin-V were measured by flow cytometry. The data shown are the mean of three biologically independent experiments. **B** Percentage of inhibition of CM-444 and CM-1758 at 10 μM against a panel of 95 epigenetic targets. HDACs, DNMTs, and UTX $IC_{50}$ values are indicated. **C** Predicted complex of CM-444 and CM-1758 with HDAC1, HDAC6, and HDAC7. **D** H3Ac and H3K27me3 levels were detected by western blot after daily treatment of an HL-60 cell line with 270 nM CM-444 or 300 nM CM-1758 for 48 h. H3 total was used as the loading control (representative experiment of 2 biologically independent studies). **E** DNA methylation of *LINE-1* analyzed by pyrosequencing after daily treatment in HL-60 cell line with 270 nM CM-444 or 300 nM CM-1758 for 48 h. The DNA methylation percentage is indicated inside the circles. As a DNA methylated control, a universally methylated DNA was used. The data shown are the mean of two biologically independent experiments. **F** Dot blot was used to detect global 5-methylcytosine levels after CM-444 and CM-1758 daily treatment for 48 h in an HL-60 cell line (270 nM and 300 nM, respectively). Methylene blue staining was used as a loading control (representative experiment of 2 biologically independent studies). PDE5: phosphodiesterase-5; HDACs: histone deacetylases; DNMTs: DNA methyltransferases; SIRTs: sirtuins; HDMs: histone demethylases; HATs: histone acetyltransferases; HMTs: histone methyltransferases; 5mC: 5-methylcytosine; MB: methylene blue. Uncropped blots and source data are provided as a Source data file.

down-regulated and 1337 up-regulated), respectively (Supplementary Fig. 2C). Gene set enrichment analysis (GSEA) showed a significant downregulation of *MYC* related gene set, essential for initiation of myeloid differentiation (Fig. 2D) and an upregulation of genes sets involved in myeloid differentiation such as *GFI1*, *GATA2* or *CEBPA* targets (Fig. 2D). This finding was corroborated by a deeper analysis of the expression of genes associated with myeloid differentiation as well as expression patterns of granulocyte and monocyte-related genes (Supplementary Fig. 2D). These results were validated in vitro, demonstrating that AML differentiation was associated, in addition to induction of CD11b, with down-regulation of *MYC* (Fig. 2E and Supplementary Fig. 2E), an increase in the cell-cycle inhibitors *CDKN2A* (*p21*) and *CDKN1A* (*p16*) (Fig. 2E and Supplementary Fig. 2E) together with cell-cycle arrest (Fig. 2F and Supplementary Fig. 2F). Moreover, treated cells demonstrated the expected overexpression of key transcription factors that govern myeloid differentiation (*GATA2, TAL1 (SCL), CEBPA, SPI1 (PU.1)*) (Fig. 2G and Supplementary Fig. 2G). Finally, myeloid cell differentiation morphology changes were observed (Fig. 2H). All these results demonstrated that CM-444 and CM-1758 induced a widespread differentiation of genetically diverse AML cells which was associated with the expression of key transcription factors involved in myeloid differentiation.

## CM-444 and CM-1758 showed in vivo myeloid differentiation induction and anti-leukemia activity in AML

Before evaluating the in vivo efficacy of our compounds, we performed absorption, distribution, metabolism, and excretion (ADME) studies and cardiovascular safety assays and examined the therapeutic window achieved by CM-444 and CM-1758 and their pharmacokinetic (PK) parameters showing suitable ADME properties (Supplementary Table 3). Next, we studied the CM-444 and CM-1758 toxicity using the non-tumoral hepatic cell line THLE-2 (lethal concentration required to kill 50% of the cells ($LC_{50}$) values were 0.794 and 0.779 μM, respectively) and peripheral blood mononuclear cells (PBMCs) from healthy donors ($LC_{50s}$ were 1.240 and 2.410 μM, respectively) (Supplementary Table 4). Compared with the in vitro activity observed in the AML cell lines, both compounds showed an optimal therapeutic window, but additional optimization might be required for human use as-well-as GLP studies. Measurements of plasma CM-444 and CM-1758 concentrations after a single intraperitoneal (i.p.) dose of 10 mg/kg suggested that an intermittent dosing schedule could achieve and maintain concentrations above the necessary in vitro levels for AML cells (Supplementary Tables 5 and 6). PK studies in mice with 10 mg/kg i.p., revealed clearance levels for CM-444 and CM-1758 of 0.53 and 0.26 l h$^{-1}$, respectively (Supplementary Tables 7 and 8). For examining the potential in vivo toxicity of both epigenetic inhibitors, we administrated 10 mg/kg i.p. of CM-444 and CM-1758 daily in Rag2$^{-/-}$ γc$^{-/-}$ mice over 3 weeks, followed by a 1-week washout period. The treatment was well tolerated, with no abnormalities in body weight, other physical indicators of sickness, or changes in hematological parameters (Supplementary Fig. 3A). Moreover, histological examination of liver tissues and liver parameters did not show abnormalities in mice treated with CM-444 or CM-1758 in comparison with control mice (Supplementary Fig. 3B, C). Based on these results, we concluded that administration of CM-444 and CM-1758 in mice was safe. Thus, we considered that a 10 mg/kg i.p. dose of CM-444 and CM-1758 was an optimal dose for in vivo studies in mice.

As an in vivo proof of concept, we examined the effect of in vitro pretreatment of AML cells lines with CM-444 and CM-1758 in a subcutaneous mouse model. Two cell lines, HL-60 and ML-2, were treated daily with 270 and 260 nM of CM-444 or 300 and 210 nM of CM-1758, respectively, for 4 days. Prior to injection in mice, the cell differentiation induction by CM-444 and CM-1758 compounds was verified by measuring CD11b by flow cytometry (percentage of CD11b was >90% in all cases). Then, cells pretreated with CM-444, CM-1758 or vehicle (5 × 10$^6$ cells) were implanted subcutaneously into the flanks of female BALB/cA- Rag2$^{-/-}$γc$^{-/-}$ mice. Interestingly, we observed that the tumor growth was significantly slower in animals transplanted with cells pretreated with CM-444 and CM-1758 (Fig. 3A and Supplementary Fig. 4A). In a second set of experiments, ML-2 and MV4-11 AML cells were injected in female BALB/cA- Rag2$^{-/-}$γc$^{-/-}$ mice (5 × 10$^6$ and 10 × 10$^6$ cells, respectively) without previous in vitro treatment and 4 days after transplant animals were treated with 10 mg/kg of CM-444 or CM-1758 i.p. ML-2 mice were treated for 5 consecutive days followed by 2 resting days for 3 weeks, and the MV4-11 mice were treated daily for 2 weeks. Both CM-444 and CM-1758 inhibited tumor growth significantly or very close to significance in the case of CM-444 in the MV4-11 model (Fig. 3B and Supplementary Fig. 4B). ML-2 and MV4-11 tumors were explanted and analyzed. Remarkably, ML-2 as well as MV4-11 cells from tumors treated with CM-444 and CM-1758 exhibited myeloid cell differentiation, as evidenced by the significant increase in *CD11b* expression (Fig. 3C and Supplementary Fig. 4C). Moreover, we observed an increase in H3Ac levels in both ML-2 and MV4-11 cells derived from mice tumors treated with CM-444 and CM-1758 in comparison with untreated tumors (Supplementary Fig. 4D, E), demonstrating that CM-444 and CM-1758 successfully inhibited HDACs in vivo. Finally, we analyzed their therapeutic effect in two different xenogeneic models of AML, generated by intravenously injection of MV4-11 (10 × 10$^6$) or MOLM-13 (1 × 10$^4$) in the tail of female BALB/cA- Rag2$^{-/-}$γc$^{-/-}$ mice. Treatment started 7 or 1 days after injection into MV4-11 or MOLM-13 cells, respectively. The different aggressiveness of these mouse models led us to use different treatment regimens for each of them. Thus, the MV4-11 and MOLM-13 mice were treated with 10 mg/kg of CM-444 and CM-1758 i.p. over 5 consecutive days followed by 2 resting days for 7 and 2 weeks, respectively. CM-444 and CM-1758 increased overall survival (OS) in MV4-11 mice in comparison with the control animals (Fig. 3D). In addition, we observed a statistically significant increase in CD11b in blood samples from mice treated with our compounds in comparison with the control mice (Fig. 3E). Despite not having observed an increase in OS in MOLM-13 mice, CM-444 and CM-1758 showed a significant induction of CD11b in blood samples ($p = 0.002$ and $p = 0.008$, respectively) (Supplementary Fig. 4F). Moreover, no significant weight loss was observed in the treated animals in any of the models generated (Supplementary Fig. 4G–J). Overall, these results

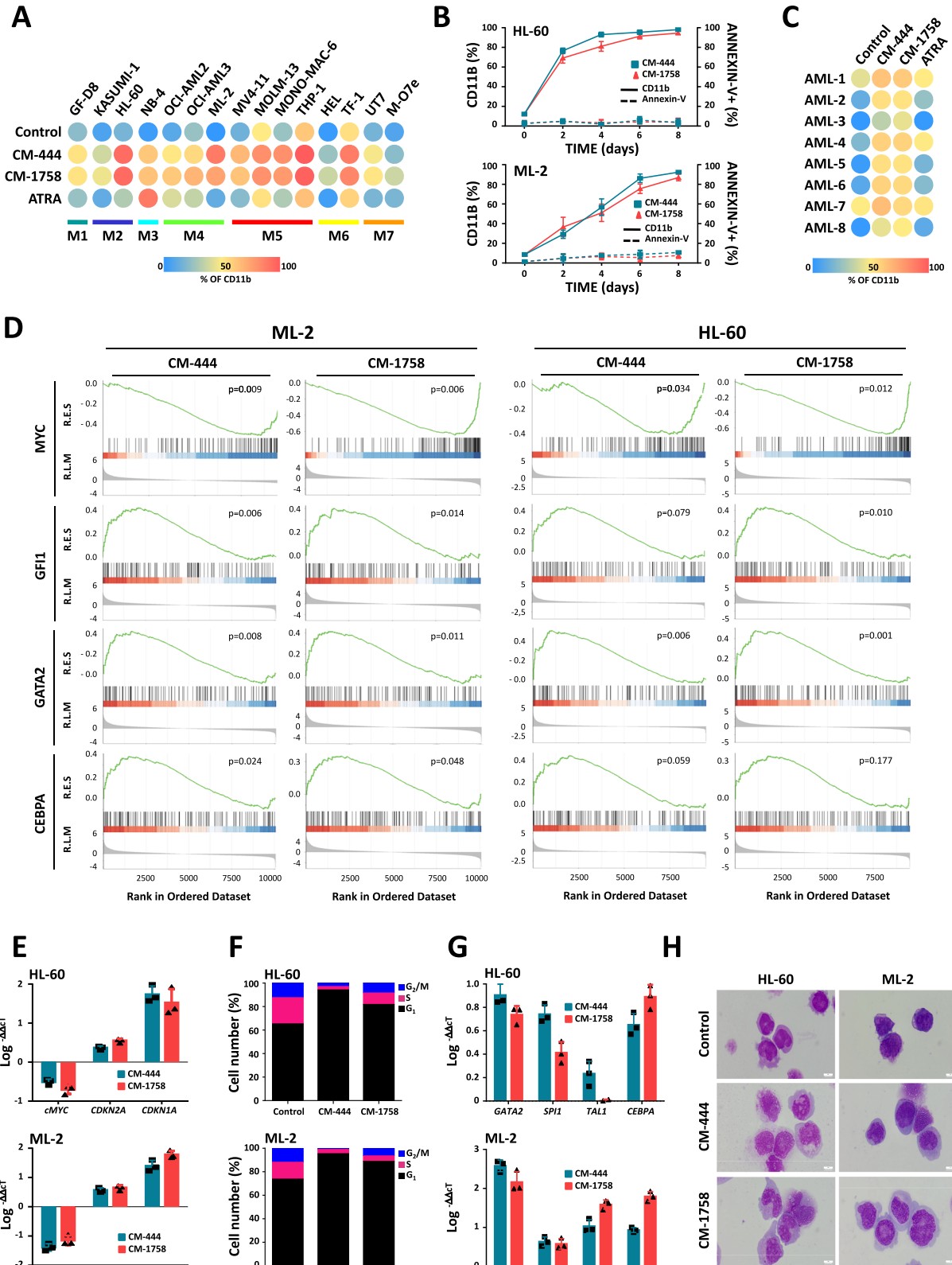

demonstrate that CM-444 and CM-1758 induce in vivo myeloid cell differentiation and potent anti-leukemia activity in AML.

## CM-444 and CM-1758 induce acetylation of non-histone proteins in AML cells

Because of the difference shown between both CM-444 and CM-1758 HDACi as compared to Panobinostat in the initial screening performed (Fig. 1A), we considered that the comparison with the commercial reference HDACi was a necessary step to understand the differences between CM-444 and CM-1758 and other HDACi. Thus, we treated four AML cell lines (HL-60, ML-2, MOLM-13 and MV4-11) with Panobinostat, Entinostat, Vorinostat, Tubastatin and Quisinostat and compared the results with those of CM-444 and CM-1758. We followed the same treatment schedule used with CM-444 and CM-1758: 25% $GI_{50}$ of each compound daily for 2 days. Then, we examined their differentiation capacity by analyzing CD11b by flow cytometry and with Sytox Green

**Fig. 2 | Induction of cell differentiation in all subtypes of AML cells by CM-444 and CM-1758. A** Cell differentiation assay measuring CD11b by flow cytometry in a panel of 15 AML cell lines belonging to different subtypes (M1 to M7, according to FAB classification). Cells were treated daily with 25% $GI_{50}$ of each compound for 48 h. ATRA was used as the differentiation therapy reference. The data shown are the mean of three biologically independent experiments. **B** CD11b and annexin-V were measured by flow cytometry at 2, 4, 6, and 8 days after treating HL-60 and ML-2 cell lines daily with CM-444 or CM-1758. Data are presented as mean values +/- S.D. of three biologically replicates. **C** Cell differentiation assay measuring CD11b by flow cytometry in eight primary AML patient samples. Samples were treated daily with 500 nM and 2 μM of each compound for 48 h. **D** GSEA enrichment analysis showing negative regulation of a MYC related gene set and positive regulation of a GFI1, a GATA2, or a CEBPA-related gene set in ML-2 and HL-60 cells after treatment with CM-444 or CM-1758. HL-60 and ML-2 cell lines were treated with CM-444 or CM-1758 daily for 48 h. Then, **E** q-polymerase chain reaction (PCR) of *MYC*, *CDKN2A* (*p16*) and *CDKN1A* (*p21*); **F** Cell-cycle analysis; **G** q-PCR of *GATA2*, *SPI1* (*PU.1*), *TAL1* (*SCL*) and *CEBP*A and **H** May−Grünwald−Giemsa staining performed in HL-60 and ML-2 cell lines after CM-444 and CM-1758 treatment. Scale bar, 10 μm. Data are presented as mean values +/- S.D. of three biologically replicates. Source data are provided as a Source data file.

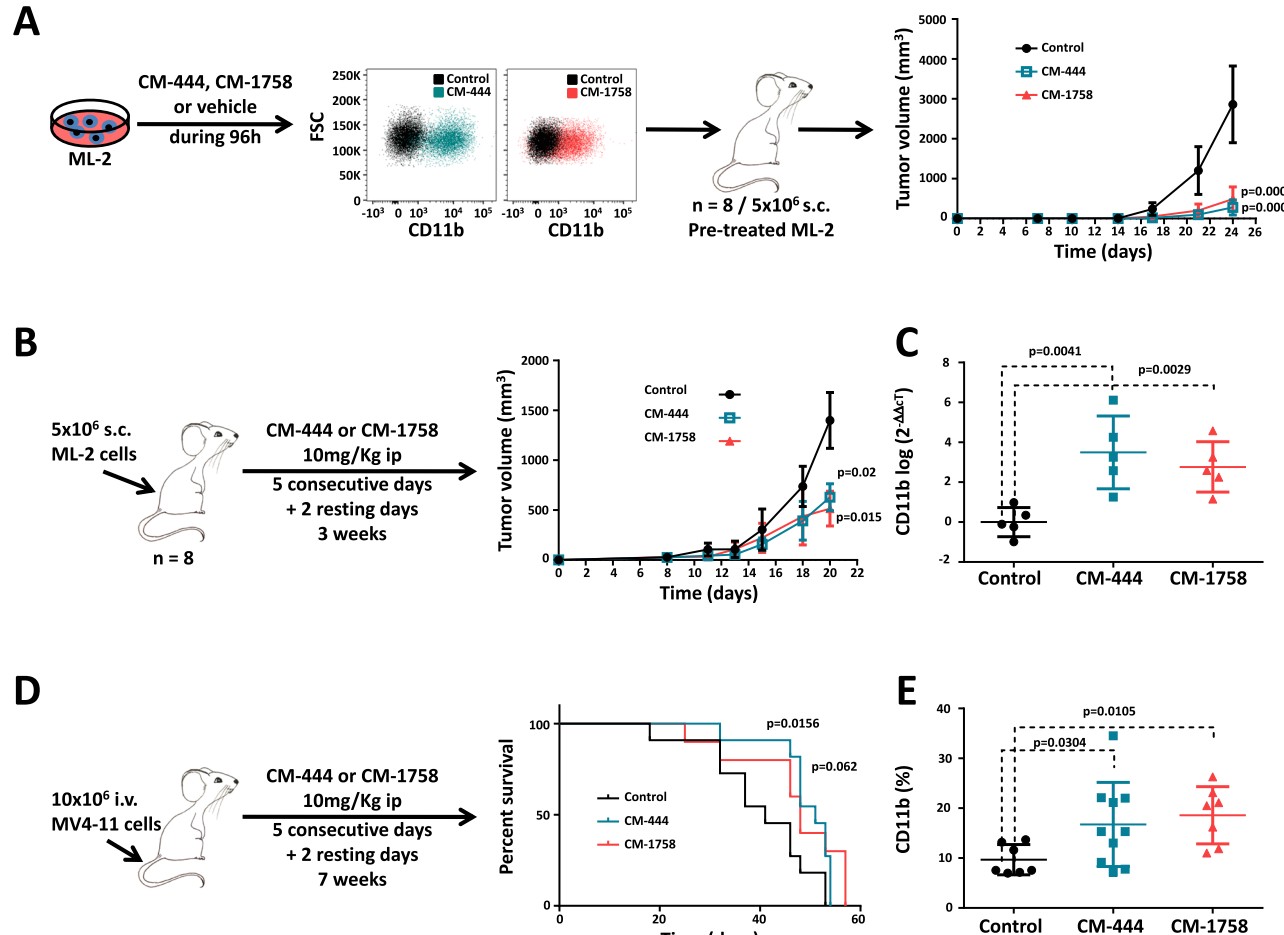

**Fig. 3 | CM-444 and CM-1758 induction of differentiation and anti-leukemic activity in vivo. A** ML-2 cells were pretreated in vitro with 260 nM of CM-444 or 210 nM of CM-1758 for 96 h. After verifying CD11b induction by flow cytometry, equal amounts of cells were injected subcutaneously in $Rag2^{-/-} \gamma c^{-/-}$ mice, and tumor volume was measured (*n* = 8). Error bars indicate the S.D. Statistical significance was calculated by a two-tailed Student's *t*-test. **B** Schematic diagram of the in vivo CM-444 and CM-1758 treatment procedure and tumor volume curve of the ML-2 subcutaneous xenograft model in $Rag2^{-/-} \gamma c^{-/-}$ mice (*n* = 8). Error bars indicate the S.D. Statistical significance was calculated by a two-tailed Student's *t*-test. **C** *CD11b* from tumors in an ML-2 subcutaneous model was measured by q-PCR (*n* = 5). Error bars indicate the S.D. Statistical significance was calculated by a two-tailed Student's *t*-test. **D** Schematic diagram of the in vivo CM-444 and CM-1758 treatment procedure and Kaplan−Meier survival curve for evaluating the survival time of $Rag2^{-/-} \gamma c^{-/-}$ mice engrafted with MV4-11 cells after intravenous administration (*n* = 10). *P*-values assessed by log-rank. **E** CD11b levels in blood samples from an intravenous MV4-11 mouse model measured by flow cytometry (*n* = 7 for Control and CM-1758 group and *n* = 10 for CM-444 group). Error bars indicate the S.D. Statistical significance was calculated by a two-tailed Student's *t*-test. Control: treatment with CM-444 or CM-1758 vehicle (80% saline, 10% Tween 20, and 10% DMSO); s.c.: subcutaneous; i.v.: intravenous. Source data are provided as a Source data file.

as a cell death marker. Surprisingly, we observed that the commercial HDACi hardly induced differentiation, at least at the doses and times tested. We only observed a few exceptions, mainly with entinostat, for which CD11b induction was accompanied by significant toxicity (Fig. 4A and Supplementary Fig. 5A).

As HDACi regulate the acetylation of histones but also non-histone proteins, to elucidate the molecular mechanism underlying the differentiation capacities of CM-444 and CM-1758 in comparison with commercial HDACi, we analyzed the total proteome and acet-ylome in an ML-2 cell line. First, we treated the ML-2 cells with 25% $GI_{50}$ of CM-444 and CM-1758 or with the reference HDACi of Panobinostat and Vorinostat for 12 h. Firstly, after cells were lysed and digested and the subsequent peptides were labeled and fractionated, 5 % of the fractions was kept for proteome analysis by liquid chromatography-tandem mass spectrometry (LC-MS/MS) (Fig. 4B). We were able to quantify a total of 8342 proteins after treatment with any of the HDACi.

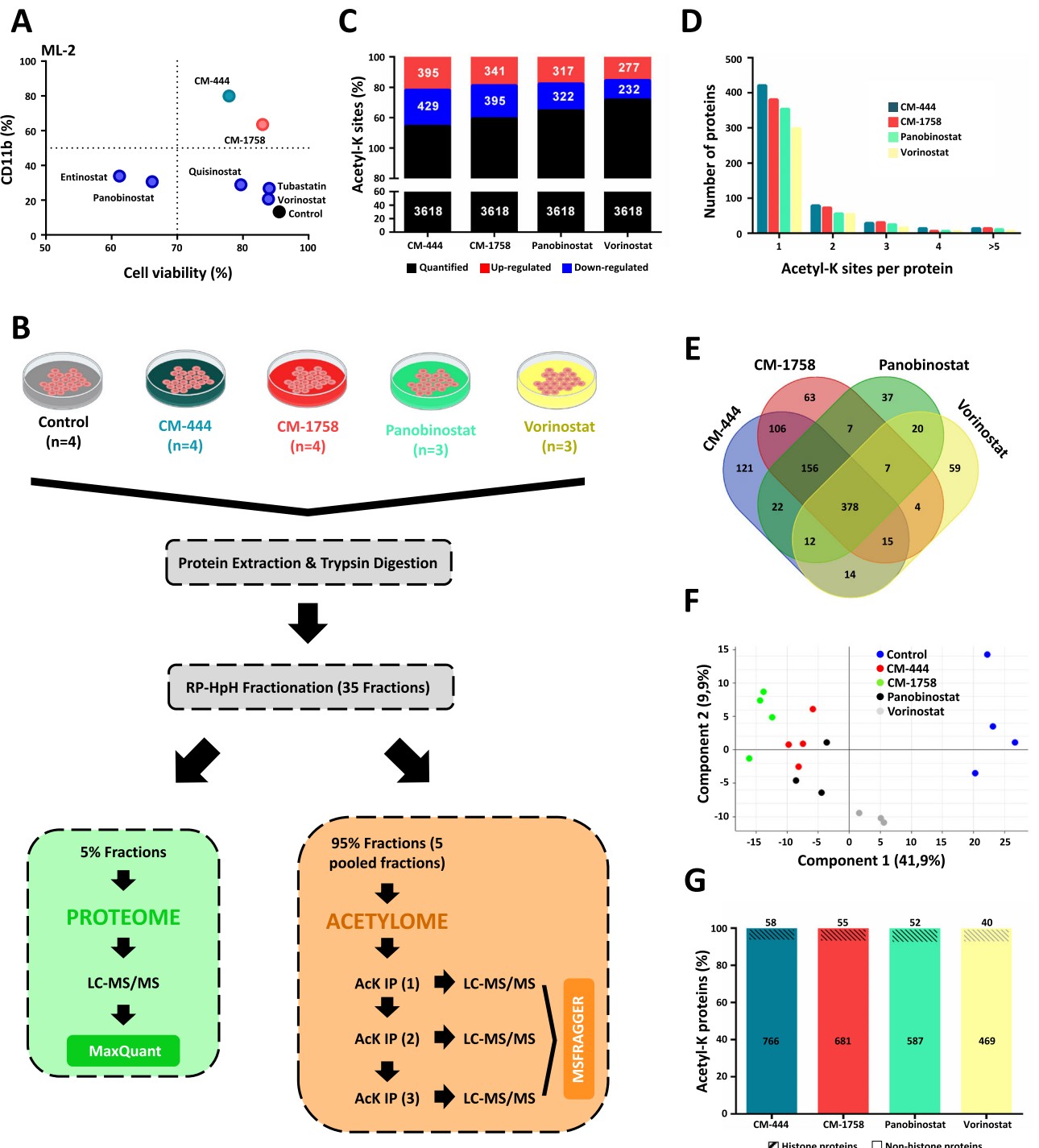

**Fig. 4 | Acetylome analysis revealed different acetylation profiles between CM-444/CM-1758 and the reference HDACi. A** Cell differentiation assay measuring CD11b and Sytox Green by flow cytometry in an ML-2 cell line treated with 25% GI$_{50}$ of CM-444 (260 nM), CM-1758 (210 nM) and the commercial HDACi Panobinostat (12.9 nM), Vorinostat (1.1 μM), Entinostat (2.3 μM), Quisinostat (18 nM) and Tubastatin (2.5 μM) for 48 h. The data shown are the mean of three biologically independent experiments. **B** Proteome and acetylome experimental design: ML-2 cells were treated with 25% GI$_{50}$ of CM-444 (260 nM) or CM-1758 (210 nM) and with reference HDACi Panobinostat (12.9 nM) or Vorinostat (1.1 μM) for 12 h. Then, cells were lysed and digested and the subsequent peptides were labeled and fractionated. 5 % of the fractions were kept for proteome analysis by LC-MS/MS. For the complete acetylome study, 95% of the obtained fractions were combined into five pools, and three consecutive IPs were performed for each pool, and the resulting samples were subsequently analyzed by LC-MS/MS. Figure 4/panel B, created with

BioRender.com released under a Creative Commons Attribution-NonCommercial-NoDerivs 4.0 International license. **C** The total number of Acetyl-K sites quantified and the fraction of Acetyl-K sites regulated by each HDACi are shown. The bar chart shows the percentage of upregulated sites (log2 FC > 1, $p$ < 0.05, shown in red) and downregulated sites (log2 FC < 1, $p$ > 0.05, shown in blue). **D** Distribution of acetylated sites in the acetylated proteins. **E** Venn diagram of the total number of regulated Acetyl-K sites in ML-2 cells after treatment with CM-444, CM-1758, Panobinostat or Vorinostat compared with untreated cells. **F** PCA of acetylome data from ML-2 cells after treatment with CM-444, CM-1758, Panobinostat or Vorinostat compared with untreated cells. **G** Representation of Acetyl-K sites regulated after HDACi in histone and non-histone proteins. The data are shown as percentages, and the number of Acetyl-K sites regulated by each HDACi is indicated in the corresponding bar. Source data are provided as a Source data file.

Among them, 9.5%, 8.8%, 5.8% and 2.7% were significantly regulated after treatment with CM-444, CM-1758, Panobinostat or Vorinostat, respectively (Supplementary Fig. 5B). Only the levels of 192 proteins were regulated by treatment with the four HDACi used (Supplementary Fig. 5C). Additionally, it is important to highlight that CM-444 and CM-1758 shared approximately 80% of the regulated proteins. Vorinostat and Panobinostat showed a shared regulation in over 90% of the proteins, while Panobinostat shared around 40% of the regulated proteins with both CM-444 and CM-1758. This global proteome analysis was performed not only to obtain an overview of the protein level changes caused by an HDACi, but also as a control for a complete acetylome study where those proteins regulated after HDACi treatment were eliminated from further acetylome analysis, allowing us to detect those changes associated to acetylation of one protein but not due to changes in the protein levels.

For the complete acetylome study, 95% of the obtained fractions were combined into five pools for acetyl-lysine (Acetyl-K) enrichment using an anti-acetyl-lysine antibody. Particularly, three consecutive immune-precipitations (IPs) were performed for each pool, and the resulting samples were subsequently analyzed by LC-MS/MS (Fig. 4B). Specifically, we quantified 3618 Acetyl-K sites for each of the experiments carried out with each of the four HDACi used (Fig. 4C). Most proteins were found acetylated on a single K residue, whereas around 7-10% of the identified proteins were highly acetylated (modified at ≥3 Acetyl-K sites) (Fig. 4D). The percentage of Acetyl-K sites regulated after CM-444, CM-1758, Panobinostat or Vorinostat treatment were 22.8%, 20.3%, 17.7% and 14.1%, respectively. The fraction of upregulated and downregulated Acetyl-K sites in the treated cells was similar, with 47.9%, 46.3%, 49.6% and 54.4% of the acetylation sites upregulated by CM-444, CM-1758, Panobinostat or Vorinostat, respectively (Fig. 4C). Despite all four compounds shared 378 of the 1021 regulated Acetyl-K sites, each HDACi modulated a specific acetylation pattern. Interestingly, the pattern of modulated Acetyl-K sites was very similar between the treatment with CM-444 and CM-1758, with approximately 80% of the Acetyl-K sites commonly regulated (Fig. 4E). PCA of acetylome data verified these observations, showing a clear separation of untreated cells from HDACi-treated cells. In addition, cells treated with CM-444 and CM-1758 clustered together and separately from Panobinostat-and above all, from Vorinostat-treated cells (Fig. 4F).

We next explored the acetylated sequences identified after HDACi treatment to determine if there was any difference in the acetylation motifs among CM-444/CM-1758 and the commercial HDACi. Thus, we analyzed the 7 residues flanking the acetylated lysines in both up- and downregulated Acetyl-K sites. This analysis revealed that the acetylation motifs were very similar between CM-444/CM-1758 and Panobinostat or Vorinostat, discarding it as the cause of the different differentiation capacity of HDACi (Supplementary Fig. 6). Interestingly, more than 90% of the Acetyl-K residues were detected in histone proteins. However, most of these sites were not modified after treatment with any of the HDACi. On the other hand, the majority of the Acetyl-K sites regulated after treatment with each of the four HDACi were found in non-histone proteins (Fig. 4G). These results suggest that CM-444 or CM-1758 could modulate specifically non-histone protein acetylation patterns in AML cells, in addition to histone protein acetylation. This differential effect could be key to discover the molecular mechanism underlying the increased ability to induce myeloid differentiation shown by CM-444 and CM-1758.

## CM-444 and CM-1758 modulate lysine acetylation on bromodomain proteins

Next, we focused on the analysis of the Acetyl-K sites differentially regulated by CM-444 and CM-1758 but not by Panobinostat and/or Vorinostat since these could explain the differences between our compounds and reference HDACi. One hundred four non-histone Acetyl-k proteins were specific to our epigenetic compounds CM-444

and CM-1758, as shown in Fig. 5A. We next generated a protein–protein interaction network using STRING with the application of K-means clustering on the main networks. K-means clustering split the network of the 104 specific Acetyl-K regulated proteins into three clusters: Cluster 1 was the smallest subnetwork of hubs (7 proteins, blue color); Cluster 2 was a subnetwork with more hubs (27 proteins, green color), and cluster 3 was the largest subnetwork (56 proteins, red color) (Fig. 5B). Cluster 1 showed no significant biological processes or molecular functions, as the number of edges was very limited. Cluster 2 was enriched with nucleic acid metabolism processes and gene expression (Supplementary Fig. 7A) and was mainly associated with nucleic acid binding (Supplementary Fig. 7B). Finally, several enriched biological processes in cluster 3 were related to histone modifications, DNA repair and notably, myeloid differentiation (Supplementary Fig. 7C). In fact, several transcription factor related to myeloid differentiation were specifically acetylated by our compounds such as CEBPA, EP300 and PML. The molecular function was associated mainly with histone, chromatin, transcription factors and histone acetyltransferase activity (Supplementary Fig. 7D). Of note, a large number of the cluster-3 proteins were related to DNA repair, bromodomains or were members of the MOZ/MORF, mediator, SWI/SNF chromatin remodeling or histone acetyltransferase complexes. A remarkable finding was that most of the cluster-3 proteins whose acetylation was specifically modulated by CM-444 and CM-1758 treatment were proteins that participate in the enhancer–promoter chromatin regulatory complex, and some have been shown to have important roles in AML, such as proteins containing bromodomains[29] and those in the cohesin complex[30], mediator complex[31] or MOZ/MORF complex[32], among others (Fig. 5C). Currently, the role of these specific acetylations in each of the non-histone proteins is unknown. However, since most of these non-histone proteins are involved in this complex enhancer-promoter chromatin structure, this observation suggests that the global modification at the acetylation level after treatment with CM-444 or CM-1758 could directly regulate the expression of transcription factors involved in myeloid differentiation.

To delve into the role that bromodomains proteins (BRDs) could play in the differentiation process induced by our compounds but not by other DACi, we performed CUT&RUN experiments with BRD4 and the specific histone marks H3K9Ac and H3K27Ac in HL-60 after treatment with CM-444 or Panobinostat for 12 h. The numbers of peaks of the different marks are shown in Fig. 6A. Interestingly, CM-444 induced changes in the location of peaks associated with BRD4, H3K27Ac and H3K9Ac. While BRD4, H3K27Ac and H3K9Ac peaks were predominantly located in promoters in untreated cells, after treatment with CM-444 the majority of the peaks were located in intronic and distal intergenic regions (Fig. 6B). The analysis of the distribution of the chromatin states was consistent with all these findings (Supplementary Fig. 8). Moreover, CM-444 treatment drove BRD4, H3K9Ac and H3K27Ac to genes related to myeloid differentiation and we observed that BRD4 peaks after CM-444 treatment were also associated with active chromatin regions as demonstrated by the increase in H3K27Ac mark (Fig. 6C). Regarding the global changes carried out by Panobinostat, it should be highlighted that they were much lower than with CM-444 (Fig. 6B). Regarding H3K27Ac, there were hardly any differences in the localization of the peaks after Panobinostat treatment compared to the control (Fig. 6B). This is reflected in the different peaks observed in genes involved in AML cells differentiation (Fig. 6C), where only minor changes were observed in comparison with untreated cells. All these results demonstrated that our compounds modulate the localization of BRDs, as well as acetyl histone marks, in a different way than Panobinostat. This could explain the different potential to induce differentiation of AML of our compounds in comparison with other DACi. To further demonstrate the role of acetylation modulation of non-histone proteins, we focused on BRD4 and BRD1, 2 members of the BRD modulated by CM-444 or CM-1758

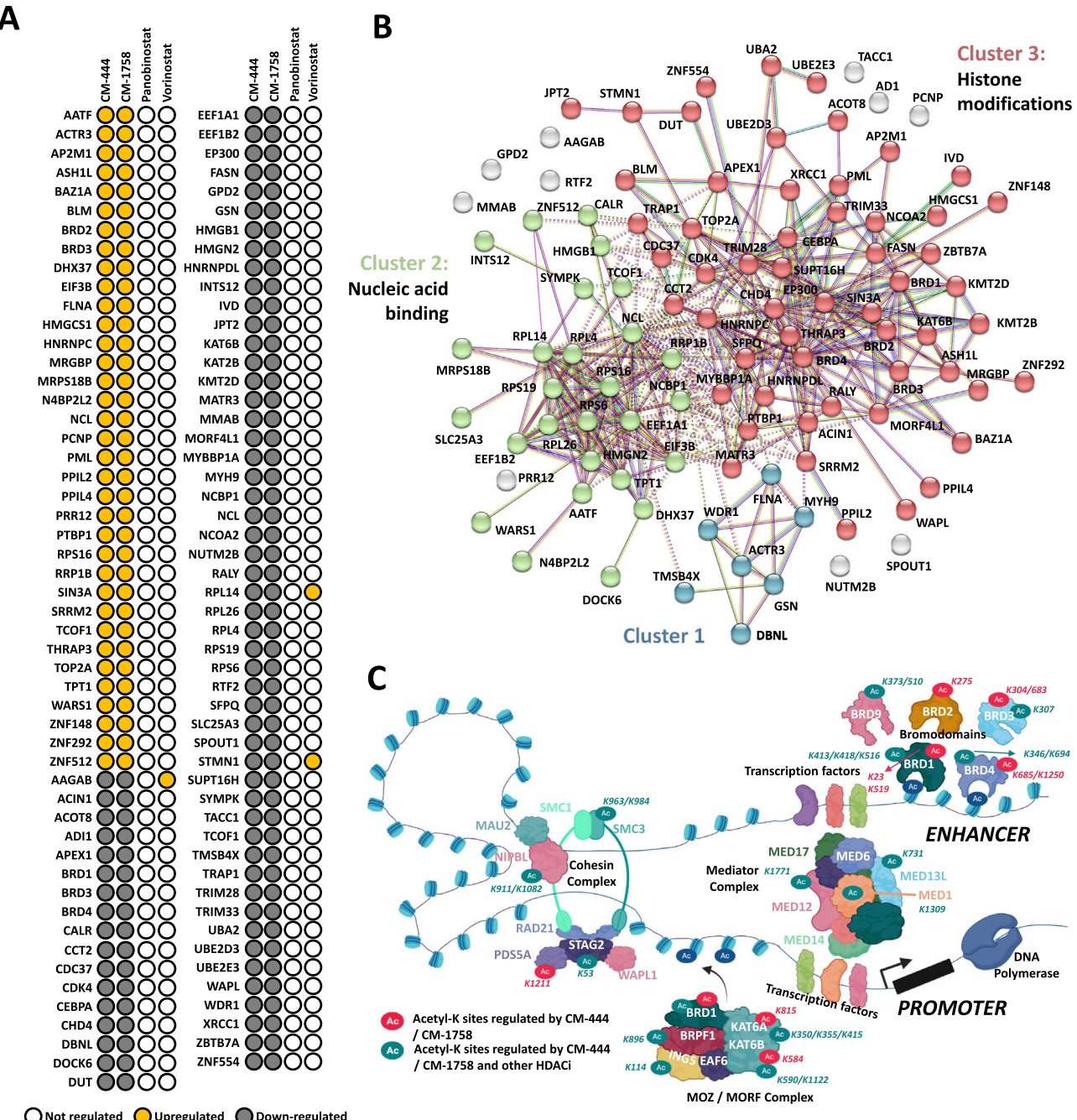

**Fig. 5 | Differential acetylation of non-histone proteins, a key factor in the molecular mechanism underlying the differentiation induction in AML with CM-444 and CM-1758. A** Acetyl-K sites specifically deregulated by CM-444 and CM-1758. **B** STRING protein–protein interaction analysis on the 104 Acetyl-K sites differentially regulated specifically by CM-444 and CM-1758. Three different functional clusters were detected. **C** Representation of the protein complexes highly acetylated by CM-444 and CM-1758 and other HDACi. The specific K-sites regulated in each protein are specified. Figure 5/panel **C**, created with BioRender.com released under a Creative Commons Attribution-NonCommercial-NoDerivs 4.0 International license.

(Fig. 5A–C). BRDs selectively recognize and bind to acetylated lysine residues, mainly histones; consequently, BRDs are bound to active enhancers having important roles in the regulation of gene expression[33]. To determine if the specific acetylation of these BRDs is important for the regulation of key transcription factors in myeloid differentiation, HL-60, ML-2, MV4-11 and MOLM-13 AML cell lines were treated with two different BRD inhibitors, Molibresib and JQ1, in combination with CM-444 or CM-1758. BRD inhibitors did not induce myeloid differentiation at the doses used in the experiment (25% GI$_{50}$). However, when combined with CM-444 or CM-1758 treatment, led to a block in the myeloid differentiation induced by our epigenetic compounds, with levels of CD11b similar to those of untreated cells (Fig. 6D–E and Supplementary Fig. 9). This effect of the BRD inhibitors on differentiation was also observed when we analyzed the expression of *MYC* and key transcription factors for myeloid differentiation (*GATA2, TAL1, CEBPA, SPI1*). Considering that BRD inhibitors are known to downregulate *MYC* transcription[34], it is not surprising that *MYC* downregulation was still observed despite the addition of Molibresib or JQ1. However, the expressions of all transcription factors were similar or downregulated relative to those of untreated AML cells (Fig. 6F–G and Supplementary Fig. 9). These results indicate that acetylation of BRD proteins, especially BRD4 and BRD1, and other non-

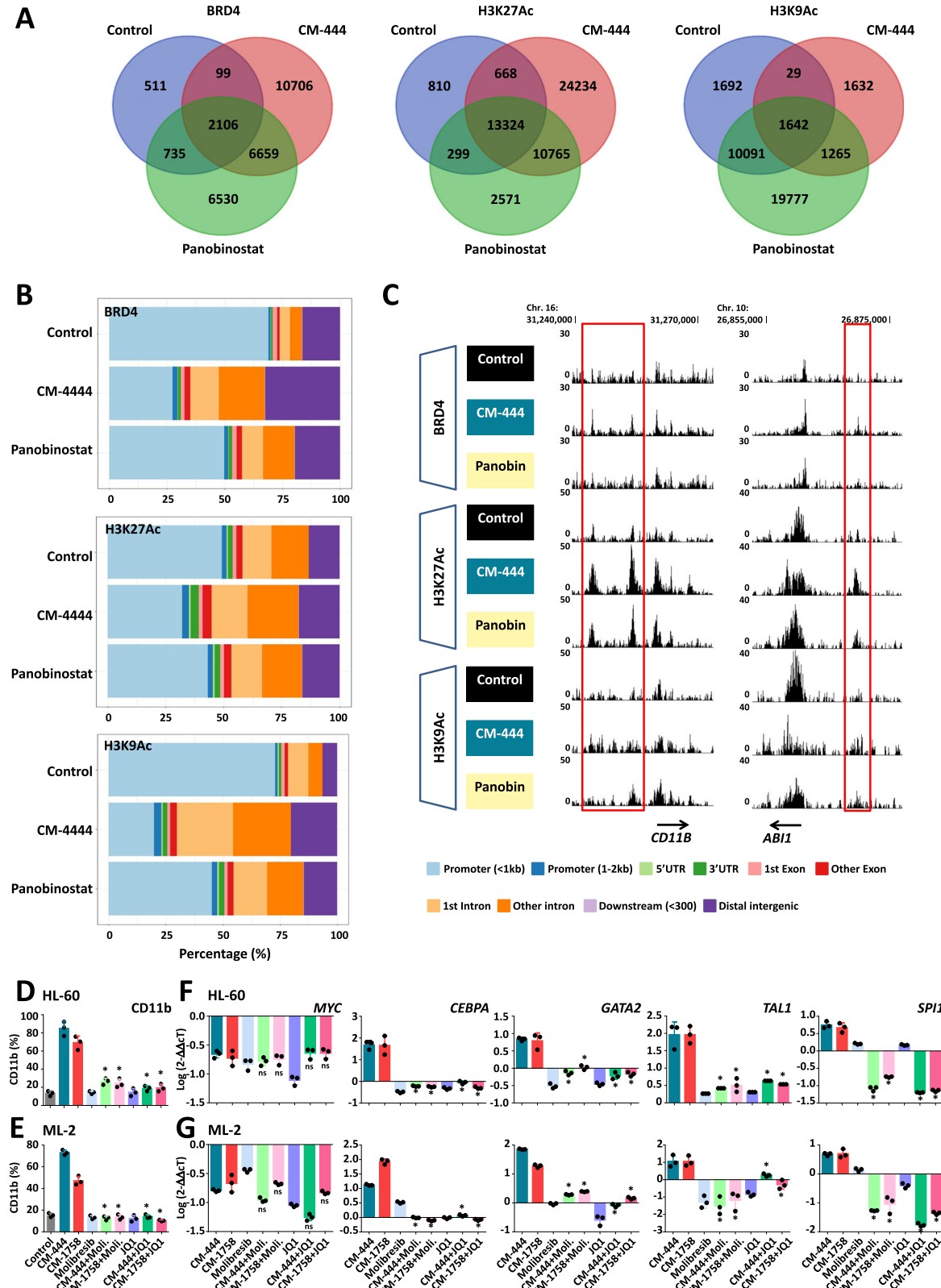

**Fig. 6 | Role of BRDs in the differentiation induction in AML with CM-444 and CM-1758. A** BRD4 (left), H3K27Ac (middle) and H3K9Ac (right) peak numbers in HL-60 cells treated with CM-444 or Panobinostat for 12 h. **B)** CUT&RUN peaks distribution of BRD4 (upper panel), H3K27Ac (middle panel) and H3K9Ac (lower panel) in HL-60 cells treated with CM-444 or Panobinostat for 12 h. **C** Examples of myeloid genes involved in differentiation of AML cells. Cell differentiation assay measuring CD11b by flow cytometry after treating **D** HL-60 and **E** ML-2 cells daily with 25% $GI_{50}$ of CM-444, CM-1758, Molibresib, JQ1, and the combination of CM-444

or CM-1758 with Molibresib or JQ1 for 48 h. Data are presented as mean values +/- S.D. of three biological replicates. Statistical significance was calculated by a two-tailed Student's t-test. n.s. = non-significant; *$p \le 0.05$. q-PCR of *GATA2, PU.1, SCL* and *CEBPA* after treating **F** HL-60 and **G** ML-2 cells daily with 25% $GI_{50}$ of CM-444, CM-1758, Molibresib, JQ1, and the combination of CM-444 or CM-1758 with Molibresib or JQ1 for 48 h. Data are presented as mean values +/- S.D. of three biological replicates. Statistical significance was calculated by a two-tailed Student's t-test. n.s. = non-significant; *$p \le 0.05$. Source data are provided as a Source data file.

histone proteins involved in this enhancer–promoter chromatin regulatory complex is essential to enhance the expression of key transcription factors that induce myeloid differentiation. In summary, these results show that BRDs have an essential role in the differentiation therapy exerted by CM-444 or CM-1758 in AML cells.

## Discussion

Since Conrad Waddington proposed the term "epigenetics" in 1942[35], numerous advancements have been made in our understanding of their role in cancer development. This makes epigenetic-targeted therapy a promising strategy for anti-cancer treatment, and the HDACi Panobinostat, Vorinostat and Romidepsin are some of the first epigenetic drugs approved for poor-prognosis hematological tumors[14,36]. The success of HDACi has been modest in AML[37], and current HDACi have been associated with clinical toxicity[38–40], some of which are serious or life-threatening, such as severe cardiac toxicity, myelosuppression and gastrointestinal and hepatic effects[38]. The development of next-generation epigenetic inhibitors that overcome these limitations is one of the challenges of epigenetic therapy.

A hallmark of AML is the inability of self-renewing malignant cells to mature into a non-dividing terminally differentiated state[1]. A very attractive way to treat AML is with differentiation therapy, exemplified by the development of ATRA for the treatment of APL[5–7]. Unfortunately, other strategies of differentiation therapy, including different HDACi, have been largely unsuccessful in other subtypes of AML. In this study, we set out to shed light on these two challenges by joining epigenetic and differentiation therapy as a single approach, by developing an epigenetics-based differentiation therapy. We describe the discovery of two potent DACi, CM-444 and CM-1758, with strong capacity to induce myeloid differentiation at low non-cytotoxic doses both in vitro and in vivo, and prolonging survival in AML models. It is important to note that our DACi showed good efficacy in AML cell lines belonging to all AML subtypes and were independent of the genetic alterations, mutations or translocations that were present. This is especially important in the case of AML differentiation therapy and might help improve the therapeutic response of AML patients. Regarding toxicity, our small molecules have been shown to be safe in mice, presenting no evidence of hematological, hepatic or cardiac toxicity. These preclinical results indicate the potential of our DACi in differentiation therapy. However, previous DACi have shown limited effects as single agents in AML[37]. Therefore, it would be useful to explore the combination of CM-444 and CM-1758 with conventional chemotherapy or other epigenetic inhibitors.

Traditionally, the abbreviation "DACs" has been used almost exclusively in HDACs that refers to the deacetylation of histones. Presently, it is well known that DACs are much more than histone deacetylases and that both histone and non-histone (de)acetylations are important regulators of different biological cell processes[41]. Acetylation is a common post-translational modification of non-histone proteins, including oncogenes, tumor-suppressor genes, transcription factors, chaperones, and cell signaling molecules, such as MYC[42], P53[43,44], or PTEN[45]. However, the significance of each of these specific acetylations in non-histone proteins has yet to be deciphered, with some of these related to changes in protein stability, protein–protein interactions, or protein–DNA interactions[43,46]. Our acetylome study, conducted with CM-444- and CM-1758-treated AML cells, has clearly shown the abundance of acetylation of non-histone proteins. Moreover, our results support the hypothesis that non-histone modifications are relevant in cancer biology and further justify the use of DACi's for cancer therapy.

Interestingly, it has been described that the profile of (de)acetylation of both histone and non-histone proteins affected by DACi treatment is DACi specific[47]. Schölz C. et al. obtained the acetylation signature of 19 different DACi, revealing DACi specificity in cells at the individual acetylation-site level. This is an important discovery and provides a framework for understanding the different modes of action and differences in the response of different DACi. In this study, we showed that the differentiation potencies of CM-444 and CM-1758 was much higher than that of other known DACi, suggesting a different molecular mechanism. We decided to study the complete acetylome of AML cells treated with our compounds CM-444 and CM-1758 compared with Panobinostat and Vorinostat to elucidate the mechanism underlying the differentiation induction of our DACi. As in the study of Schölz C. et al., we observed large differences in the acetylation signatures among the four DACi tested but great similarities between our two DACi, CM-444 and CM-1758. Interestingly, the acetylome study revealed that modulation by CM-444 and CM-1758 of non-histone protein acetylation, which occurs in the chromatin enhancer–promoter complex, such as by BRDs, have a more important role than classical histone acetylation in the regulation of key TFs for myeloid differentiation in AML. BRDs selectively recognize and bind to histone-acetylated lysine residues in active enhancers that have important roles in the regulation of gene expression[33]. We showed that BRDs have a very important role in the differentiation therapy induced by CM-444 or CM-1758 in AML cells. The identification of this mechanism together with the capacity of inducing myeloid differentiation in all AML subtype cells adds further importance to the discovery of these DACi, which represent a promising approach to differentiation-based therapy for testing in AML patients.

## Methods

Our research complies with all relevant ethical regulations and has been approved by the committee of ethics of the research of the University of Navarra and follows their prescribed ethical guidelines.

### Cell culture

KASUMI-1 (ACC 220, DSMZ), HL-60 (ACC 3, DSMZ), NB-4 (ACC 207, DSMZ), OCI-AML3 (ACC 582, DSMZ), MV4-11 (ACC 102, DSMZ), MOLM-13 (ACC 554, DSMZ), HEL (ACC 11, DSMZ), GF-D8 (ACC 615, DSMZ), TF-1 (CRL-2003, ATCC), THP-1 (TIB-202, ATCC) and M-O7e (ACC 104, DSMZ) cell lines were cultured in RPMI-1640 medium supplemented with 20% fetal bovine serum (FBS). GM-CSF was added to GF-D8 (50 ng/μL), TF-1 (50 ng/μL) and THP-1 (10 ng/μL) culture medium. M-O7e cells were also supplemented with 10 ng/μL of IL-3. OCI-AML2 (ACC 99, DSMZ) cells were cultured with DMEM medium + 20% FBS. The MONO-MAC-6 (ACC 124, DSMZ) cell line was maintained in RPMI-1640 medium supplemented with 20% FBS, 2 mM L-glutamine, non-essential amino acids, 1 mM sodium pyruvate, and 10 μg/ml human insulin. The UT-7 (ACC 137, DSMZ) cell line was cultured with Alpha-MEM medium supplemented with ribo- and deoxyribonucleosides, 20% FBS, and 5 ng/ml GM-CSF. All cell lines were maintained at 37 °C in a humid atmosphere containing 5% CO$_2$. The cell lines were obtained from the DSMZ or the American Type Culture Collection (ATCC). All cell lines were authenticated by performing a short tandem repeat allele profile and were tested for mycoplasma (MycoAlert Sample Kit, Cambrex).

### Compounds

Panobinostat (HY-10224-CS-0267), Quisinostat (HY-15433) and Molibresib (HY-13032) were purchased from MedChemExpress (Monmouth Junction, NJ, USA). Entinostat (S1053) and JQ1 (S7110) were obtained from Selleck Chemicals (Houston, TX, USA). Vorinostat (10009929), Tubastatin (6270) and Retinoic acid (R2625) were purchased from Cayman Chemical (MI, USA), Tocris (Germany) and Sigma-Aldrich (St Louis, Missouri, USA), respectively.

### Synthesis of compounds CM-444 and CM-1758
**General chemistry information.** Unless otherwise noted, all starting materials, reagents, and solvents were purchased from commercial suppliers and used without further purification. Air-sensitive reactions

were performed under $N_2$. The NMR spectroscopic data were recorded on a Bruker AV400 or VARIAN 400MR spectrometer with standard pulse sequences. Chemical shifts ($\delta$) are reported in parts per million (ppm). The abbreviations used to explain multiplicities are s = singlet, d = doublet, t = triplet, q = quadruplet, m = multiplet and br s = broad singlet. The coupling constant, $J$, is reported in Hertz units. Melting points were determined on a Mettler FP82 hot stage controlled by a Mettler FP80 central processor.

**Protocol for analytical high performance liquid chromatography (HPLC).** The purity of final compounds was measured by HPLC. HPLC-analysis was performed using a Shimadzu LC-20AB with a Luna-C18(2), 5 μm, 2.0 × 50 mm column at 40 °C with a diode-array detector. Solvent A: water with 0.037% trifluoroacetic acid; Solvent B: acetonitrile with 0.018% trifluoroacetic acid. Gradient: After 0.01 min at the initial condition of 90% A and 10% B, Solvent B was increased to 80% over 4 min, maintained at 80% for 0.9 min, and then a linear gradient to initial conditions was applied for 0.02 min and maintained for 0.58 min to re-equilibrate the column, giving a cycle time of 5.50 min. The flow rate was 0.8 mL/min from 0.01 to 4.90 min, increased to 1.2 mL/min in 0.03 min, and maintained until the end of the run.

**Protocols for preparative HPLC purification methods.** The HPLC measurement was performed using a SHIMADZU preparative HPLC system, an autosampler, and a UV detector. The fractions were detected by LC-MS. The MS detector was configured with an electrospray ionization source. The source temperature was maintained at 300 °C–350 °C. Reversed-phase HPLC was performed on a Luna C18 column (250×50 mm; 10 μm). Solvent A: water with 0.1% trifluoroacetic acid (TFA); Solvent B: acetonitrile.

Method 1: Gradient: At room temperature, 25% to 55% B over 20 min; then 100% B over 15 min. Flow rate: 80 mL/min.

Method 2: Gradient: At room temperature, 10% to 40% B over 20 min; then 100% B for 15 min. Flow rate: 80 mL/min.

**High-resolution mass spectrometry of final compounds.** High-resolution mass spectrometry (HRMS) $m/z$ was performed on an Agilent Technologies 1200 liquid chromatographic system equipped with a 6220 Accurate-Mass time of flight (TOF) LC/MS, operated in positive electrospray ionization mode (ESI+) controlled by MassHunter Workstation 06.00 software (Agilent Technologies, Barcelona, Spain). The separation was performed on a Zorbax SB-C18 (15 cm × 0.46 cm; 5 μm) from Agilent Technologies with a SB-C18 precolumn from Teknokroma (Barcelona, Spain). Solvent A: water with 0.1% formic acid; Solvent B: acetonitrile. The gradient elution was 5–100% B from 0–25.0 min; 100% B from 25–26 min; 100–5% B from 26–28 min; and 5% B from 28–30 min. The injection volume was 10 μL, and the flow rate was 0.5 mL min⁻¹ during the complete run. Chromatography was performed at 40 °C. The ESI conditions were as follows: gas temperature, 350 °C; drying gas, 10 L min⁻¹; nebulizer, 45 psig; capillary voltage, 3500 V; fragmentor, 175 V; and skimmer, 65 V. Acquisition was from 100–1000 m/z at a rate of 1.03 spectra s⁻¹. All solvents used were liquid chromatography–mass spectrometry (LC-MS)-grade and obtained from Scharlau (Scharlab, Sentmenat, Spain). Water (18.2 MΩ) was obtained from an Ultramatic system from Wasserlab (Barbatáin, Spain).

The synthetic process of the small molecules CM-444 and CM-1758 is outlined in Supplementary Fig. 10.

**Synthesis of 2,4-Dichloro-6,7-dimethoxy-quinoline (2).** A mixture of commercially available 3,4-dimethoxyaniline (**1**) (20.00 g, 130.57 mmol) and malonic acid (20.38 g, 195.85 mmol, 20.38 mL) in $POCl_3$ (100 mL) was degassed and purged with $N_2$ three times, and then the mixture was stirred at 100 °C for 16 h. The mixture was cooled

to 20 °C and concentrated in reduced pressure at 40 °C. The residue was poured into $H_2O$ (300 mL) and extracted with DCM (200 mL × 3). The combined organic phase was washed with brine (300 mL × 2), dried with anhydrous $Na_2SO_4$, filtered and concentrated under a vacuum. The residue was purified by silica gel column chromatography (petroleum ether/EtOAc = 20:1 to 1:1) to afford pure compound **2** (20.00 g, 59%) as a yellow solid. ¹H NMR (CDCl₃, 400 MHz): δ 7.38 (s, 1H), 7.37–7.36 (m, 2H), 4.06 (s, 3H), 4.03 (s, 3H). ESI-MS $m/z$ calcd. for $C_{11}H_9Cl_2NO_2$: 257.0, found 258.0 [M + H]⁺.

**Synthesis of 4-Chloro-6,7-dimethoxy-2-(5-methyl-2-furyl)quinoline (3a).** A mixture of compound **2** (5 g, 19.37 mmol), commercially available 4,4,5,5-tetramethyl-2-(5-methyl-2-furyl)-1,3,2-dioxaborolane (4.03 g, 19.37 mmol), $K_2CO_3$ (5.35 g, 38.74 mmol), Pd(PPh₃)₄ (1.12 g, 968.62 μmol) in 1,4-dioxane/$H_2O$ (8:1, 90 mL) was degassed and purged with N2 three times, and then the mixture was stirred at 100 °C for 12 h. The mixture was filtered, and the filtrate was concentrated under vacuum. The residue was purified by flash silica gel chromatography (ISCO®; 80 g SepaFlash® Silica Flash Column, eluent of 0%–40% EtOAc/petroleum ether gradient at 100 mL/min) to afford pure compound **3a** (4.20 g, 71%) as a light-yellow solid. ¹H NMR (CDCl₃, 400 MHz): δ 7.74 (s, 1H), 7.46 (s, 1H), 7.38 (s, 1H), 7.04 (d, $J$ = 3.1 Hz, 1H), 6.18 (d, $J$ = 2.4 Hz, 1H), 4.07 (s, 3H), 4.05 (s, 3H), 2.46 (s, 3H). ESI-MS $m/z$ calcd. for $C_{16}H_{14}ClNO_3$: 303.1 found 304.1 [M + H]⁺.

**Synthesis of Ethyl 2-[4-[[[6,7-dimethoxy-2-(5-methyl-2-furyl)-4-quinolyl]amino]methyl]-1-piperidyl]pyrimidine-5-carboxylate (4a).** A mixture of compound **3a** (1 g, 3.29 mmol), commercially available ethyl 2-[4-(aminomethyl)-1-piperidyl]pyrimidine-5-carboxylate (1.13 g, 4.28 mmol), $Cs_2CO_3$ (3.22 g, 9.88 mmol), BINAP (615.00 mg, 987.69 μmol) and Pd₂(dba)₃ (602.96 mg, 658.46 μmol) in 1,4-dioxane (30 mL) was degassed and purged with $N_2$ three times, and then the mixture was stirred at 120 °C for 12 h. The mixture was filtered and the filtrate was concentrated under vacuum. The residue was purified by flash silica gel chromatography (ISCO®; 40 g SepaFlash® Silica Flash Column, eluent of 0%–100% EtOAc/Petroleum ether gradient at 100 mL/min) to afford pure compound **4a** (1.5 g, 86%) as a light-yellow solid. The purity was 97.21% according to the HPLC analytical method (described above), and the retention time ($t_R$) was 2.948 min. ¹H NMR (MeOD, 400 MHz): δ 8.81 (s, 2H), 7.71 (s, 1H), 7.53 (d, $J$ = 3.4 Hz, 1H), 7.46 (s, 1H), 7.01 (s, 1H), 6.44 (d, $J$ = 2.8 Hz, 1H), 4.99 (d, $J$ = 13.4 Hz, 2H), 4.34 (q, $J$ = 7.2 Hz, 2H), 4.05 (d, $J$ = 3.0 Hz, 6H), 3.59 (d, $J$ = 6.9 Hz, 2H), 3.05 (t, $J$ = 12.0 Hz, 2H), 2.53 (s, 3H), 2.26 (br s, 1H), 2.01 (d, $J$ = 11.8 Hz, 2H), 1.40–1.36 (m, 5H). ESI-MS $m/z$ for $C_{29}H_{33}N_5O_5$ calcd.: 531.2, found: 532.4 [M + H]⁺.

**Synthesis of 2-[4-[[[6,7-dimethoxy-2-(5-methyl -2-furyl)-4-quinolyl]amino]methyl]-1-piperidyl]pyrimidine-5-carboxylic acid (5a).** A mixture of compound **4a** (1.5 g, 2.82 mmol) and LiOH•$H_2O$ (591.98 mg, 14.11 mmol) in THF/$H_2O$ (2:1, 30 mL) was stirred at 25 °C for 12 h. Then, the reaction mixture was adjusted to pH 5 with aqueous HCl (2.0 M) at room temperature, and the precipitate was filtered to afford compound **5a** (1.2 g, 84%) as a yellow solid. The purity was 97.20% according to the HPLC analytical method (described above); and the $t_R$ was 2.373 min. ¹H NMR (MeOD, 400 MHz): δ 8.80 (s, 2H), 7.69 (s, 1H), 7.51 (d, $J$ = 3.4 Hz, 1H), 7.44 (s, 1H), 6.99 (s, 1H), 6.43 (d, $J$ = 3.3 Hz, 1H), 4.97 (d, $J$ = 13.1 Hz, 2H), 4.03 (s, 3H), 4.02 (s, 3H), 3.57 (d, $J$ = 7.2 Hz, 2H), 3.03 (t, $J$ = 11.7 Hz, 2H), 2.51 (s, 3H), 2.24 (br s, 1H), 1.99 (d, $J$ = 12.7 Hz, 2H), 1.41–1.32 (m, 2H). ESI-MS $m/z$ calcd. for $C_{27}H_{29}N_5O_5$: 503.2, found 504.3 [M + H]⁺.

**Synthesis of 2-[4-[[[6,7-dimethoxy-2-(5-methyl-2-furyl)-4-quinolyl]amino]methyl]-1-piperidyl]pyrimidine-5-carbohydroxamic acid (6a, CM-444).** A mixture of compound **5a** (1.2 g, 2.38 mmol), O-tetrahydropyran-2-ylhydroxylamine (558.34 mg, 4.77 mmol), HOBt

(644.01 mg, 4.77 mmol), EDCI (913.68 mg, 4.77 mmol) and DIEA (1.54 g, 11.92 mmol, 2.08 mL) in DMF (30 mL) was degassed and purged with $N_2$ three times, and then the mixture was stirred at 25 °C for 12 h. Next, 0.5 M aqueous HCl (5 mL) was added in one portion at room temperature. The mixture was concentrated under vacuum to give a residue, which was purified by preparative HPLC (General procedure described above, method 1) to afford pure compound **6a** (**CM-444**, 794.3 mg, 62%) as a light-yellow solid, m.p. 136 °C–137 °C. The purity was 96.65% according to the HPLC analytical method (described above), and the $t_R$ was 2.077 min. $^1$H NMR (MeOD, 400 MHz): δ 8.65 (s, 2H), 7.69 (s, 1H), 7.51 (d, $J$ = 3.5 Hz, 1H), 7.44 (s, 1H), 6.99 (s, 1H), 6.43 (d, $J$ = 3.3 Hz, 1H), 4.94 (br s, 2H), 4.03 (s, 3H), 4.02 (s, 3H), 3.57 (d, $J$ = 7.2 Hz, 2H), 3.01 (t, $J$ = 11.7 Hz, 2H), 2.51 (s, 3H), 2.23 (br s, 1H), 1.98 (d, $J$ = 11.9 Hz, 2H), 1.40–1.35 (m, 2H). $^{13}$C NMR (DMSO-$d_6$, 100 MHz): δ 161.2, 157.0 (2 C), 156.7, 154.0, 153.8, 148.6, 144.0, 139.1, 134.3, 116.1, 114.1, 109.9 (2 C), 102.2, 100.3, 91.7, 56.3, 55.9, 47.9, 43.3 (2 C), 35.5, 29.3 (2 C), 13.6. HRMS $m/z$: [(M + H)]$^+$ calcd. for $C_{27}H_{30}N_6O_5$, 519.2350; found, 519.2400.

**Synthesis of 4-chloro-2-(2,5-dimethyl-3-furyl)-6,7-dimethoxy-quinoline (3b).** A mixture of compound **2** (2.00 g, 7.75 mmol), commercially available 2-(2,5-dimethyl-3-furyl)-4,4,5,5-tetramethyl-1,3,2-dioxaborolane (1.72 g, 7.75 mmol), $K_2CO_3$ (2.14 g, 15.50 mmol), Pd(PPh$_3$)$_4$ (895.43 mg, 775.00 μmol) in 1,4-dioxane/H$_2$O (10:1, 55 mL) was degassed and purged with $N_2$ three times, and then the mixture was stirred at 90 °C for 16 h. The reaction mixture was concentrated under vacuum to give a residue. The residue was purified by silica gel column chromatography (petroleum ether/EtOAc = 20:1 to 5:1) to afford pure compound **3b** (1.9 g, 77%) as a white solid. $^1$H NMR (CDCl$_3$, 400 MHz): δ 7.46 (s, 1H), 7.39–7.36 (m, 2H), 6.39 (s, 1H), 4.06 (m, 6H), 2.69 (s, 3H), 2.31 (s, 3H). ESI-MS $m/z$ calcd. for $C_{17}H_{16}ClNO_3$: 317.1, found 318.1 [M + H]$^+$.

**Synthesis of ethyl 2-[4-[[[2-(2,5-dimethyl-3-furyl)-6,7-dimethoxy-4-quinolyl] amino]methyl]-1-piperidyl]pyrimidine-5-carboxylate (4b).** A mixture of compound **3b** (1 g, 3.15 mmol), ethyl 2-[4-(aminomethyl)-1-piperidyl]pyrimidine-5-carboxylate (1.08 g, 4.09 mmol), Cs$_2$CO$_3$ (3.08 g, 9.44 mmol), BINAP (587.86 mg, 944.09 μmol) and Pd$_2$(dba)$_3$ (576.35 mg, 629.39 μmol) in 1,4-dioxane (30 mL) was degassed and purged with $N_2$ three times, and then the mixture was stirred at 120 °C for 12 h. The mixture was filtered and the filtrate was concentrated under vacuum. The residue was purified by flash silica gel chromatography (ISCO®; 40 g SepaFlash® Silica Flash Column, eluent of 0%–100% EtOAc/petroleum ether gradient at 100 mL/min) to afford pure compound **4b** (1.2 g, 70%) as a light-yellow solid. $^1$H NMR (CDCl$_3$, 400 MHz): δ 8.84 (s, 2H), 7.37 (s, 1H), 6.87 (br s, 1H), 6.51 (br s, 1H), 6.38 (br s, 1H), 4.99 (d, $J$ = 13.2 Hz, 2H), 4.73 (br s, 1H), 4.35 (q, $J$ = 7.1 Hz, 2H), 4.03 (s, 6H), 3.32 (t, $J$ = 6.2 Hz, 2H), 2.99 (t, $J$ = 11.8 Hz, 2H), 2.65 (s, 3H), 2.32 (s, 3H), 2.14 (br s, 1H), 2.00 (d, $J$ = 11.5 Hz, 2H), 1.41–1.32 (m, 5H). ESI-MS $m/z$ calcd. for $C_{30}H_{35}N_5O_5$: 545.3, found 546.3 [M + H]$^+$.

**Synthesis of 2-[4-[[[2-(2,5-dimethyl-3-furyl)- 6,7-dimethoxy-4-quinolyl]amino]methyl]-1-piperidyl]pyrimidine-5-carboxylic acid (5b).** A mixture of compound **4b** (1.2 g, 2.20 mmol) and LiOH•H$_2$O (461.41 mg, 11.00 mmol) in THF/H$_2$O (2:1, 30 mL) was stirred at 25 °C for 12 h. Then, the reaction mixture was adjusted to pH 5 with aqueous HCl (2.0 M) at room temperature, and the precipitate was filtered to afford compound **5b** (1.1 g, 97%) as a yellow solid. ESI-MS $m/z$ calcd. for $C_{28}H_{31}N_5O_5$: 517.2, found 518.3 [M + H]$^+$.

**Synthesis of 2-[4-[[[2-(2,5-dimethyl-3-furyl)-6,7-dimethoxy-4-quinolyl]amino]methyl] -1-piperidyl]pyrimidine-5-carbohydroxamic acid (6b, CM-1758).** A mixture of compound **5b** (1.1 g, 2.13 mmol), O-tetrahydropyran-2-ylhydroxylamine (497.94 mg, 4.25 mmol), HOBt

(574.34 mg, 4.25 mmol), EDCI (814.84 mg, 4.25 mmol) and DIEA (1.37 g, 10.63 mmol, 1.85 mL) in DMF (15 mL) was degassed and purged with $N_2$ three times, and then the mixture was stirred at 25 °C for 12 h. The residue was poured into ice water (w/w = 1/1) (30 mL). The aqueous phase was extracted with ethyl acetate (20 mL ×3). The combined organic phase was washed with brine (30 mL) and dried with anhydrous Na$_2$SO$_4$. Then, aqueous HCl (2 M, 3 mL) was added in one portion at room temperature and concentrated under vacuum to give a residue, which was purified by preparative HPLC (General procedure described above, method 2) to afford pure compound **6b** (**CM-1758**, 503.7 mg, 43%) as an off-white solid. m.p. 176 °C–177 °C. The purity was 96.29% according to the HPLC analytical method (described above), and the $t_R$ was 2.073 min. $^1$H NMR (MeOD, 400 MHz): δ 8.65 (s, 2H), 7.71 (s, 1H), 7.32 (s, 1H), 6.68 (s, 1H), 6.47 (s, 1H), 4.92 (d, $J$ = 13.2 Hz, 2H), 4.03 (s, 6H), 3.53 (d, $J$ = 7.1 Hz, 2H), 2.99 (t, $J$ = 11.9 Hz, 2H), 2.56 (s, 3H), 2.35 (s, 3H), 2.21 (br s, 1H), 1.96 (d, $J$ = 11.0 Hz, 2H), 1.38–1.30 (m, 2H). $^{13}$C NMR (DMSO-$d_6$, 100 MHz): δ 161.2, 157.0 (2 C), 154.1, 153.9, 151.4, 150.6, 148.8, 144.3, 134.8, 115.3, 114.2, 109.8, 106.3, 102.1, 100.2, 96.3, 56.3, 56.0, 47.8, 43.3 (2 C), 35.2, 29.3 (2 C), 13.3, 12.9. HRMS $m/z$: [(M + H)]$^+$ calcd. for $C_{28}H_{32}N_6O_5$, 533.2507 found, 533.2549.

**1H-NMR of CM-444**

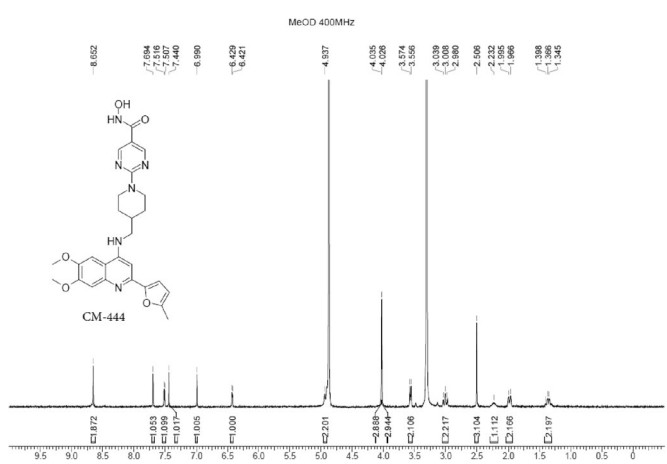

**HPLC trace of CM-444**

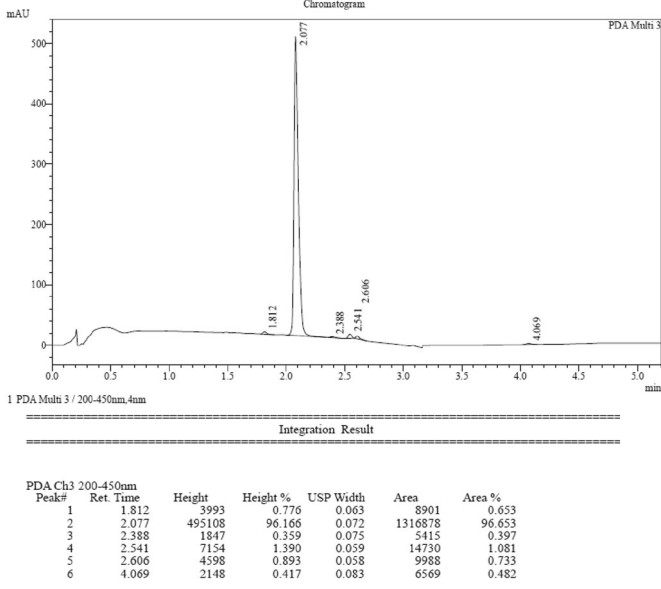

### 1H-NMR of CM-1758

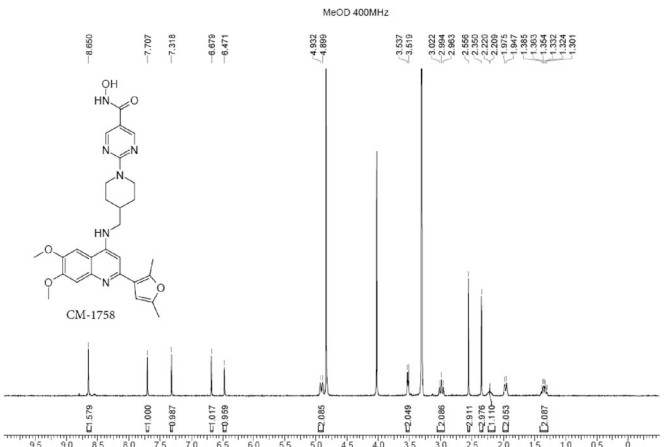

### HPLC trace of CM-1758

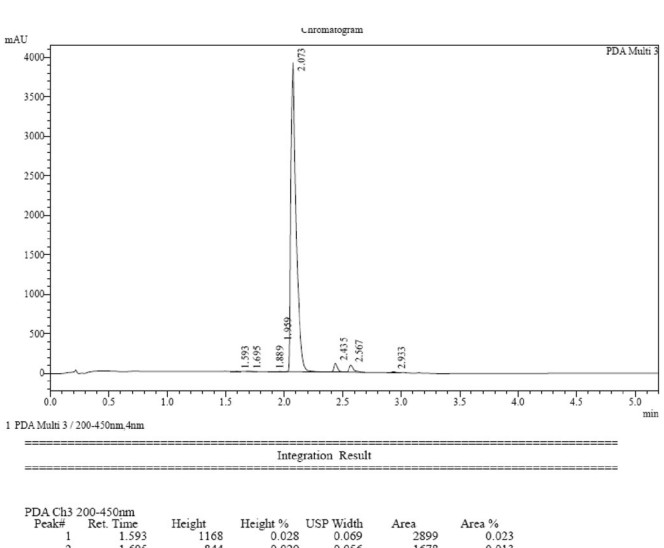

### Cell proliferation assay: GI$_{50}$ calculation

Cell proliferation was analyzed using the CellTiter 96 Aqueous One Solution Cell Proliferation Assay (Promega, Madison, WI). This is a colorimetric method for determining the number of viable cells in proliferation. For the assay, suspension cells were cultured in triplicate at a density of $1 \times 10^6$ cells/mL in 96-well plates (100.000 cells/well, 100 µL/well), using only the 60 inner wells to avoid any border effects. After 48 h of treatment with different doses of each compound (0, 0.001, 0.01, 0.1, 0.2, 0.4, 0.6, 0.8, 1, 2, 3, 4, 5 and 10 µM), the plates were centrifuged at 800 $g$ for 10 min followed by removal of the medium. Then, the cells were incubated with 100 µL/well of medium and 20 µL/well of CellTiter 96 Aqueous One Solution reagent for 1–3 h. The absorbance was recorded at 490 and 650 nm as reference wavelengths using 96-well plate readers until the absorbance of the control cells without treatment was approximately 0.8 AU. The background

absorbance was measured in wells with only cell line medium and solution reagent. First, the average of the absorbance from the control wells was subtracted from all other absorbance values. Data were calculated as the percentage of total absorbance of treated cells/absorbance of non-treated cells. The GI$_{50}$ values of the different compounds were determined using non-linear regression plots made with Graph-Pad Prism v5 software.

### Flow cytometry

For CD11b / annexin-V detection, 100,000 cells of ML-2, KASUMI-1, HL-60, NB-4, OCI-AML3, MV4-11, MOLM-13, HEL, GF-D8, TF-1, THP-1, M-O7e, OCI-AML2, MONO-MAC-6 and UT-7 cell lines were cultured at a density of $1 \times 10^6$ cells/mL and treated daily for 48 h with CM-444, CM-1758, or ATRA (see Supplementary Table 2 for GI$_{50}$ data). The cell lines HL-60, ML-2, MV4-11 and MOLM-13 were also treated daily up to 8 days with CM-444 and CM-1758 at 25% GI$_{50}$. HL-60, ML-2, MOLM-13 and MV4-11 cell lines were also treated for 48 h with 25% GI$_{50}$ of Panobinostat, Quisinostat, Entinostat, Vorinostat, or Tubastatin (Supplementary Table 9) or with the BRD inhibitors JQ1, Molibresib, and the combination of CM-444 or CM-1758 with JQ1 and Molibresib (Supplementary Table 9). Next, the cells were washed twice with phosphate-buffered saline (PBS) and resuspended in 1X Binding Buffer at a concentration of $1 \times 10^6$ cells/mL. An 8-µL aliquot of APC-CD11b (BD Pharmingen, Franklin Lakes, NJ, USA) and 1 µL of FITC annexin-V (AV) (BD Pharmingen, Franklin Lakes, NJ, USA) antibodies were added and incubated for 15 min at room temperature in the dark. Finally, after the addition of 400 µL of 1X Binding Buffer to each tube, samples were collected on a BD FACSCanto flow cytometer (Becton Dickinson, San Jose, CA, USA) and analyzed using FlowJo software.

For the rest of experiments, the mix of antibodies (2 µL of BV786-CD13, Cat No, 744748 BD; 1 µL of diluted 1:10 antibody of PB-HLADR, Cat No 307633, Biolegend; 5 µL of APCH7-CD14, Cat No 641394, BD) was added to 10 µL of Brilliant Stain Buffer Plus (Becton Dickinson, San Jose, CA, USA). Then, 1 million cells (volume of 250 µL) were added and incubated for 30 min at room temperature in darkness. After 10 min incubation at room temperature with 2 mL of FACSLysing 1X, cells were centrifuged at 540 $g$ for 5 min and washed twice with FACSBuffer. Finally, after centrifugation at 540 $g$ for 5 min cells were resuspended in FACSBuffer.

The detailed gating strategy for flow cytometry is shown in Supplementary Fig. 11.

### Enzymatic assays

HDAC1, HDAC2, HDAC3 and HDAC6 IC$_{50}$ values of compounds CM-444 and CM-1758 were determined using a specific fluorescence-labeled substrate (BPS Biosciences, Cat # 50037) after its deacetylation by HDAC, as described previously[28]. Results are the average of two biological replicates.

The HDAC4, HDAC5, HDAC7, HDAC8, HDAC9, HDAC10 and HDAC11 percent inhibition at 10 µM and IC$_{50}$ values were determined by Eurofins (https://www.eurofins.com/), in duplicate.

HDAC11 IC$_{50}$ values of CM-444 and CM-1758 were determined by a continuous and direct activity assay for HDAC11 based on internal fluorescence quenching[48]. Results are the average of two biological replicates.

DNMT1 and G9a IC$_{50}$ values were measured using time-resolved fluorescence energy transfer assays, as described previously[18,25,26]. The results are the average of two biological replicates.

The selectivity of CM-444 and CM-1758 against other epigenetic targets, including histone demethylase (JMJD2A, JMJD2B, JMJD2C, JMJD2D, JMJD2E, JMJD3, JMJD1A, LSD1, Jarid1A, Jarid1B, Jarid1C, FBXL10, FBXL11, JHDM1D, JMJD1B, PHF8, LSD2 and UTX); histone acetyltransferases (GCN5, P300); histone methyltransferases (G9a, GLP, MLL1, SET7/9, SUV39H1, SUV39H2, PRMT1, PRMT3, PRMT4, PRMT5, PRMT6, PRMT8, PRMT9, NSD1, NSD2, EZH1, EZH2, SETDB1, SETD2,

SET8, SUV4-20H1, SMYD2, SMYD3, DOT1L, EZH2 (A677G), EZH2 (A677G), EZH2 (A738T), EZH2 (Y641S), EZH2 (P132S), EZH2 (Y641C), EZH2 (Y641F), EZH2 (Y641H), EZH2 (Y641N)); bromodomains (ATAD2B, ATAD2A, BRD1, BRD9, BPTF/FALZ, CeCR2, BRPF3, BRD3 (BD1 + BD2), BAZ2B, WDR9, TAF1 (BD2), TAF1L (BD2), TAF1L (BD1 + BD2), BRD2 (BD1), BRD2 (BD2), BRD2 (BD1 + BD2), BRD3 (BD1), BRD3 (BD2), BRD4 (BD1), BRD4 (BD2), BRD4 (BD1 + BD2), BRDT (BD1), BRDT (BD1 + BD2), CREBBP, SMARCA2A, SMARCA4/BRG1); DNA methyltranferases (DNMT1, DNMT3A/3L and DNMT3B/3L), and sirtuin histone deacetylases (SIRT1, SIRT2, SIRT3, SIRT5 and SIRT6), were determined by BPS Bioscience (http://www.bpsbioscience.com/index.ph). Binding experiments were performed in duplicate at a concentration of 10 µM. If the percentage of inhibition was >50%, dose-response $IC_{50}$ values were determined (in duplicate).

### Docking of CM-444 and CM-1758 against HDAC isoforms

The GoldSuite 5.3.6 program (Cambridge Crystallographic Data Centre, https://www.ccdc.cam.ac.uk/pages/Home.aspx) was used to perform docking of CM-444 and CM-1758 to different HDAC isoforms with available crystallographic structures: HDAC1 (PDB entry 4BKX[49]), CD2 domain of HDAC6 complexed with trichostatin A (PDB entry 5EDU[50]), and HDAC7 complexed with trichostatin A (PDB entry 3C10[51]). The binding site was defined as a 20 Å sphere around the Zn atom of the catalytic cavity of HDACs and the remaining configuration parameters as described previously[52].

### Western blot

After treating HL-60, ML-2, MV4-11 and MOLM-13 cell lines daily for 48 h with 25% $GI_{50}$ of CM-444 and CM-1758.48 h, the cells were washed twice with PBS, the last centrifugation of 1520 g for 10 min at 4 °C. Histone extraction was performed with a Histone extraction Kit (Abcam, Cambridge, UK) following the manufacturer's instructions. The histone concentration in the extract was measured using the dye-binding assay of Bradford. A 10 µg amount of histone was separated on 15% sodium dodecyl sulfate-polyacrylamide gel electrophoresis and transferred to a nitrocellulose membrane. The membrane, after being blocked with Tropix I-block blocking reagent (Cat No AI300, Tropix) in PBS with 0.1% Tween 20 and 0.02 $NaN_3$ was incubated with the primary antibody against Acetyl H3 (rabbit polyclonal antibody, Cat No 06-599, Millipore) diluted 1:50,000 o/n at 4 °C or against H3K27 me3 (mouse monoclonal antibody to histone H3 trimethyl K27, Cat No ab6002, Abcam) diluted 1:2000 in bovine serum albumin o/n at 4 °C and then with alkaline phosphatase-conjugated secondary antibodies. Bound antibodies were revealed by a chemiluminiscent reagent (Tropix) and visualized using the Chemidoc Imaging Systems (Bio-Rad Laboratories). Total H3 was used as a loading control (diluted 1:50,000 o/n at 4 °C or for 1 h at room temperature) (Anti-Histone H3, CT, pan, rabbit polyclonal, Cat No 07-690, Millipore). Images have been cropped for presentation.

### Dot blot

After treating HL-60, ML-2, MV4-11 and MOLM-13 cell lines daily for 48 h with 25% $GI_{50}$ of CM-444 and CM-1758.48 h, the cells were washed twice with PBS, and genomic DNA was extracted using a DNA Kit (Nucleo Spin Tissue, Cat No 74095250, Macherey-Nagel) following the manufacturer's instructions. DNA purity and concentration were measured using a NanoDrop spectrophotometer (Thermo Scientific). A 500 ng amount of genomic DNA was loaded onto a nitrocellulose membrane (Amersham Hybond_N+, RPN203B, GE Healthcare), pre-wetted in 6X SSC for 10 min, using the Bio-Dot microfiltration apparatus (Cat No 170-6545, Bio-Rad) following the manufacturer's instructions. Next, the membrane was incubated with 2X SSC for 5 min and cross-linked for 2 h at 80 °C. The membrane, after being blocked with Tropix I-block blocking reagent (Cat No AI300, Tropix) in PBS with 0.1% Tween 20 and 0.02 $NaN_3$, was incubated with the primary antibody against 5-methylcytosine (monoclonal antibody 5-methylcytidine, Cat No BI-MECY-1000, Eurogentec) diluted 1:4000 o/n at 4 °C and then with alkaline phosphatase-conjugated secondary antibody. Bound antibodies were revealed by a chemiluminiscent reagent (Tropix) and visualized using the Chemidoc Imaging Systems (Bio-Rad Laboratories). To ensure equal loading of total DNA on the membrane, the same blot was stained with 0.02% methylene blue in 0.3 M sodium acetate (pH 5.2).

### LINE-1 pyrosequencing

DNA methylation of the repetitive element *LINE-1* was analyzed using a pyrosequencing technique. First, HL-60, ML-2, MV4-11 and MOLM-13 cell lines were daily treated for 48 h with 25% $GI_{50}$ of CM-444 and CM-1758. Cells were washed twice with PBS, and then genomic DNA was extracted using a DNA Kit (Nucleo Spin Tissue, Cat No 74095250, Macherey-Nagel) following the manufacturer's instructions. DNA purity and concentration were measured using a NanoDrop spectrophotometer (Thermo Scientific). A 1 µg amount of genomic DNA was treated and modified using a CpGenome DNA modification Kit (Cat No S7820, Chemicon International) following the manufacturer's instructions. After bisulfite modification, a "hot start" polymerase chain reaction (PCR) (PyroMark PCR Kit, Cat No 978703, Qiagen) was performed with denaturalization at 95 °C for 15 min, followed by 45 cycles consisting of denaturation at 94 °C for 1 min, annealing at 55 °C for 1 min, and extension at 72 °C for 1 min followed by a final 10-min extension. This PCR was performed using 2 µL of modified DNA, 12.5 µL of 2X Buffer, 1 µL of 10 µM of each specific primer (final concentration of 0.4 µM) (LINE-1-F: 5'- TTTTGAGTTAGGTGTGGGATATA -3' and LINE-1-R: 5'-Biotin- AAAATCAAAAAATT CCCTTTC -3') in a final volume of 25 µL. The resulting biotinylated PCR products were immobilized to Streptavidin Sepharose ® High-Performance beads (GE Healthcare) and processed to yield high-quality ssDNA using the PyroMark Vacuum Prep Workstation (Biotage) according to the manufacturer's instructions. The pyrosequencing reactions were performed using the PyromarkTM ID (Biotage), and sequence analysis was performed using the PyroQ-CpG analysis software (Biotage).

### Primary samples

Peripheral blood (PB) specimens were obtained from preliminarily diagnosed AML patients who had not received any treatment before specimens were collected. All patients gave informed consent for their participation in this study, which was approved by the Clinical Research Ethics Committee of Clínica Universidad de Navarra. Written informed consent was obtained from each participant. PB was mixed with an ammonium chloride-potassium (ACK) buffer lysis (0.15 mol/L $NH_4Cl$, 10 mmol/L $KHCO_3$, 0.1 mmol/L ethylenediaminetetraacetic acid, pH 7.2) at a 1:2 volume ratio by gentle inversion, incubated for 10 min and centrifuged at 200 g for 10 min. The cell pellet was washed twice with PBS and finally resuspended in EGM2 medium (Cambrex Bioscience, Walkersville, MD). EGM2 contains 2% fetal calf serum, hydrocortisone, heparin, antibiotics, epidermal growth factor, human fibroblast growth factor, insulin-like growth factor, vascular endothelial growth factor, and ascorbic acid. The cells were plated at $1 \times 10^6$ cells/mL and treated daily with 200 and 500 nM of CM-444 and CM-1758 for 48 h. Next, the differentiation assay was performed as described above.

### Quantitative polymerase chain reaction (q-RT-PCR)

The expression of *MYC*, *CDKN2A*, *CDKN1A*, *GATA2*, *TAL1*, *CEBPA* and *SPI1* were analyzed by q-RT-PCR in HL-60, ML-2, MOLM-13 and MV4-11 cell lines after 48 h of CM-444, CM-1758, JQ1, Molibresib and the combinations of CM-444 or CM-1758 with JQ1 and Molibresib treatments. RNA was extracted with TRIzol Reagent (Invitrogen) according to the manufacturer's instructions. First, cDNA was synthesized from 1 µg of total RNA using the PrimeScript RT reagent Kit (Perfect Real

Time) (Cat No RR037A, TaKaRa) following the manufacturer's instructions. The quality of cDNA was checked by a multiplex PCR that amplifies *PBGD*, *ABL*, *BCR* and *β2-MG* genes. A QuantStudio 5 Real-Time PCR System (Applied Biosystems) using 20 ng of cDNA in 2 µL, 1 µL of each specific primer at 10 µM and 5 µL of SYBR Green PCR Master Mix 2X (Cat No 4334973, Applied Biosystems) in a 10-µL reaction volume was used to perform q-RT-PCR. The following program conditions were used for the q-PCR: 50 °C for 2 min, 95 °C for 60 s following by 45 cycles at 95 °C for 15 s and 60 °C for 60 s; melting program, one cycle at 95 °C for 15 s, 40 °C for 60 s, and 95 °C for 15 s. The relative expression of each gene was quantified by the log $2(-\Delta\Delta Ct)$ method using the gene *GUS* as an endogenous control. The sequence of primers used can be found in Supplementary Table 10.

### Cell-cycle analysis

For cell-cycle analysis, 250,000 cells of HL-60, ML-2, MV4-11 and MOLM-13 cell lines were cultured at a density of $1 \times 10^6$ cells/mL and treated for 24 h with CM-444 and CM-1758. Next, the cells were washed twice with PBS and resuspended in 0.2% Tween 20 in PBS and 0.5 mg/mL Rnase A (Ribonuclease A Type III-A from bovine pancreas, Cat No. R5125, Sigma), and then incubated for 30 min at 37 °C. Subsequently, the cells were stained with 25 µg/mL of propidium iodide (Cat No P4170, Sigma) and analyzed using a BD FACSCanto flow cytometer (Becton Dickinson, San Jose, CA, USA).

### May−Grünwald Giemsa staining

After treatment of the HL-60, ML-2, MV4-11 and MOLM-13 cell lines daily with CM-444 and CM-1758 at 25% $GI_{50}$ concentration for 96 h, the cells were washed twice with PBS and resuspended at 20,000 cells in 100 µL. Cytospins were prepared using a cytospin centrifuge and we stained slides with May−Grünwald Giemsa (Sigma, St. Louis, MO, USA). We examined cellular morphology using a Nikon Eclipse 90i microscope (Nikon, Melville, NY, USA).

### RNA-seq

Transcriptomic analyses were performed following low-input 3′ end RNA sequencing approximation for characterization of transcriptional changes in ML-2 and HL-60 samples treated with CM-444 and CM-1758.

Low-input 3′ end RNA-seq was performed as previously described[53,54] with minor modifications. Briefly, RNA from 0.1 ×106 cells from each experimental condition was purified using a Dynabeads mRNA DIRECT Purification Kit (Thermo Fisher Scientific). For barcoded cDNA generation, polyadenylated RNA was retrotranscribed using an Affinity Script cDNA Synthesis Kit (Agilent Technologies), and barcoded primers also harboring T7 promoter and a partial Read2 sequence for illumine sequencing (CGATTGAGGCCGGTAATA CGACTCACTATAGGGGGCGACGTGTGCTCTTCCGATCTXXXXXXNNN NTTTTTTTTTTTTTTTTTTTTTT, where XXXXXX is the cell barcode and NNN is a unique molecular identifier or UMI, which represents the same initial RNA molecule for future quantification). At this point, the samples were pooled (≤6 samples per pool) at equimolar ratios assessed by q-PCR.

The samples were then treated with Exonuclease I (Thermo Fisher Scientific following a 1.2x positive SPRI-selection. Second-strand cDNA synthesis was then performed using a NEBNext Ultra II Directional RNA Second-Strand Synthesis Module (New England Biolabs) for 2 h at 16 °C, followed by a 1.4x positive solid-phase reverse immobilization (SPRI) cleanup. The samples were linearly amplified by T7 in vitro transcription using a T7 RNA Polymerase (New England Biolabs) for 16 h at 37 °C. After DNAse treatment (TURBO DNAse I, Thermo Fisher Scientific) for 15 min at 37 °C to remove dsDNA, samples were cleaned with a positive 1.2x SPRI-selection. RNA size, quality, and concentration were determined by analysis with a Qubit Fluorometer (Termo Fisher Scientific) and TapeStation (Agilent 4200 TapeSation System, Agilent Technologies). If necessary, RNA was fragmented in 200- to 300-bp

fragments using Zn2+divalent cations followed by a 2x positive SPRI cleanup. Fragmented RNA was dephosphorylated using FastAP Thermosensitive Alkaline Phosphatase (Thermo Fisher Scientific) for 10 min at 37 °C to enhance RNA ligation yield. In this step, a partial Read1 (AGATCGGAAGAGCGTCGTGTAG) sequence for illumine sequencing was ligated to the 3′ end of the fragmented RNA using T4 RNA Ligase I (New England Biolabs). After a 1.5x positive SPRI cleanup, the RNA product was retrotranscribed using an Affinity Script cDNA Synthesis Kit (Agilent Technologies) and a specific retrotranscription primer (TCTAGCCTTCTCGCAGCACATC). The library was then purified by 1.5x positive SPRI cleanup, and the yield of the previous steps was determined by q-PCR. Finally, PCR using KAPA HiFi HotStar Ready Mix (Kapa Biosystems) was performed for second-strand cDNA synthesis, illumine primers addition (Read1-P5: AATGATACGGCGACC ACCGAGATCTACACTCTTTCCCTACACGACGCT CTTCCGATCT and Read2-P7: CAAGCAGAAGACGGCATACGAGATGTGACTGGAGTTCAGA CGTGTGCTCTTCCGATCT) and library amplification using the recommended cycle numbers to minimize over-amplification. The product was purified by a 0.7x positive SPRI cleanup. The quality of the final library was determined by q-PCR (based on the threshold cycle, or Ct, obtained for *GAPDH* (Fw: CCAGCAAGAGCACAAGAGGAA, Rv: GATTCAGTGTGGTGGGGG) or *GUSB* (Fw: CAAGTGCCTCCTGGACT GTT, Rv: TCCACCTTTAGTGTTCCCTGC) housekeeping genes calibrated per individual set of samples, Qubit Fluorometer (Thermo Fisher Scientific) and TapeStation (Agilent 4200 TapeStation System, Agilent Technologies). All libraries were pooled at equimolar ratios and sequenced on a NextSeq500 (Illumina) using 60 bp single-end reads at a minimum sequence depth of 10 million reads per sample.

Raw reads were demultiplexed using bcl2fastq2 Conversion Software v2.19 (Illumina) to convert base call (BCL) files into FastQ files. The quality control of FastQ files was performed using FastQC (Bioinformatics Babraham Institute). Sequencing reads were aligned to hg19 human reference using Bowtie 2 (Johns Hopkins University) and quantified using quant3p script (github.com/ctlab/quant3p). The DEseq2 package (R) was used for filtering, normalization, and analysis of differential gene expression.

### Functional analysis

The Database for Annotation, Visualization, and Integrated Discovery database (david.ncifcrf.gov/home.jsp) was used to perform gene ontology (GO). A false discovery rate (FDR) < 0.05 was considered to be significant.

The fold change ranked list of genes was used as input to the non-parametric Kolmogorov−Smirnoff rank test as implemented in the GSEA software.

### CYP inhibition

This study was conducted at Wuxi (http://www.wuxi.com/). The inhibitory effects of CM-444 and CM-1758 on five human cytochrome P450s (1A2, 2C9, 2C19, 2D6 and 3A4) were evaluated in human liver microsomes. Test compounds (10 µM final concentration) or control compounds (α-naphthoflavone, sulfaphenazole, (+)-N-3-benzylnirvanol, quinidine, ketoconazole; 3-µM final concentration), and the corresponding substrates for each P450 isoform (1A2: phenacetin, 10-µM final concentration; 2C9: diclofenac, 5-µM final concentration; 2C19: S-mephenytoin, 30-µM final concentration, 2D6: dextromethorphan, 5-µM final concentration; 3A4: midazolam, 2-µM final concentration) were incubated with human liver microsomes (0.2-mg/mL final concentration; Corning Cat # 452117) and NADPH cofactor (1-mM final concentration, Chem-impex Cat # 00616) for 10 min at 37 °C. The reaction was terminated by adding 400 µL of cold stop solution (200 ng/mL tolbutamide and labetalol in acetonitrile), and the samples were centrifuged at 1520 $g$ for 20 min. Next, 200-µL aliquots of supernatants were diluted with 100 µL of ultra-pure water and shaken for 10 min. The samples were analyzed by LC-MS/MS using an analyte/

internal standard peak area ratio. Test compounds and positive controls were tested in duplicate. The percentage of inhibition was calculated as the ratio of substrate metabolite detected in treated and non-treated wells.

## Plasma protein binding

This study was conducted by Equilibrium Dialysis at Wuxi (http://www.wuxi.com/). HT Dialysis plate (Model HTD 96 b, Cat# 1006) and the dialysis membrane (molecular weight cutoff of 12–14 kDa, Cat# 1101) were purchased from HT Dialysis LLC (Gales Ferry, CT). Human plasma (BioIVT, # HUMANPLK2P2N) and CD-1 mouse plasma (Beijing Vital River Laboratory Animal Technology Co., Ltd.) were thawed prior to experiments under running cold tap water and centrifuged at $3320 \times g$ for 5 min to remove any clots, and the pH was checked. Only plasma from pH 7.0 to 8.0 was used. The dialysis membrane was pretreated according to the manufacturer's instructions. Test compounds and warfarin (control) were assayed at a final concentration of 2 μM. Aliquots of 50 μL loading plasma (matrix) containing test or control compound in triplicate were transferred to Sample Collection Plates, respectively. The samples were matched with opposite blank buffer to obtain a final volume of 100 μL with a volume ratio of plasma (matrix) to dialysis buffer (1:1, v-v) in each well immediately. The stop solution (methanol–acetonitrile (50:50, v-v) containing tolbutamide at 200 ng/mL, labetalol at 200 ng/mL, and metformin at 50 ng/mL) were added to these T0 samples of test compound and control compound. The plate was sealed and shaken at 800 rpm for 10 min. Then, these T0 samples were stored at 2 °C–8 °C pending further processing along with other post-dialysis samples. Aliquots of 150 μL of the loading plasma (matrix) containing test compound or control compound were transferred to the donor side of each dialysis well in triplicate, and 150 μL of the dialysis buffer (100 mM sodium phosphate and 150 mM NaCl, pH 7.4 ± 0.1) was loaded to the receiver side of the well. Then, the plate was rotated at approximately 100 rpm in a humidified incubator with 5% $CO_2$ at 37 °C ± 1 °C for 4 h. At the end of the dialysis, aliquots of 50 μL samples from the buffer side and plasma (matrix) side of the dialysis device were taken into new 96-well plates (Sample Collection Plates). An equal volume of the opposite blank matrix (buffer or plasma) in each sample was added to reach a final volume of 100 μL with a volume ratio of plasma (matrix) to dialysis buffer at 1:1 (v:v) in each well. Samples were further processed by protein precipitation. Concentrations of test compounds in the starting solution (before dialysis), the plasma side of the membrane, and the buffer side of the membrane were quantified by LC-MS/MS using a peak area ratio of analyte/internal standard. The fraction of unbound, bound, and recovery were calculated. Results are the average of three biological replicates.

## Kinetic solubility

This study was conducted at Wuxi (http://www.wuxi.com/). Stock solutions of test compounds (10 mM; 100% DMSO) and 490 μL of PBS buffer at pH = 7.4 were added in the lower chamber of Whatman's Mini-Uniprep vials. After vortex mixing for ≥2 min, the vials were shaken at 800 rpm for 24 h and finally centrifuged at 1520 g for 20 min. The concentration of the filtrate was quantified by HPLC (calibration curve from 1.5624 μM to 200 μM). Amiodarone hydrochloride, carbamazepine and chloramphenicol were used as controls.

## CACO-2 permeability

This study was conducted at Wuxi (http://www.wuxi.com/). Caco-2 cells (ATCC) were seeded onto polyethylene membranes in 96-well BD insert plates at $1 \times 10^5$ cells/cm², and the medium was changed every 4 days until the 21st to 28th day for confluent cell monolayer formation. Test compounds were diluted with transport buffer (HBSS with 10 mM HEPES at pH 7.4 ± 0.05) to a concentration of 2 μM (final DMSO concentration adjusted to <1%) and tested bi-directionally to investigate

P-glycoprotein-mediated efflux in duplicate. Controls: digoxin (10 μM, bi-directionally), nanodolol and metoprolol (2 μM, apical, *A*, to basolateral, *B*). The permeation was assessed over a 120-minute incubation at 37 °C ± 1 °C and 5% $CO_2$ at saturated humidity without shaking. Samples were mixed with acetonitrile containing internal standard and centrifuged at 1520 g for 10 min. Then, 100-μL aliquots of supernatant solutions were diluted with 100 μL of pure water. Concentrations of test and control compounds in starting solution, donor solution, and receiver solution were quantified by LC-MS/MS using a peak area ratio of analyte to internal standard. Permeation of lucifer yellow through the monolayer was measured to evaluate the cellular integrity. The results are the average of two biological replicates.

## Human ether-a-go-go related gene (hERG) blockade assay

This study was conducted at Wuxi (http://www.wuxi.com/), using CHO cells with stable expression of hERG potassium channels (Sophion Biosciences, Ballerup, Denmark). Cells were cultured (F12 medium, supplemented with 10% FBS, 1% Geneticin® selective antibiotic, 89 μg/mL hygromycin B) in a humidified and air-controlled (5% $CO_2$) incubator at 37 °C ± 2 °C. Test compounds were diluted at different concentrations (0.3 μM–30μM), with a final DMSO concentration in an extracellular solution of <0.3%. The external solution consisted of 2 mM $CaCl_2$, 1 mM $MgCl_2$, 4 mM KCl, 145 mM NaCl, 10 mM glucose and 10 mM HEPES at pH 7.3–7.4. The internal solution consisted of 5.37 mM $CaCl_2$, 1.75 mM $MgCl_2$, 120 mM KCl, 10 mM HEPES, 5 mM EGTA and 4 mM $Na_2ATP$ at pH 7.2–7.3. The assay was performed at room temperature using the whole-cell patch clamp techniques controlled with Patchmaster Pro software. For quality control, minimum seal membrane resistance was set at 500 MOhms and the minimum/maximum specific hERG current (pre-compound) was 400 pA/3000 pA. A 100-nM amount of cisapride was used as the positive control to evaluate the reliability of the test system. Results are the average of two biological replicates.

## PAMPA permeability

This assay was performed in triplicate as described previously[25,26].

## Cytotoxicity in THLE-2 cells and PBMCs cells

The cytotoxicity of compounds CM-444 and CM-1758 against the immortalized human liver THLE-2 cell line (ATCC CRL-2706) following 24-h and 72-h exposure and PBMCs following a 72-h exposure was performed in duplicate (pLC$_{50}$ difference <0.30 and 0.50 log units, for THLE-2 and PBMCs, respectively), following the protocol described previously[18,25,26].

## Human and mouse liver microsomal stability

This study was conducted at Wuxi (http://www.wuxi.com/). The data collected were analyzed to calculate a half-life ($t_{1/2}$, min) for test compounds at a final concentration of 1 μM. A 5-μL aliquot of stock solution of test compound (10 mM; 100% DMSO) was diluted in 495 μL of 1:1 methanol/water (final concentration of 100 μM, 99% MeOH). Then, 50 μL of this intermediate solution was diluted in 450 μL of 100 mM potassium phosphate buffer to a concentration of 10 μM (working solution, 9.9% MeOH). The β-nicotinamide adenine dinucleotide phosphate reduced form (NADPH) regenerating system also contained tetrasodium salt, NADPH•4Na (Sigma, Cat.# No.00616). Human liver microsomes were obtained from Corning (Cat.#452117), and C57 mouse liver microsomes from Xenotech (Cat.#M5000), at a final concentration of 0.5 mg protein/mL. A volume of 10 μL of working solution of test and control compounds (testosterone, diclofenac and propafenone) and 80 μL/well microsome solution were added to a 96-well plate and incubated for 10 min at 37 °C. The reaction was started by the addition of 10 μL of NADPH regenerating system and stopped by the addition of 300 μL of stop solution (acetonitrile at 4 °C, including 100 ng/mL tolbutamide and 100 ng/mL of labetalol as

internal standard) at different incubation times (0, 5, 10, 20, 30, and 60 min). Then, the samples were shaken for 10 min and centrifuged at 1699 $g$ for 20 min at 4 °C. An aliquot of supernatant (100 µL) was transferred from each well and mixed with 300 µL of ultra-pure water before submitting to LC-MS/MS analysis. Concentrations of test and control compounds were quantified by LC-MS/MS using a peak area ratio of analyte/internal standard, and the percent loss of the parent compound was calculated at each time point to determine the half-life ($t_{1/2}$, min). Experiments were performed in duplicate.

### Human and mouse liver cryopreserved hepatocytes
This study was conducted at Wuxi (http://www.wuxi.com/). The data collected were analyzed to calculate a half-life ($t_{1/2}$, min) for test compounds at a final concentration of 1 µM. Stock solutions of test compound (10 mM; 100% DMSO) and positive controls (7-ethoxycoumarin and 7-hydroxycoumarin; 30 mM; 100% DMSO) were diluted to 1 mM and 3 mM with DMSO and then diluted to 100 µM and 300 µM with acetonitrile (dosing solutions). Cryopreserved cells (from Human Donors, Bioreclamation IVT, Cat No. X008001 and from male C57 Mice, Bioreclamation IVT, Cat No. S00463) were thawed, isolated and suspended in Williams' Medium E, then diluted with pre-warmed Williams' Medium E to $0.5 \times 10^6$ cells/mL. Cell viability was tested by trypan blue exclusion after thawing. A volume of 2 µL of dosing solution and 198 µL of pre-warmed cell suspensions were added to a 96-well plate and incubated at 37 °C (5% $CO_2$) with constant shaking at 600 rpm. Following different incubation times (0, 15, 30, 60 and 90 min), 20 µL of each sample was transferred to a well containing 80 µL of ice-cold stop solution (acetonitrile containing 200 ng/mL tolbutamide and 200 ng/mL labetalol as internal standards). Plates were vortexed at 27 $g$ for 10 min and then centrifuged at 3220 × $g$ for 20 min at 4 °C. An aliquot of supernatant (75 µL) was transferred from each well and mixed with 75 µL of ultra-pure water before submitting to LC-MS/MS analysis. Concentrations of test and control compounds were quantified by LC-MS/MS using a peak area ratio of analyte/internal standard, and the percent loss of the parent compound was calculated at each time point to determine the half-life ($t_{1/2}$, min). Experiments were performed in duplicate.

### Plasma stability assay
This study was conducted at Wuxi (http://www.wuxi.com/). Fresh female C57 BL/6 mouse plasma (6 mL; Wuxi Apptec) was collected on the day of the experiment, and the pooled frozen human plasma (BioreclamationIVT, Cat No. HMPLEDTA2) was thawed in a water bath at 37 °C prior to the experiment. EDTA-K2 was used as an anticoagulant. Plasma was centrifuged at 1520 $g$ for 5 min, and any clots were removed. The pH was adjusted to 7.4 ± 0.1 if required. Half volumes of mouse and human plasma were pre-warmed in a water bath at 37 °C, and another half volume each of mouse and human plasma were stored at room temperature prior to the experiment. Intermediate solutions (1 mM) of tested compounds and positive control (propantheline) were prepared by diluting 5 µL of the stock solution (10 mM; 100% DMSO) with 45 µL DMSO (test compounds) or ultra-pure water (propantheline). Then, working solutions (100 µM) were prepared by diluting 10 µL of intermediate solutions with 90 µL of 45% MeOH/$H_2O$. Next, 98 µL of blank plasma was spiked with 2 µL of dosing solution (100 µM) to achieve a 2 µM final concentration in duplicate, and samples were incubated at 37 °C in a water bath. At each time point (0, 1, 2, 4, 6, and 24 h), the reactions were stopped by removing the plates from a water bath and by addition of 400 µL of ice-cold stop solution (200 ng/mL tolbutamide and 200 ng/mL labetalol in MeOH/acetonitrile (v:v = 1:1)). The samples were immediately vortexed on a plate shaker at 800 rpm for 10 min. Samples were centrifuged at 1699 $g$ for 10 min, and an aliquot of supernatant (100 µL) was transferred from each well and mixed with 200 µL ultra-pure water before submission for LC-MS/MS analysis (Acquity UPLC HSS). Concentrations of test compounds were quantified by LC-MS/MS using the peak area ratio of analyte/internal standard, and the percentage loss of the parent compound was calculated at each time point to determine the half-life. Experiments were performed in duplicate.

### CM-444 and CM-1758 toxicity assay: hematological and liver parameters
After treating Rag2$^{-/-}$γc$^{-/-}$ mice with i.p. 10 mg/kg of CM-444 and CM-1758 for 5 consecutive days followed by 2 rest days over 3 weeks, followed by a 7-day washout period. Hematological and liver parameters were measured.

Hematological parameters, including white blood cells, red blood cells, platelet count, hemoglobin and hematocrit, were measured on a Hemavet Hematology Analyzer (Drew Scientific). Liver enzymes (albumin, alkaline phosphatase, aspartate transaminase, alanine transaminase), cholesterol, bilirubin, urea and bile acid levels were analyzed from mice serum using a C311 Cobas Analyzer (Roche Diagnostics).

Mice livers were fixed in paraformaldehyde at 4% for 6–8 h, washed twice with saline solution and stored in 70% ethanol. Samples were included in paraffin, and 3-µm serial sections were cut, deparaffinated, and stained with hematoxylin–eosin.

### Pharmacokinetic study of CM-444 and CM-1758 in plasma samples
Compounds CM-444 and CM-1758 were measured in plasma using an Acquity UPLC system (Waters, Manchester, UK) coupled to an Xevo-TQ MS triple-quadrupole mass spectrometer with ESI source.

Compound solutions were prepared by dissolving the solid in dimethyl sulfoxide (DMSO) and this solution was made up to a final volume by the addition of a mixture of Tween 20 and 0.9% NaCl (1/1/8, v-v:v, DMSO/Tween 20/saline). For the pharmacokinetic experiments, a drug dosage of 10 mg/kg was administered as a single intraperitoneal injection to RAG-2 mice. Blood was collected at predetermined post-injection times (15 min, and 1, 2, 4, 8 and 24 h) into EDTA-containing tubes, and plasma was obtained via centrifugation (4 °C, 657 g, 5 min) and stored at −80 °C until analysis.

Chromatographic separation was performed by gradient elution at 0.6 mL/min using an Acquity UPLC BEH C18 column (50 × 2.1 mm, 1.7 µm; Waters). The mobile phase consisted of A: water with 0.1% formic acid, B: methanol with 0.1% formic acid. The autosampler temperature was set at 10 °C and the column temperature at 40 °C. For detection and quantification, the ESI operated in the positive mode was set up for multiple reaction monitoring. The collision gas used was ultra-pure argon at a flow rate of 0.15 mL/min.

Quantification was achieved by external calibration using matrix-matched standards. Concentrations were calculated using a weighted least-squares linear regression (W = 1/x). Calibration standards were prepared by adding the appropriate volume of diluted compound solution (made in a mixture of methanol and water, 50:50, v-v) to aliquots of 20 µL of blank plasma. The calibration standard and sample preparation were as follows: 3% formic acid in methanol was added to precipitate the proteins. The mixture was then vortex-agitated for 1 min and centrifuged at 16,522 $g$ for 10 min at 4 °C. A 5-µL aliquot of the supernatant solution was injected into the LC-MS/MS system for analysis. Frozen plasma samples were thawed at room temperature, vortex mixed thoroughly and subjected to the above-described extraction procedure.

The pharmacokinetic parameters were obtained by fitting the blood concentration-time data to a non-compartmental model with WinNonlin software (Pharsight, Mountain View, CA).

### In vivo experiments
All animal studies had previous approval from the Animal Care and Ethics Committee of the University of Navarra.

Mice were housed in a suitable temperature and humidity environment (25 °C, suitable humidity (typically 50%), 12 h dark/light cycle), and fed with sufficient water and food. All mice were euthanized via cervical dislocation.

For these studies, female mice were used due to higher engraftment rates as compared to male mice.

The human AML HL-60 and ML-2 cell lines were first pretreated in vitro daily with 270 and 260 nM of CM-444 or 300 and 210 nM of CM-1758, respectively, for 4 days. Prior to injection in mice, the cell differentiation induction by CM-444 and CM-1758 compounds was verified by measuring CD11b by flow cytometry, as described earlier. Then, $5 \times 10^6$ cells of cells, pretreated with CM-444, CM-175, or vehicle (80% saline solution, 10% DMSO and 10% Tween 20) and diluted in 100 µL of saline solution, were implanted subcutaneously into the flank of female BALB/cA Rag2$^{-/-}$γc$^{-/-}$ mice between 6 and 8 weeks of age ($n = 8$). Tumor size was analyzed every 2–3 days using the following method: V¼ D_d2/2, where D and d corresponded to the longest and shortest diameter, respectively. The maximal tumor size/burden permitted by the Animal Care and Ethics Committee of the University of Navarra is 2000 mm³. In some cases, this limit was reached on the last day of measurement and the mice were immediately euthanized.

The subcutaneous xenograft models of ML-2 and MV4-11 derived cells, $5 \times 10^6$ and $10 \times 10^6$ cells, respectively, diluted in 100 µL of saline solution were subcutaneously inoculated in the back left flank of female BALB/cA Rag2$^{-/-}$γc$^{-/-}$ mice. When tumors became palpable, the mice were randomized into three groups: vehicle, CM-444 and CM-1758, (eight animals/group). Treatment with 10 mg/kg of CM-444 or CM-1758 i.p. was started 4 days after injection when all mice presented subcutaneous tumors and was administered for 5 consecutive days followed by 2 resting days over 3 weeks for the ML-2 model or daily for 2 weeks for the MV4-11 model. The control group received only 80% saline solution, 10% DMSO or 10% Tween 20 (diluents of CM-444 and CM-1758). Tumor size was analyzed every 2–3 days using the following method: V¼ D_d2/2, where D and d corresponded to the longest and shortest diameter, respectively. The maximal tumor size/burden permitted by the Animal Care and Ethics Committee of the University of Navarra is 2000 mm³. This maximal tumor size/burden was not exceeded in any case. ML-2 and MV4-11 tumors were explanted for differentiation and global histone acetylation analysis. Histones were extracted, and a western blot of H3Ac was performed as described above. For the differentiation study, RNA was extracted, and q-PCR of *CD11b* was performed as described above (*CD11b* sequence primer can be found in Supplementary Table 10).

The human AML MV4-11 and MOLM-13 xenograft mice models were generated by intravenous (i.v.) injection of $10 \times 10^6$ (MV4-11) or $1 \times 10^4$ (MOLM-13) cells diluted in 100 µL of saline solution in the tail vein of a 6-8-week-old female BALB/cA Rag2$^{-/-}$γc$^{-/-}$ mice ($n = 10$). Treatment started 7 or 1 day after MV4-11 or MOLM-13 cells injection, respectively. MV4-11 and MOLM-13 mice were treated with i.p. 10 mg/kg of CM-444 and CM-1758 during 5 consecutive days followed by 2 resting days for 7 and 2 weeks, respectively. The control group received only 80% saline solution, 10% DMSO and 10% Tween 20 (diluents of CM-444 and CM-1758). The mice weight was controlled and remained stable during the treatment. Statistical results were calculated using the statistical software Medcalc. CD11b levels in blood samples from an intravenous MV4-11 mice model were measured by flow cytometry, as described above.

### Acetylome and proteome analysis
**Sample preparation.** Cellular pellets were lysed for 15 min at 95 °C in 4% SDS, 100 mM Hepes/NaOH pH 8.0, containing 5 mM sodium butyrate and 5 mM Nicotinamide. After cooling down, lysates were incubated at 25 °C with 10 units of DNAse (Benzonase, Merk) and sonicated for 15 min in a Bioruptor for DNA shearing. Protein concentration was determined using microBCA, using BSA as standard. Samples were digested using on-bead protein aggregation capture (PAC) with ReSyn Biosciences MagReSyn® Hydroxyl microparticles (ratio Protein/Beads 1:5) in an automated King Fisher instrument (Thermo). Proteins were digested for 16 h at 37 C with LysC/Trypsin mix at a protein: enzyme ratio of 1:100 in 50 mM TEAB pH 8.0 (Trypsin Sequence grade, Sigma; LysC, Wako). The resulting peptides were speed-vac dried and dissolved in 100 µL of 200 mM HEPES pH 8.5.

Peptides (500 µg) were labeled with Thermo Scientific TMTpro 18plex™ Isobaric reagents for 1 h at 25 C. Reaction was quenched by adding 5% hydroxylamine for 15 min. Samples were pooled at a 1:1 ratio based on the total peptide amount, determined by comparing overall signal intensities on a regular LC-MS/MS run. The final mixture was desalted using a 500 mg Sep-Pak C18t cartridge (Waters) and dried prior to high pH reverse phase HPLC pre-fractionation.

**High pH reverse phase chromatography.** Labeled peptides were fractionated offline through a high pH reverse phase chromatography using an Ultimate 3000 HPLC system equipped with a sample collector. Briefly, peptides were dissolved in 1200 µL of Buffer A (10 mM NH4OH) and loaded onto a XBridge BEH130 C18 column (3.5 µm, 250 mm length and 3.5 mm ID, Waters). Buffer B was composed of 10 mM NH4OH in 90% CH3CN. The following gradient was used at a flow rate of 500 µL/min: 0-50 min 0-25% B, 50-56 min 25-60% B, 56-57 min 60-90% B. One-minute fractions (500 µL) from minute 15 to 65 were collected and neutralized with 10% formic acid. Based on the UV absorbance at 280 nm, 95% of the volume of 40 fractions was pooled in 5 fractions for Acetyl-Lysine enrichment. The 5% of these fractions was kept for proteome analysis. Pooled fractions were snap-frozen and lyophilized.

**Immuno-Purification of Lysine Acetylated Peptides.** Lyophilized peptides (1.5 mg) from each pool were immuno-affinity purified with the PTMScan HS Acetyl-Lysine Motif (Ac−K) kit (Cell Signaling Technology). For each pool, three consecutive immuno-purifications (IPs) were performed using the manufacturer's protocol in an automated King Fisher instrument (Thermo). Elution of purified peptides was performed in two consecutive steps of 50 µl of 0.15% TFA. Samples were desalted with C18 stage tips and eluates were vacuum dried and resuspended in 0.1% formic acid for subsequent analysis by LC−MS/MS.

**Mass spectrometry.** LC-MS/MS was done by coupling an UltiMate 3000 RSLCnano LC system to an Orbitrap Exploris 480 mass spectrometer (Thermo Fisher Scientific). Samples were loaded into a trap column (Acclaim™ PepMap™ 100 C18 LC Columns 5 µm, 20 mm length) for 3 min at a flow rate of 10 µL/min in 0.1% FA. Peptides were then transferred to an EASY-Spray PepMap RSLC C18 column (Thermo) (2 µm, 75 µm x 50 cm) operated at 45 °C and separated using a 90 min effective gradient (buffer A: 0.1% FA; buffer B: 100% ACN, 0.1% FA) at a flow rate of 250 nL/min. The gradient used was, from 4% to 6% of buffer B the first 2 min, from 6% to 33% B the next 58 min, plus 10 additional minutes at 98% B.

The mass spectrometer was operated in a data-dependent mode, with an automatic switch between MS and MS/MS scans using a top 15 method. (Intensity threshold ≥ 1e4, dynamic exclusion of 20 sec and excluding charges unassigned, +1 and ≥ +7). MS spectra were acquired from 350 to 1500 m/z with a resolution of 60,000 FMHW (200 m/z). Ion peptides were isolated using a 0.7 Th window and fragmented using higher-energy collisional dissociation (HCD) with a normalized collision energy NCE of 36. MS/MS spectra were acquired with a fixed first mass of 120 m/z and a resolution of 45,000 FMHW (200 m/z). The ion target values were 3e6 for MS (maximum IT 25 ms) and 1e4 for MS/MS (maximum IT, auto).

**Data analysis**. For acetyl-lysine enriched TMT-labeled samples, raw files were processed with MSFragger (Yu, F et al, Nat. Comm. 2020) against a human protein database (UniProtKB/TrEMBL, 20,593 sequences, UP000005640_9606, May 2023) supplemented with contaminants. Carbamidomethylation of cysteines, N-term and lysine TMTpro-label were set as fixed modifications, whereas oxidation of methionine, protein N-term acetylation, N/Q de-amidation and acetylation of lysines were set as variable modifications. Minimal peptide length was set to 7 amino acids and a maximum of four tryptic missed-cleavages were allowed. Results were filtered at 0.01 FDR (peptide and protein level).

For total protein analysis, raw files were processed with MaxQuant (v 2.4.9.0) using the standard settings against the same protein database.

The acetyl-K sites and protein intensities files were loaded in Prostar (v1.32.3) (Wieczorek et al, Bioinformatics 2017) using the intensity values for further statistical analysis. Briefly, proteins/sites with less than eighteen valid values were filtered out. Then, a global normalization of log2-transformed intensities across samples was performed using the LOESS function. Differential analysis was done using the empirical Bayes statistics Limma. Proteins with a p-value < 0.01 and a log2 ratio >0.3 or <-0.3 were defined as regulated (Martinez-Val et al, Proteome Res, 2016). The FDR was estimated to be below 2% by Benjamini-Hochberg.

Functional analysis, hierarchical clustering, PCA, motif annotation and Fisher exact tests were performed using the Perseus software platform[55].

## Analysis of the acetylation motifs

Acetyl-lysine motif analysis was performed and presented using the iceLogo motif builder[56] using a *p*-value of 0.01.

We constructed the reference multiple sequence alignment set by using the 15 amino acids surrounding the K-acetylation site of the quantified K-acetylation sites from the experiment. For the positive set, we used the same 15-amino-acid surrounding sequence window corresponding to the histones. The standard deviation (σ) is calculated using the sample size (N) and the frequency (f%) of an amino acid in the reference set, with the formula:

$$\sigma = \sqrt{((f\%)/N)}$$

For every position, the amino acid frequencies in the positive set will be compared with the frequencies in the reference set. An amino acid will be regulated if the Z-score is not a part of the confidence interval. For the calculation of the confidence interval the Wichura algorithm was used (Supplementary Table 11) The Z-score is calculated with the formula: Z-score = X − μ σ. The formula will calculate how many times the frequency (X) is deviated from the mean (μ, the frequency of a specific amino acid on a specific position in the reference set) in terms of the standard deviation (σ).

For visualization the percentage difference, using the difference in frequency for an amino acid in the experimental set and the reference set as a measure of the height of a letter in the amino acid stack, was used[57].

## STRING database

The protein–protein interactions network was built from the online STRING database (https://string-db.org/, version 11.5). The interactions (the default threshold > 0.4 in the STRING database) were selected to create the PPI network and further clustered by the K-means method.

## CUT&RUN

CUT&RUN was performed with the kit CUTANA ChIC/CUT&RUN (Epicypher, 14-1048) following the manufacturer's instructions. Briefly, 0.5 million of HL-60 cells were harvested and resuspended in 100ul wash buffer [20 mM HEPES pH 7.5, 150 mM NaCl, 0.5 mM Spermidine, supplemented with Protease Inhibitor EDTA-Free tablet (Roche 11836170001)]. Activated Concanavalin A was incubated with cells at room temperature for 10 min to let the cells bind to the beads. For each target protein factor, 0.5 μg of BRD4 CUTANA CUT&RUN antibody (Cat No 13-2003, Epicypher), Acetyl-Histone H3 (Lys 9) (C5B11) rabbit monoclonal antibody diluted 1:50 (Cat No 9649 T, Cell Signaling) or Acetyl-Histone H3 (Lys 27) (D5E4) XP rabbit monoclonal antibody diluted 1:100 (Cat No 8173T, Cell Signaling) was added to each sample and incubated in the antibody buffer (wash buffer +0.01% Digitonin and 2 mM EDTA) overnight at 4 °C. Isotype control IgG was used as a negative control. The beads were then washed twice with digitonin buffer [wash buffer +0.01% Digitonin] and 2.5 μL pAG-MNase was added to each sample. After ten minutes of incubation at room temperature, excessive pAG-MNase was washed out by a two-time digitonin buffer wash. Then targeted chromatin was digested and released from cells by 2 h of incubation with the presence of 2 mM CaCl2 at 4 °C. After incubation with stop buffer [340 mM NaCl, 20 mM EDTA, 4 mM EGTA, 50 μg/mL RNase A, 50 μg/mL Glycogen] for 10 min at 37 °C the supernatant containing CUT&RUN-enriched DNA was purified using the CUTANA DNA Purification kit (Epicypher). Finally, Illumina sequencing libraries were prepared from 6 ng of purified CUT&RUN DNA using NEBNext UltraTM II DNA Library Prep Kit for Illumina, following the manufacturer's instructions. For BRD4, some modifications were added following the protocol of Nan Liu (Harvard University; https://doi.org/10.17504/protocols.io.wvgfe3w). Libraries were sequenced using NEXTseq500 (Illumina).

The Cut&Run samples were analyzed using the CUT&RUN tools pipeline with default settings and no filtering for 120 bp fragments[58]. Peak calling was performed on deduplicated reads with macs2 version 2.2.6 using the respective IgG samples as control, narrow peak calling and paired-end settings, and a threshold of q = 0.01.

To analyze the chromatin state distribution in the three conditions (Control, CM-444 treatment and Panobinostat treatment) we overlapped the called regions of each sample of the study with the chromatin states of each AML sample. Specifically, 12 chromatin states, based on the combination of 6 histone marks, for 38 AML samples were used[59]. The vast majority of the regions could be entirely annotated to a single chromatin state while some could be entirely annotated to a single chromatin state after the replacement of consecutive chromatin states with similar characteristics with one of them (i.e combination of Transcription Transition / Transcription Elongation / Weak Transcription to Transcription Transition or combination of Heterochromatin; Represesed/Heterochromatin; Low Signal to Heterochromatin; Repressed etc). The regions that couldn't be entirely annotated to a single chromatin state or simplified chromatin state were removed. Using this approach, we achieved to annotate between 82.2%-92.2% of the regions called in the nine samples. We then calculated the percentage of each chromatin state taking into account the width of the regions. As background, we used all the regions of each AML sample and we calculated the percentage of the chromatin states similarly. Finally, we calculated the fold change of each chromatin state as a percentage in the sample to a percentage in the background AML sample.

## Statistical analysis

Comparisons were made with the Mann–Whitney *U*-test (for unpaired samples without a normal distribution) and Student's *t*-test (for paired samples showing Gaussian distribution). Survival analyses (recurrence-free) according to various variables were performed using the Kaplan–Meier method, and differences between the different groups of patients or mice were tested with the log-rank test. GraphPad Prism 6.0 was used. *P* < 0.05 was considered to be indicative of statistically significant differences.

**Reporting summary**

Further information on research design is available in the Nature Portfolio Reporting Summary linked to this article.

## Data availability

The raw RNA-seq and CUT&RUN data generated in this study have been deposited in the NCBI Gene Expression Omnibus (GEO) repository under the accession numbers GSE219230 and GSE268008, respectively. The raw mass spectrometry proteomics data generated in this study have been deposited in the ProteomeXchange Consortium via the PRIDE partner repository with the data set identifier PXD050623. The remaining data are available within the article, Supplementary Information, or Source data file. Source data are provided in this paper.

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

## Acknowledgements

We thank the patients and their families. We particularly thank the Biobank of the University of Navarra for its collaboration. We thank Carmen Sanmartin from the School of Pharmacy and Nutrition of the University of Navarra for melting-point determinations and Angel Irigoyen Barrio and Dr. Ana Romo Hualde from the University of Navarra for HMRS determination. We also thank Maite García Férnandez de Barrena from CIMA University of Navarra for liver parameters determination and hematoxylin-eosin staining. This research was funded by grants from the Instituto de Salud Carlos III (ISCIII) and co-financed by FEDER: PI17/00701 (FP), PI19/01352 (XA), PI20/01306 (ESJ-E), PI22/00947 (XA), PI23/00488 (ESJ-E), CIBERONC CB16/12/00489 (FP) and CB16/12/00369 (BP). Gobierno de Navarra: Departament of Health (44/2021 (XA) and GN2023/11 (ESJ-E) (also co-financed by FEDER) and Department of Economic Development AGATA (0011-1411-2020-000011 (FP) and 0011-1411-2020-000010 (FP)), NanoRC (0011-1411-2022-000068 (FP and XA)) and MEET-AML (0011-2750-2019-000001) (FP), MEET-AML is cofunded by ERA-NET program. AECC Innova (INNOV211822SANJ) (ESJ-E). The study was also supported by Cancer Research UK [C355/A26819] and FC AECC and AIRC under the Accelerator Award Programme, the Multiple Myeloma Research Foundation Networks of Excellence 2017 Immunotherapy Program Grant Award, the International Myeloma Foundation (Brian van Novis) (XA), the Paula and Rodger Riney Foundation (FP), the Fundación Fuentes Dutor, La Caixa Foundation (GR-NET NORMAL-HIT HR20-00871) (JO).

## Author contributions

Conception and design: E.S.J.-E., N.G.-C., O.R., X.A., F.P. and J.O. Development of methodology: E.S.J.-E., N.G.-C., O.R., F.G., J.M., A.P.-L., X.A. Acquisition of data and assistance with experiments: L.G., E.M., N.G.-E., F.G., S.C., E.S., A.V.-Z., P.S.M.-U., N.B., M.J.L., M.J.C., J.M., D.A., B.P., L.V.-V., N.M.-C., L.E.T.-A., A.P.-R., S.H., M.S. and A.A.-P. Analysis and interpretation of data: E.S.J.-E., N.G.-C., O.R., J.I.M.-S., J.M., M.I., A.P.-L., J.O., X.A. and F.P. Writing, review, and/or revision of the manuscript: E.S.J.-E., N.G.-C., O.R., A.P.-L., J.I.M.-S., J.O., X.A. and F.P. Study supervision: E.S.J.-E., X.A., F.P. and J.O.

## Competing interests

The authors declare no competing interests.

## Additional information

[1]Hemato-Oncology Program, Center for Applied Medical Research (CIMA), Universidad de Navarra, IDISNA, CCUN, Avenida Pío XII 55, 31008 Pamplona, Spain. [2]Centro de Investigación Biomédica en Red Cáncer (CIBERONC), 28029 Madrid, Spain. [3]Small-Molecule Discovery Platform, Molecular Therapeutics Program, Center for Applied Medical Research (CIMA), Universidad de Navarra, Avenida Pío XII 55, 31008 Pamplona, Spain. [4]ProteoRed-ISCIII, Unidad de Proteómica, Centro Nacional de Investigaciones Oncológicas (CNIO), Melchor Fernández Almagro 3, 28029 Madrid, Spain. [5]Institut d'Investigacions Biomèdiques August Pi I Sunyer (IDIBAPS), Casanova 143, 08036 Barcelona, Spain. [6]TECNUN, Universidad de Navarra, Manuel de Lardizábal 13, 20018 San Sebastián, Spain. [7]Departmento de Hematología, Clínica Universidad de Navarra, and CCUN, Universidad de Navarra, Avenida Pío XII 36, 31008 Pamplona, Spain. [8]Biomedical Engineering Program, Center for Applied Medical Research (CIMA), Universidad de Navarra, IDISNA, Avenida Pío XII 55, 31008 Pamplona, Spain. [9]Department of Enzymology, Charles Tanford Protein Center, Institute of Biochemistry and Biotechnology, Martin-Luther-University Halle-Wittenberg, 06120 Halle, Germany. [10]Department of Medicinal Chemistry, Institute of Pharmacy, Martin-Luther-University Halle-Wittenberg, 06120 Halle, Germany. [11]CIMA LAB Diagnostics, Universidad de Navarra, Avenida Pío XII 55, 31008 Pamplona, Spain. [12]Biocruces Bizkaia Health Research Institute, Cruces Plaza, 48903 Barakaldo, Spain. [13]Ikerbasque, Basque Foundation for Science, Plaza Euskadi 5, 48009 Bilbao, Spain. [14]Departamento de Fundamentos Clínicos, Universitat de Barcelona, Casanova 143, 08036 Barcelona, Spain. [15]Institució Catalana de Recerca i Estudis Avançats (ICREA), Passeig de Lluís Companys 23, 08010 Barcelona, Spain. [16]These authors contributed equally: Edurne San José-Enériz, Naroa Gimenez-Camino. ✉e-mail: julenoyarzabal@external.unav.es; xaguirre@unav.es; fprosper@unav.es

