## [Peer Review File · Nature Communications]

Epigenetic based differentiation therapy for Acute Myeloid LeukemiaReviewers' Comments:

Reviewer #1:

Remarks to the Author:

In this well written manuscript, José-Enériz et al., present two novel structurally similar (and similarly derived) compounds CM-444 and CM-1758 that were identified from multiple chemical series the authors had generated for epigenetic inhibitors.

Using established assays, they undertake with the compounds

- 1) biochemical characterisation of inhibitory activity of epigenetic enzymes, that demonstrate particular activity of CM-444 and CM-1758 against deacetylase enzymes
- 2) assays to assess differentiation of AML cell lines and primary samples by monitoring expression of CD11b using "low dose" 25% GI50 of tested compounds
- 3) characterisation of gene expression changes with AML differentiation
- 4) establish ADME and PK parameters in mouse and LC50 values on hepatic THLE-2 and human PBMCs
- 5) in vivo transplant assays into immunodeficient mice demonstrating CM-444 and CM-1758 differentiated cell lines demonstrated growth disadvantage
- 6) proteomic analysis after acetyl-K enrichment identifying several acetylated protein classes including DNA repair, bromodomain, Moz/FORF, mediator complexes

The authors propose a model whereby DAC inhibition by CM-444 and CM-1758, promotes acetylation of non-histone proteins including BRD4 and BRD1. This epigenetic modulation of transcription factor expression including down regulation of c-MYC and up regulation of myeloid associated transcription factors promotes myeloid differentiation of AML cell lines.

Overall, this is a thorough and extensive program of work.

Major comments

CM-444 and CM-1758 exhibit a very broad spectrum of activity as the authors recognise. Elucidating a "definitive pathway" for the mechanism of action CM-444 and CM-1758 is therefore challenging but important. This is especially relevant as the authors propose a model promoting BRD1 and BRD4 and "non histone" molecule acetylation as an epigenetic mechanism of action.

The genetic (transcriptional) evidence for the primary or secondary effects of CM-444 and CM-1758 action on AML cell lines is reasonably compelling, however, as the authors have developed and proposed these compounds as epigenetic modifiers, there is a relative paucity of examination of epigenetic mechanisms of action.

While global acetylation has been assessed, validation of specific targets is relatively lacking. My major comments predominantly relate to this mechanistic aspect of their study.

1. How does acetylation of BRD1 and BRD4 modulate their function? Are they inhibited or activated by CM-444 and CM-1758?

For instance, in Figure 5D and S8, the authors demonstrate BRD inhibitors GSK525762 (molibresib) and JQ1 appear to mitigate the action of CM-444 and CM-1758 when used in combination therapy across multiple gene loci other than for c-MYC. The authors therefore propose CM-444 and CM-1758 promotion of BRD acetylation activate their activity.

However, can the authors then explain why there is concordance of activity of CM-444 and CM-1758 and molibresib and JQ1 for MYC gene expression?

2. To better delineate the effect of CM-444 and CM-1758 on BRD activity, the authors should examine whether the differentiation phenotype manifestly related to disruption of acetylation of non-histone proteins eg. BRDs versus for eg. acetylation of histone targets eg. H3K9 H3K122 H3K14 (see point 2 & This can be undertaken via

- a. Assessment of chromatin accessibility (eg. ATACseq), and distribution of chromatin marks eg. H3K9Ac, and BRD4 associated H3K122 (Devhaiah et al., Nat Struct Mol Biol 2016) and BRD1 associated H3K14 (Mishima et al., Blood 2011) with treatment by CM-444 and CM-1758 on AML cell line(s), in addition to distribution of BRD4 and BRD1 themselves by ChIP with and without treatment.
- b. Relating these epigenetic changes to MYC and other myeloid transcription factors as the authors have examined in their transcriptional analysis

The findings from these experiments should support/or not support the hypotheses, including whether the primary activity these agents is via activation of the BRD proteins as the authors propose.

Minor comment

1. If the CM-444 and CM-1758 compounds act via a predominant BRD axis as the authors propose, the authors should discuss whether the broad spectrum of activity CM-444 and CM-1758 is relevant. Similarly, if BRD activation is indeed a mechanism of action, they should contrast this activity to BRD inhibitory compounds that have already been developed and the implications as a therapeutic strategy (eg. novelty).
2. I would suggest the authors enunciate the central tenet in their strategic approach more in their manuscript, including in the discussion ie. adopting a strategy for "differentiation potential" rather than proliferation and cell death endpoints. This represents the novel aspect of their drug discovery strategy that has resulted in identification of compounds that minimise tissue toxicity.
3. The authors should discuss the future aims regarding their compounds, including the scope for therapeutic development. Would their broad spectrum of activity preclude further therapeutic development in human or is this mitigated by preclinical evidence for a large therapeutic window of activity? Are these pilot compounds useful for biochemical and biological characterisation in preclinical studies? Can and will these compounds be modified for specificity of targeting against DAC protein groups?
4. Please check the use of the word "beside" on page 5.
5. Better representative images (Figure 2H and Figure S2F) should be provided to support the author claims (eg. oil immersion lens) and better white balance, as well as provide an image for ML-2 treated with CM-1758. While the image of ML-2 treated with CM-444 may support the contention for morphological differentiation, I remain to be convinced for both the HL-60 treated panels. (These findings however, are not materially significant to the authors' ultimate conclusions).
6. I am unable to find the Figures panel S2F the authors refer to in the manuscript (Page 5)

Reviewer #2:

Remarks to the Author:

Review

The manuscript presented by Eurne San José-Enériz et al. describes two novel HDAC inhibitors with potential to differentiate AML cell lines, which might be applicable as a therapeutic regimen for cancer therapy. The inhibitors induce acetylation of histone and non-histone proteins and inhibit AML tumor growth in vivo in mouse models. The authors suggest BRD1 and BRD4, two acetyl-lysine reader proteins, as the main targets of the compound to induce AML differentiation.

In general, the results presented are of interest for the field, however additional evidence is needed as to the mechanism of action of the drugs. Some of the presented data lack detail and the used methodology for histone-acetylomics appears unsuitable.

Specific comments:

- 1) CM-444 is a bispecific inhibitor of HDACs and DNMTs. The authors claim, that no changes in DNA methylation was detectable after incubation of HL-60 cells for 48 hours, due to low doses used (Fig 1E). In order to detect DNA methylation changes, cells need to proliferate for several rounds, since DNA demethylation will occur only during replication as a passive effect of DNMT inhibition. Thus, the experiments should be repeated for longer time-points. The illustration of DNA methylation levels in color gradients is hard to assess and should be shown as percent methylation. A positive control such as decitabine treatment should be added. The quality of histone western blots in Fig. 1D lower panel and S1A should be improved.
- 2) The authors use Cd11b as a proxy for differentiation into myeloid lineage. It would be interesting to add additional surface markers to define specific subsets of myeloid cells.
- 3) Although both compounds induced high levels of differentiation in HL-60 and ML-2 cells, CM-444 also induced significant apoptosis in MOLM-13 and MV4-11 cells. The authors should comment on this.
- 4) The RNA-seq data should be described in more detail, eg. adding heatmaps and PCA blots to see overall changes following the treatments. How many genes were deregulated upon treatment, how many up- or down-regulated?
- 5) The in vivo data show reduced engraftment of pretreated cells. Fig.3A and S4A do not show error bars. It is unclear how many replicates were used and how significance was calculated? The same is true for Fig.3B and S4B, where replicates are not visible in the "Tumor volume" graphs.
- 6) For Fig.S4D and S4E the authors should provide the original histone blots.
- 7) In their proteomic analysis, the authors reveal 29 proteins that were commonly deregulated upon treatment with CM-444 and CM-1758 or with the reference HDACi panobinostat and vorinostat. The authors should show these proteins and comment on their function and relevance. Did the authors observe overlap also between RNA-seq and proteome data in terms of up and down regulated genes/proteins?
- 8) Fig. 4C shows the number of deregulated acetyl-K sites. Based on the numbers, the percentage of up-regulated sites for CM-444 should be 49% and not 79% as stated by the authors.
- 9) Fig.4E shows the overlap of regulated sites between the 4 treatments. More detail on the overlapping and unique sites/affected proteins should be shown and discussed.
- 10) Commonly, histone acetylomics is performed via derivatization, since trypsin digest of histone proteins would yield very small fragments that are difficult to retain on HPLC columns and used for MS. Could the authors please comment on their methodology? The authors should also provide a list of mapped histone peptides including the acetylation sites.
- 11) The statement: "Interestingly, we observed that most of the Acetyl-K sites regulated after treatment with each of the four HDACi belong to non-histone proteins (>90%) instead of histone proteins" is unclear. Is this assumption based on the number of sites detected? Presumably histone proteins represent only a small fraction of overall proteins, thus this conclusion is not valid.
- 12) Acetylomics of histones showed less separation in the PCA plot provided in Fig.4I and the authors argue "...that CM-444 or CM-1758 modulate specifically non-histone protein acetylation patterns in AML cells, suggesting that this differential effect could be key to discover the molecular mechanism underlying the increased ability to induce myeloid differentiation shown by CM-444 and CM-1758". This conclusion is somewhat misleading, since presumably, all inhibitors induce hyperacetylation of histones to a similar level but show differences in non-protein acetylation. Please rephrase.
- 13) Focusing on non-histone proteins in their further analysis, the authors identify 87 proteins that are specifically regulated following treatment with CM-444 and CM-1758. They suggest that the BRD proteins might be essential for the differentiation of AML cells following treatment. Their conclusion is based on using BRD inhibitors in combination with their compounds, which represses differentiation. Based on their illustration, BRD proteins are also targeted by other HDACi. It is unclear whether their compounds target different lysines. Again, the authors need to show the specific sites.
- 14) In order to confirm the essential function of BRD proteins in the differentiation process, the authors should perform further mechanistical studies including site directed mutagenesis. What is the function of BRD acetylation? Changes in their binding affinity to acetyl-lysine? Changes in overall stability or localization? Are there differences in binding on the respective promoters/enhancers of genes induced upon differentiation?

- 15) Did the authors detect changes in MYC acetylation after treatment?
16) HDAC inhibitors induce DNA damage. Was this observed also for their compounds? The authors should analyze this e.g by gH2AX immunofluorescence staining.

Minor comments:

Fig. 1A: some of the circles are cut off.

Page 8, last paragraph, typo: "Kmeans clustering split the network of the 85 specific Acetyl-K regulated proteins ...". It should be 87.

Reviewer #3:

Remarks to the Author:

The authors propose two molecules for a "Novel epigenetic based differentiation therapy for Acute Myeloid Leukemia". These two molecules feature the same warhead as Quisinostat.

Overall the manuscript gathers a wide-range of assays from docking to in-vivo evaluation. The authors describe these molecules as pan-HDAC inhibitors, which in contrast to the approved ones induce differentiation (CD11B marker) without excessive apoptosis (annexin-V). Whereas the pre-treatment of cells before injection in the murine model is convincing, the efficacy of the molecules for AML without pre-treating the cells is not as obvious. Interestingly, BRD inhibitors prevented the differentiation induced by the two inhibitors.

A large part of the manuscript is devoted to acetyloomics to try and understand the molecular cause of the observed differences.

Whereas the approach and the molecules are interesting, I deem this manuscript unsuitable for publication for the following two major reasons:

1) The acetyloomics study is conducted with a single replicate of cellular treatment per drug (two technical replicates of the IP of the same pellet and 2 MS injections of these IPs) rendering all the interpretation dubious.

2) The lack of consideration for medicinal chemistry is disconcerting. Whereas the two proposed molecules are obvious Quisinostat analogues, this molecule only features in one assay. It should be a control in most, with a fair determination of its optimal concentration. The 2 molecules are very close analogues, CM-444 being initially presented as a DNMTs/HDACs inhibitor while CM-1758 is presented as a HDACs inhibitor (fig1A). Yet no SAR discussion about this difference is to be found. The panHDAC nature of these molecules is also to be questioned, where the assays stem from Eurofins. Recent papers indeed cast doubt on the usually used enzymatic assays for HDACs and this would at least need commenting. Because of the similarity to Quisinostat, the assays of the two new molecules should be complemented by the same assay for Quisinostat and compared to recent publications that outline major issues notably for HDAC11 and class IIa HDACs. E.g:

<https://doi.org/10.1021/acsomega.9b02808>

<https://doi.org/10.1038/s41589-022-01015-5>

<https://doi.org/10.3390/ijms24054720>

Response to the Reviewers' comments

We would like to thank the reviewers for their thorough revision, constructive criticisms and insightful suggestions regarding our manuscript. We believe that this revised version of our study has markedly improved following the reviewers' advice.

Below you can find the detailed answers to all the issues raised by the reviewers.

Reviewer #1 (Remarks to the Author):

In this well written manuscript, José-Enériz et al., present two novel structurally similar (and similarly derived) compounds CM-444 and CM-1758 that were identified from multiple chemical series the authors had generated for epigenetic inhibitors.

Using established assays, they undertake with the compounds

- 1) biochemical characterisation of inhibitory activity of epigenetic enzymes, that demonstrate particular activity of CM-444 and CM-1758 against deacetylase enzymes
- 2) assays to assess differentiation of AML cell lines and primary samples by monitoring expression of CD11b using "low dose" 25% GI50 of tested compounds
- 3) characterisation of gene expression changes with AML differentiation
- 4) establish ADME and PK parameters in mouse and LC50 values on hepatic THLE-2 and human PBMCs
- 5) in vivo transplant assays into immunodeficient mice demonstrating CM-444 and CM-1758 differentiated cell lines demonstrated growth disadvantage
- 6) proteomic analysis after acetyl-K enrichment identifying several acetylated protein classes including DNA repair, bromodomain, Moz/FORF, mediator complexes

The authors propose a model whereby DAC inhibition by CM-444 and CM-1758, promotes acetylation of non-histone proteins including BRD4 and BRD1. This epigenetic modulation of transcription factor expression including down regulation of c-MYC and up regulation of myeloid associated transcription factors promotes myeloid differentiation of AML cell lines.

Overall, this is a thorough and extensive program of work.

Major comments

CM-444 and CM-1758 exhibit a very broad spectrum of activity as the authors recognise. Elucidating a "definitive pathway" for the mechanism of action CM-444 and CM-1758 is therefore challenging but important. This is especially relevant as the authors propose a model promoting BRD1 and BRD4 and "non histone" molecule acetylation as an epigenetic mechanism of action.

The genetic (transcriptional) evidence for the primary or secondary effects of CM-444 and CM-1758 action on AML cell lines is reasonably compelling, however, as the authors have developed and proposed these compounds as epigenetic modifiers, there is a relative paucity of examination of epigenetic mechanisms of action.

While global acetylation has been assessed, validation of specific targets is relatively lacking.

My major comments predominantly relate to this mechanistic aspect of their study.

1. How does acetylation of BRD1 and BRD4 modulate their function? Are they inhibited or activated by CM-444 and CM-1758?

For instance, in Figure 5D and S8, the authors demonstrate BRD inhibitors GSK525762 (molibresib) and JQ1 appear to mitigate the action of CM-444 and CM-1758 when used in combination therapy across multiple gene loci other than for c-MYC. The authors therefore propose CM-444 and CM-1758 promotion of BRD acetylation activate their activity.

However, can the authors then explain why there is concordance of activity of CM-444 and CM-1758 and molibresib and JQ1 for MYC gene expression?

These are indeed interesting issues raised by the reviewer. To address the mechanisms of action of CM-444 and CM-1758 we performed CUT&RUN experiments against BRD4 after treatment for 12h with CM-444 in HL-60 cells. Globally, we detected 3451 peaks in HL-60 cells (control) that were located predominantly in promoter regions (Reviewer Figure 1A). After CM-444 treatment, the number of peaks increased to 19570 that were mostly located in intronic and distal intergenic regions instead of promoters, also associated with active chromatin regions as demonstrated by the increase in H3K27Ac mark (Reviewer Figure 1B,C). Moreover, we also observed that CM-444 treatment drove BRD4 to genes related to myeloid differentiation such as *CB11b*, *ABII*, *GATA2* and *GFII* (Reviewer Figure 1C). These results suggest that the acetylation induced by our compounds induced a shift of BRD4 towards regulatory regions of genes involved in differentiation. This is consistent with the effect of BET inhibitors Molibresib and JQ1 in AML cell lines, where we observed that inhibition of BRD4 abrogates the differentiation of AML cells induced by CM-444 and CM-1758 (Reviewer Figure 1D).

These results have been included in the revised version of the manuscript (Results section, page 10) and the Reviewer Figure 1 has been included as Figure 6.

Results section, page 10: *“To delve into the role that BRDs could play in the differentiation process induced by our compounds but not by other DACi, we performed CUT&RUN experiments with BRD4 and the specific histone marks H3K9Ac and H3K27Ac in HL-60 after treatment with CM-444 or Panobinostat for 12 hours. The numbers of peaks of the different marks are shown in Figure 6A. Interestingly, CM-444 induced changes in the location of peaks associated with BRD4, H3K27Ac and H3K9Ac. While BRD4, H3K27Ac and H3K9Ac peaks were predominantly located in promoters in untreated cells, after treatment with CM-444 the majority of the peaks were located in intronic and distal intergenic regions (Figure 6B). Moreover, CM-444 treatment drove BRD4, H3K9Ac and H3K27Ac to genes related to myeloid differentiation and we observed that BRD4 peaks after CM-444 treatment were also associated with active chromatin regions as demonstrated by the increase in H3K27Ac mark (Figure 6C). Regarding the global changes carried out by Panobinostat, it should be highlighted that they were much lower than with CM-444 (Figure 6B). Regarding H3K27Ac, there were hardly any differences in the localization of the peaks after Panobinostat treatment compared to the control (Figure 6B). This is reflected in the different peaks observed in genes involved in AML cells differentiation (Figure 6C), where only minor changes were observed in comparison with untreated cells. All these results demonstrated that our compounds modulate the localization of BRDs, as well as acetyl histone marks, in a different way than Panobinostat. This could explain the different potential to induce differentiation of AML of our compounds in comparison with other DACi”.*

Reviewer Figure 1: Role of BRDs in the differentiation induction in AML with CM-444 and CM-1758. **A**) BRD4 (left), H3K27Ac (middle) and H3K9Ac (right) peak numbers in HL-60 cells treated with CM-444 or Panobinostat for 12 hours. **B**) CUT&RUN peaks distribution of BRD4 (left), H3K27Ac (middle) and H3K9Ac (right) in HL-60 cells treated with CM-444 or Panobinostat for 12 hours. **C**) Examples of myeloid genes involved in differentiation of AML cells. **D–E**) Cell differentiation assay measuring CD11b by flow cytometry after treating **D**) HL-60 and **E**) ML-2 cells daily with 25% GI_{50} of CM-444, CM-1758, Molibresib, JQ1, and the combination of CM-444 or CM-1758 with Molibresib or JQ1 for 48 h. Error bars indicate the S.D. of three replicates. Statistical significance was calculated by a two-tailed Student's *t*-test. n.s. = non-significant; **p* ≤ 0.05. **F–G**) q-PCR of *GATA2*, *PU.1*, *SCL*, and *CEBPA* after treating **F**) HL-60 and **G**) ML-2 cells daily with 25% GI_{50} of CM-444, CM-1758,

Molibresib, JQ1 and the combination of CM-444 or CM-1758 with Molibresib or JQ1 for 48 h. Error bars indicate the S.D. of three replicates. Statistical significance was calculated by a two-tailed Student's *t*-test. n.s. = non-significant; * $p \leq 0.05$.

We agree with the reviewer that treatment with our compounds as well as reference BRD inhibitors induced a downregulation of *MYC*, which may seem at odds with the previous described differences between them. There are several potential explanations: 1) BRD inhibitors have demonstrated to directly induce a downregulation of *MYC* (Delmore J.E. et al, Cell, 2011). In that sense, the use of BRD inhibitors in combination with CM-444 or CM-1758 may not be able to revert the downregulation of *MYC* as the direct effect of BRD inhibition is a decreased in *MYC* expression. 2) The differences between the behavior of genes involved in differentiation and *MYC* may also be explained by the fact that BET inhibitors as well as our compounds induced an inhibition of proliferation, which is associated with *MYC* downregulation. 3) Finally, results from the acetylome, indicate that treatment with CM-444 and CM-1758 induced a downregulation of *MYC* acetylation in the residue K148. *MYC* acetylation in the residue K148 by the acetyltransferase EP300 has been associated with *MYC* stabilization (Lynch J.T. et al., Cell Death Dis, 2013; Faiola F, Mol Cell Biol, 2005). In this sense, the inhibition of *MYC* K148 acetylation after treatment with our compounds could lead to a decrease in *MYC* levels. Finally, unlike *MYC*, genes involved in differentiation do not undergo changes in protein acetylation after treatment with our compounds. This may explain the different effect observed in the expression of *MYC* and the expression of differentiation genes in AML cells after treatment with BET inhibitors and our compounds.

2. To better delineate the effect of CM-444 and CM-1758 on BRD activity, the authors should examine whether the differentiation phenotype manifestly related to disruption of acetylation of non-histone proteins eg. BRDs versus for eg. acetylation of histone targets eg. H3K9 H3K122 H3K14 (see point 2 & This can be undertaken via

a. Assessment of chromatin accessibility (eg. ATACseq), and distribution of chromatin marks eg. H3K9Ac, and BRD4 associated H3K122 (Devhaiah et al., Nat Struct Mol Biol 2016) and BRD1 associated H3K14 (Mishima et al., Blood 2011) with treatment by CM-444 and CM-1758 on AML cell line(s), in addition to distribution of BRD4 and BRD1 themselves by ChIP with and without treatment.

b. Relating these epigenetic changes to *MYC* and other myeloid transcription factors as the authors have examined in their transcriptional analysis

The findings from these experiments should support/or not support the hypotheses, including whether the primary activity these agents is via activation of the BRD proteins as the authors propose

We thank the reviewer for his/her comments. Following the reviewer suggestion, we have performed additional CUT&RUN experiments with H3K9Ac, H3K27Ac, H3K122Ac and H3K14Ac in the AML cell line HL-60 after treatment with CM-444 treatment for 12 hours. Unfortunately, we were not able to successfully analyze the H3K122Ac and H3K14Ac marks either using CUT&RUN or ChIP-seq as a high background was observed undistinguishable from IgG negative control.

We were successful using H3K9Ac, H3K27Ac marks where we observed 15101 and 13454 peaks for H3K27Ac and H3K9Ac respectively in the control samples (HL-60 before treatment) and 48991 and 7568 peaks after treatment for 12 hours with CM-444. As we have described for BRD4, we observed a shift in the location of the peaks to active intronic and distal intergenic regions instead of promoter regions for both H3K9Ac, H3K27Ac. Moreover, CM-444 treatment drove H3K9Ac and H3K27Ac to genes related to myeloid differentiation such as *CB11b*, *ABII*, *GATA2* and *GFII* (Reviewer Figure 1C). These results suggest that both histone and non-

histone protein acetylation are important for the induction of myeloid differentiation by CM-444 and CM-1758.

To further differentiate the mechanism of action of our compounds with the reference DACi we performed CUT&RUN experiments in HL-60 cells using Panobinostat. Treatment with Panobinostat was associated with 16030, 26959 and 32775 peaks for BRD4, H3K27Ac and H3K9Ac respectively. As shown in Reviewer Figure 1 there was some overlap in the peaks between Panobinostat and our compounds. Interestingly, we observed some changes in the location of peaks in BRD4 and H3K9Ac with an increase in active regions similar to what was observed after treatment with CM-444 (although to a significantly lesser extent) while there were hardly any differences in the localization of the peaks in H3K27Ac after Panobinostat treatment compared to the control (Reviewer Figure 1B). These changes are exemplified in the differences observed in genes involved in AML cells differentiation (Reviewer Figure 1C).

These results have been included in the revised version of the manuscript (Results section, page 10) and the Reviewer Figure 1 has been included as Figure 6.

Results section, page 10: *“To delve into the role that BRDs could play in the differentiation process induced by our compounds but not by other DACi, we performed CUT&RUN experiments with BRD4 and the specific histone marks H3K9Ac and H3K27Ac in HL-60 after treatment with CM-444 or Panobinostat for 12 hours. The numbers of peaks of the different marks are shown in Figure 6A. Interestingly, CM-444 induced changes in the location of peaks associated with BRD4, H3K27Ac and H3K9Ac. While BRD4, H3K27Ac and H3K9Ac peaks were predominantly located in promoters in untreated cells, after treatment with CM-444 the majority of the peaks were located in intronic and distal intergenic regions (Figure 6B). Moreover, CM-444 treatment drove BRD4, H3K9Ac and H3K27Ac to genes related to myeloid differentiation and we observed that BRD4 peaks after CM-444 treatment were also associated with active chromatin regions as demonstrated by the increase in H3K27Ac mark (Figure 6C). Regarding the global changes carried out by Panobinostat, it should be highlighted that they were much lower than with CM-444 (Figure 6B). Regarding H3K27Ac, there were hardly any differences in the localization of the peaks after Panobinostat treatment compared to the control (Figure 6B). This is reflected in the different peaks observed in genes involved in AML cells differentiation (Figure 6C), where only minor changes were observed in comparison with untreated cells. All these results demonstrated that our compounds modulate the localization of BRDs, as well as acetyl histone marks, in a different way than Panobinostat. This could explain the different potential to induce differentiation of AML of our compounds in comparison with other DACi”.*

Minor comment

1. If the CM-444 and CM-1758 compounds act via a predominant BRD axis as the authors propose, the authors should discuss whether the broad spectrum of activity CM-444 and CM-1758 is relevant. Similarly, if BRD activation is indeed a mechanism of action, they should contrast this activity to BRD inhibitory compounds that have already been developed and the implications as a therapeutic strategy (eg. novelty).

We believe that the main mechanism of action behind the induction of differentiation of AML cells is mediated through BRD and thus the impact of BET inhibitors on the effect of CM-444 and CM-1758 as we have described above. However, we should stress that CM-444 and CM-1758 are DACi, and like other DACi have the potential to induce acetylation of other histone and non-histone proteins which may induce other effects and involved different mechanisms of action that should be explored. The broad spectrum of activity might be consider an advantage for future applications provided it is not associated with deleterious or adverse effects. These should be explored during the development of the new compounds.

2. I would suggest the authors enunciate the central tenet in their strategic approach more in their manuscript, including in the discussion ie. adopting a strategy for "differentiation potential" rather than proliferation and cell death endpoints. This represents the novel aspect of their drug discovery strategy that has resulted in identification of compounds that minimise tissue toxicity.

We thank the reviewer for his/her suggestion. We agree with the reviewer that part of the novelty of this work is the potential of CM-444 and CM-1758 DACi as differentiation therapy for AML together with the description of their mechanism of action. Thus, we have tried to highlight this aspect along the manuscript as it can be seen below (red colored texts):

Title, page 1: "Novel epigenetic based differentiation therapy for Acute Myeloid Leukemia".

Abstract, page 3: "In summary, these compounds may represent effective differentiation-based therapeutic agents across AML subtypes with a novel mechanism for treatment of AML".

Introduction section, page 4: "These compounds represent a novel and promising approach for a differentiation-based therapy for testing in AML patients".

Results section, page 5: "Identification of CM-444 and CM-1758 as novel pan-HDACi with capacity to induce differentiation of AML cells".

Results section, page 5: "In pursuit of epigenetic inhibitors capable of inducing myeloid differentiation in AML cells, we performed a dual differentiation-apoptosis assay with these epigenetic small molecules in the HL-60 cell line".

Results section, page 5: "Finally, we selected CM-444 and CM-1758 as our lead compounds because they showed higher differentiation capacity with lower apoptosis induction (Figure 1A and Table S1)".

Results section, page 6: "In summary, these results confirmed that CM-444 and CM-1758 were novel and potent pan-HDACi compounds with a high capacity to promote myeloid differentiation in AML cell lines at low non-cytotoxic doses".

Results section, page 6: "CM-444 and CM-1758 induce cell differentiation of genetically diverse subtypes of AML".

Results section, page 6: "Based on our findings demonstrating the CM-444 and CM-1758 capacity of differentiating HL-60 cells, we evaluated the effects of both compounds in a panel of 15 AML cell lines belonging to different AML subtypes (M1 to M7, according to FAB classification)."

Results section, page 6: "We observed that CM-444 and CM-1758 induced cell differentiation in all subtypes of AML, independently of the genetic alterations, mutations or translocations that were present (Figure 2A)."

Results section, page 6: "Remarkably, CM-444 and CM-1758 also promoted myeloid differentiation of 8 different patient-derived AML myeloid blast cells with distinct genetic translocations and gene mutations".

Results section, page 6: "All these results demonstrated that CM-444 and CM-1758 induced a wide spread differentiation of genetically diverse AML cells which was associated with expression of key transcription factors involved in myeloid differentiation."

Results section, page 7: “CM-444 and CM-1758 showed *in vivo* myeloid differentiation induction and anti-leukemia activity in AML.”

Results section, page 7: “Remarkably, ML-2 as well as MV4-11 cells from tumors treated with CM-444 and CM-1758 exhibited myeloid cell differentiation, as evidenced by the significant increase in CD11b expression (Figure 3C and Figure S4C)”:

Results section, page 8: “Overall, these results demonstrate that CM-444 and CM-1758 induce *in vivo* myeloid cell differentiation and potent anti-leukemia activity in AML.”

Results section, page 8: “As HDACi regulate the acetylation of histones but also non-histone proteins, to elucidate the molecular mechanism underlying the differentiation capacities of CM-444 and CM-1758 in comparison with commercial HDACi, we analyzed the total proteome and acetylome in an ML-2 cell line.”

Results section, page 9: “This differential effect could be key to discover the molecular mechanism underlying the increased ability to induce myeloid differentiation shown by CM-444 and CM-1758.”

Results section, page 10: “However, since most of these non-histone proteins are involved in this complex enhancer–promoter chromatin structure, this observation suggests that the global modification at the acetylation level after treatment with CM-444 or CM-1758 could directly regulate the expression of transcription factors involved in myeloid differentiation.”

Results section, page 11: “These results indicate that acetylation of BRD proteins, especially BRD4 and BRD1, and other non-histone proteins involved in this enhancer–promoter chromatin regulatory complex is essential to enhance the expression of key transcription factors that induce myeloid differentiation. In summary, these results show that BRDs have an essential role in the differentiation therapy exerted by CM-444 or CM-1758 in AML cells.”

Discussion section, page 12: “These preclinical results indicate the potential of our DACi in differentiation therapy.”

Discussion section, page 13: “The identification of this mechanism together with the capacity of inducing myeloid differentiation in all AML subtype cells adds further importance to the discovery of these DACi, which represent a novel and promising approach to differentiation-based therapy for testing in AML patients.”

3. The authors should discuss the future aims regarding their compounds, including the scope for therapeutic development. Would their broad spectrum of activity preclude further therapeutic development in human or is this mitigated by preclinical evidence for a large therapeutic window of activity? Are these pilot compounds useful for biochemical and biological characterisation in preclinical studies? Can and will these compounds be modified for specificity of targeting against DAC protein groups?

The potential for clinical application is eventually the aim of any medical chemistry development. In our work we have developed new probes that have been characterized at the biochemical and biological level and have demonstrated promising *in vitro* and *in vivo* effects inducing AML differentiation. Next steps should involve identification and optimization of a lead compound. We have performed ADME studies (Table S6), cardiovascular safety assays (Table S6), pharmacokinetic studies (Table S8, S9, S10 and S11) as well as cytotoxicity assays in the non-tumoral hepatic cell line THLE-2 and peripheral blood mononuclear cells from healthy donors (Table S7) as well as *in vivo* toxicity studies, including hematological and hepatic parameters and have found no evidence of limiting toxicities (Figure S3A, S3B and

S3C). A therapeutic window has also been demonstrated but additional optimization might be required for human use as well as GLP studies.

This has been commented in the revised version of the manuscript (Results section, page 7).

Results section, page 7: “Compared with the *in vitro* activity observed in the AML cell lines, both compounds showed an optimal therapeutic window, but additional optimization might be required for human use as well as GLP studies”.

4. Please check the use of the word “beside” on page 5.

It has been modified in the revised version of the manuscript.

Results section, page 6 “These results were validated *in vitro*, demonstrating that AML differentiation was associated, in addition to induction of *CD11b*, with down-regulation of *MYC* (Figure 2E and Figure S2F), an increase in the cell-cycle inhibitors *CDKN2A* (p21) and *CDKN1A* (p16) (Figure 2E and Figure S2F) together with cell-cycle arrest (Figure 2F and Figure S2G).”

5. Better representative images (Figure 2H and Figure S2F) should be provided to support the author claims (eg. oil immersion lens) and better white balance, as well as provide an image for ML-2 treated with CM-1758. While the image of ML-2 treated with CM-444 may support the contention for morphological differentiation, I remain to be convinced for both the HL-60 treated panels. (These findings however, are not materially significant to the authors’ ultimate conclusions).

We agree with the reviewer and following his/her suggestion we have repeated these experiments obtaining better and more representative images of HL-60 and ML-2 cell lines (Reviewer Figure 2). Reviewer Figure 2 has been included as Figure 2H in the revised version of the manuscript.

6. I am unable to find the Figures panel S2F the authors refer to in the manuscript (Page 5)

We apologize for this mistake. Any indication to Figure S2F has been removed in the revised version of the manuscript.

Reviewer #2 (Remarks to the Author):

Review

The manuscript presented by Edurne San José-Enériz et al. describes two novel HDAC inhibitors with potential to differentiate AML cell lines, which might be applicable as a therapeutic regimen for cancer therapy. The inhibitors induce acetylation of histone and non-histone proteins and inhibit AML tumor growth in vivo in mouse models. The authors suggest BRD1 and BRD4, two acetyl-lysine reader proteins, as the main targets of the compound to induce AML differentiation.

In general, the results presented are of interest for the field, however additional evidence is needed as to the mechanism of action of the drugs. Some of the presented data lack detail and the used methodology for histone-acetylomics appears unsuitable.

Specific comments:

1) CM-444 is a bispecific inhibitor of HDACs and DNMTs. The authors claim, that no changes in DNA methylation was detectable after incubation of HL-60 cells for 48 hours, due to low doses used (Fig 1E). In order to detect DNA methylation changes, cells need to proliferate for several rounds, since DNA demethylation will occur only during replication as a passive effect of DNMT inhibition. Thus, the experiments should be repeated for longer time-points. The illustration of DNA methylation levels in color gradients is hard to assess and should be shown as percent methylation. A positive control such as decitabine treatment should be added. The quality of histone western blots in Fig. 1D lower panel and S1A should be improved.

We completely agree with the reviewer that 48 hours is not enough time to detect DNA methylation changes. Therefore, we have repeated the experiments using 4 different AML cells lines. HL-60, ML-2, MOLM-13 and MV4-11 were treated with CM-444 and CM-1758 using their corresponding GI_{50} for 10 days (HL-60 and ML-2) or for 5 days (MOLM-13 and MV4-11). As suggested by the reviewer, we also included Decitabine treatment as positive control. As shown in Reviewer Figure 3, we did not detect any changes in DNA methylation levels of *LINE-1* after treatment with CM-444 or CM-1758. However, DNA methylation of *LINE-1* decreased after treatment with Decitabine (positive control). These results indicate that CM-444 and CM-1758 do not induce changes in DNA methylation, at least at the doses tested.

Reviewer Figure 3. DNA methylation analysis after CM-444 and CM-1758 treatment. DNA methylation of *LINE-1* analyzed by pyrosequencing after daily treatment with CM-444 or CM-1758 in HL-60 and ML-2 cell lines for 10 days or MOLM-13 and MV4-11 cells for 5 days. The DNA methylation percentage is indicated inside the circles. Treatment with Decitabine was used as a positive control of DNA demethylation. As a DNA methylated control, a universally methylated DNA was used. The data shown are the mean of two independent experiments.

In accordance with the reviewer's suggestion, we have modified Figure 1E and Figure S1B in the revised version of the manuscript, indicating the percentage of DNA methylation in each case (Reviewer Figure 4 and Reviewer Figure 5).

Reviewer Figure 4. DNA methylation analysis after CM-444 and CM-1758 treatment. DNA methylation of *LINE-1* analyzed by pyrosequencing after daily treatment in HL-60 cell line with 270 nM CM-444 or 300 nM CM-1758 for 48 h. The DNA methylation percentage is indicated inside the circles. As a DNA methylated control, a universally methylated DNA was used. The data shown are the mean of two independent experiments.

Reviewer Figure 5. DNA methylation analysis after CM-444 and CM-1758 treatment. DNA methylation analysis by pyrosequencing of *LINE-1* was performed after treatment. Universally methylated DNA was used as DNA methylated control. The DNA methylation percentage is indicated inside the circles. The data shown the mean of two independent experiments.

As suggested also by the reviewer, we have repeated the western blots included in Figure 1D and Figure S1A in order to improve their quality (Reviewer Figure 6 and Reviewer Figure 7).

Reviewer Figure 6. H3Ac and H3K27me3 western blot in HL-60 cell line. H3Ac and H3K27me3 levels detected by western blot after daily treatment of HL-60 cell line with 270 nM CM-444 or 300 nM CM-1758 for 48 h. H3 total was used as the loading control (representative experiment of 2 independent studies).

Reviewer Figure 7. H3Ac and H3K27me3 western blot in ML-2, MOLM-13 and MV4-11 cell lines. ML-2, MV4-11, and MOLM-13 cell lines were treated daily for 48 hours with 260, 160, and 280 nM of CM-444 and 210, 80, and 140 nM of CM-1758, respectively. H3aC and H3K27me3 levels were detected by western blot after treatment. The experiment was repeated twice with similar results.

In the revised version of the manuscript, changes have been included in the results section (page 5-6). Reviewer Figure 3 has been included as Figure S1D, Reviewer Figure 4 as Figure 1E, Reviewer Figure 5 as Figure S1B, Reviewer Figure 6 as Figure 1D and finally Reviewer Figure 7 as Figure S1A.

Results section, page 5-6: “To definitely demonstrate that our compounds do not induce changes in DNA methylation, we treated AML cell at long-term (10 days for HL-60 and ML-2 and 5 days for MOLM-13 and MV4-11) with CM-444 and CM-1758 and then analyzed DNA methylation levels of LINE-1 (Figure S1D).”

2) The authors use Cd11b as a proxy for differentiation into myeloid lineage. It would be interesting to add additional surface markers to define specific subsets of myeloid cells.

Following the reviewer suggestions, we have performed a more complete differentiation analysis including additional myeloid surface markers. HL-60, ML-2, MOLM-13 and MV4-11 were treated with CM-444 and CM-1758 for 48h after which we analyzed the levels of CD11b, CD15, CD14, CD13, HLA-DRPB and CD33 by flow cytometry. CD15 and CD33 markers turned out to be uninformative since the AML cell lines tested had already 100% expression at baseline. However, CD11b, CD13, CD14 and HLA-DR levels increased with both treatments in all AML cell lines. The only exceptions were the case of CD14 for MV4-11 and CD13 for HL-60, where no changes were observed. These results indicate that CM-444 and CM-1758 induce myeloid differentiation in AML cell lines.

Reviewer Table 1. Analysis of surface myeloid markers after CM-444 and CM-1758 treatment.

Cell line	Treatment	CD11b (%)	CD13 (%)	CD14 (%)	HLA-DR (%)
HL-60	Control	11.7	99.6	2.35	2.48
	CM-444	74.3	100	21.5	9.16
	CM-1758	71.2	100	18.9	12.2
ML-2	Control	15	77.3	2.10	4.38
	CM-444	80.1	100	5.20	35.7
	CM-1758	75.6	98.6	2.86	17.4
MOLM-13	Control	39.6	10.5	18.5	10.4
	CM-444	92.8	49.3	66.5	35.1
	CM-1758	93.0	54.8	35.6	27.9
MV4-11	Control	10.2	62.7	12.3	12.5
	CM-444	58.9	98.0	57.6	42.1
	CM-1758	57.9	98.1	56.2	38.4

These results have been included in the revised version of the manuscript with a new table (**Table S4**) and in the results section (page 6).

Results section, page 6: “Furthermore, a more complete analysis including more myeloid markers revealed an increase of CD13, CD14 and HLA-DR after CM-444 and CM-1758 treatment (Table S4)”.

3) Although both compounds induced high levels of differentiation in HL-60 and ML-2 cells, CM-444 also induced significant apoptosis in MOLM-13 and MV4-11 cells. The authors should comment on this.

Differentiation of all AML cell lines after treatment with CM-444 and CM-1758 was associated with some common changes: 1) overexpression of several mature myeloid markers such as CD11b, CD13, CD14 and HLA-DR (Reviewer Table 1); 2) *MYC* downregulation which is associated with myeloid cell differentiation, (Figure 2D, 2F and Figure S2D) and 3) inhibition of cell proliferation, as shown in Figure 2E and Figure S2C (upregulation of cell cycle inhibitors p16 and p21) and in Figure 2F and Figure S2D (cell cycle arrest) in all tested AML cell lines. However, some changes, as pointed out by the reviewer, were specific in certain cells lines. Treatment with CM-444 and CM-1758 specifically induced apoptosis in MOLM-13 and MV4-11 cell lines and not in HL-60 and ML-2. These differences could be related to differences in sensitivity to the compounds associated with the different genetic background of the cells lines, as for instance only MOLM-13 and MV4-11 are *FLT3-ITD* positive cells. In addition, cell differentiation eventually should lead to apoptosis/cell death, which might also be observed if cultures were to be prolonged for longer times. In any case, we believe that the fact that cell differentiation is observed consistently in all cell lines is the most relevant point representing as a potential therapeutic approach in AML.

4) The RNA-seq data should be described in more detail, eg. adding heatmaps and PCA blots to see overall changes following the treatments. How many genes were deregulated upon treatment, how many up- or down-regulated?

Following the reviewer suggestions, a detailed analysis of the RNA-seq data has been included in the revised version of the manuscript. Differentially expressed genes were considered significant when a fold change (FC) > 0.4 or < -0.4 and $\text{padj} < 0.05$ was observed. In the case of HL-60 cell line there were 227 (39 down-regulated and 188 up-regulated) and 3299 (1610 down-regulated and 1589 up-regulated) deregulated genes by CM-444 and CM-1758 treatment, respectively. In the case of ML-2 cells, the number of deregulated genes by CM-444 and CM-1758 were 558 (88 down-regulated and 470 up-regulated) and 3750 (2024 down-regulated and 1726 up-regulated), respectively (Reviewer Figure 8A). The majority of the genes modulated by CM-444 were also modified by CM-1758 even though CM-1758 induced additional transcriptional changes as can be observed in the PCA (Reviewer Figure 8A-B). We also observed differences in transcriptional regulation between cell lines (Reviewer Figure 8C).

Reviewer Figure 8: RNA-seq data of HL-60 and ML-2 treated with CM-444 or CM-1758. A) Venn diagram of differentially expressed genes after CM-444 or CM-1758 treatment in HL-60 and ML-2 cell lines. **B)** PCA of RNA-seq data from HL-60 and ML-2 cells after treatment with CM-444 or CM-1758 compared with untreated cells. **C)** Unsupervised hierarchical cluster of differentially expressed genes after treatment with CM-444 or CM-1758 compared with untreated cells in HL-60 and ML-2 cell lines.

These results have been included in the revised version of the manuscript as a new supplemental figure (**Figure S2B, S2C and S2D**) and in the results section (page 6). In addition the RNA-seq results can be downloaded from the Gene Expression Omnibus public functional genomics data repository under the accession number GSE219230.

Results section, page 6: “After treatment, 227 (39 down-regulated and 188 up-regulated) and 3299 (1610 down-regulated and 1589 up-regulated) genes were deregulated by CM-444 and CM-1758, respectively in HL-60 cells. In the case of ML-2 cell line, the number of deregulated genes by CM-444 and CM-1758 were 558 (88 down-regulated and 470 up-regulated) and 3750 (2024 down-regulated and 1726 up-regulated), respectively (Figure S2B). Principal component analysis (PCA) and unsupervised hierarchical clustering analysis of RNA-seq data showed first, the differences between cell lines and subsequently, between treatments (Figure S2C and S2D).”

5) The in vivo data show reduced engraftment of pretreated cells. Fig.3A and S4A do not show error bars. It is unclear how many replicates were used and how significance was calculated? The same is true for Fig.3B and S4B, where replicates are not visible in the “Tumor volume” graphs.

We thank the reviewer for the observation, as this was a mistake that has been corrected in the revised version of the manuscript. The number of animals used is included in the materials and methods section as well as the statistical analysis employed.

Figure legends section, page 20-21: “Figure 3: CM-444 and CM-1758 induction of differentiation and anti-leukemic activity in vivo. A) ML-2 cells were pretreated in vitro with 260 nM of CM-444 or 210 nM of CM-1758 for 96 hours. After verifying CD11b induction by flow cytometry, equal amounts of cells were injected subcutaneously in Rag2^{-/-} γc^{-/-} mice, and tumor volume was measured (n = 8). Statistical significance was calculated by a two-tailed Student’s t-test. ***p ≤ 0.001. B) Schematic diagram of the in vivo CM-444 and CM-1758 treatment procedure and tumor volume curve of the ML-2 subcutaneous xenograft model in

*Rag2^{-/-} γc^{-/-} mice (n = 8). Statistical significance was calculated by a two-tailed Student's t-test. *p ≤ 0.05”.*

Materials and methods section, page 37: “Then, 5 × 10⁶ cells pretreated with CM-444, CM-1758, or vehicle (80% saline solution, 10% DMSO, and 10% Tween 20) and diluted in 100 μL of saline solution, were implanted subcutaneously into the flank of female BALB/cA Rag2^{-/-} γc^{-/-} mice between 6 and 8 weeks of age (n = 8).”

Materials and methods section, page 37-38: “The subcutaneous xenograft models of ML-2 and MV4-11 derived cells, 5 × 10⁶ and 10 × 10⁶ cells, respectively, diluted in 100 μL of saline solution were subcutaneously inoculated in the back left flank of female BALB/cA Rag2^{-/-} γc^{-/-} mice. When tumors became palpable, the mice were randomized into three groups: vehicle, CM-444, and CM-1758, (eight animals/group).”

Supplemental information, page 7: “CM-444 and CM-1758 induction of differentiation and anti-leukemic activity. A) HL-60 cells were pretreated *in vitro* with 270 nM of CM-444 or 300 nM of CM-1758 for 96 hours. After verifying CD11b induction by flow cytometry, equal amount of cells were injected subcutaneously in Rag2^{-/-} γc^{-/-} mice, and tumor volumes were measured (n = 8). Statistical significance was calculated by a two-tailed Student's t-test. **p ≤ 0.01.”

Supplemental information, page 7-8: “Schematic diagram of *in vivo* CM-444 and CM-1758 treatment procedure and tumor volume curve of MV4-11 subcutaneous xenograft model in Rag2^{-/-} γc^{-/-} mice (n = 8). Statistical significance was calculated by a two-tailed Student's t-test. n.s. = non-significant; *p ≤ 0.05.”

In the revised version of the manuscript, the number of mice and error bars have been included in each figure (Reviewer Figures 9-12 which are Figure 3A and 3B and Figure S4A and S4B in the revised version of the manuscript).

Reviewer Figure 9. CM-444 and CM-1758 induction of differentiation and anti-leukemic activity *in vivo*. ML-2 cells were pretreated *in vitro* with 260 nM of CM-444 or 210 nM of CM-1758 for 96 hours. After verifying CD11b induction by flow cytometry, equal amounts of cells were injected subcutaneously in Rag2^{-/-} γc^{-/-} mice, and tumor volume was measured (n = 8). Statistical significance was calculated by a two-tailed Student's t-test. ***p ≤ 0.001.

Reviewer Figure 10. CM-444 and CM-1758 induction of differentiation and anti-leukemic activity *in vivo*. Schematic diagram of the *in vivo* CM-444 and CM-1758 treatment procedure and tumor volume

curve of the ML-2 subcutaneous xenograft model in Rag2^{-/-} γc^{-/-} mice (n = 8). Statistical significance was calculated by a two-tailed Student's t-test. *p ≤ 0.05.

Reviewer Figure 11. CM-444 and CM-1758 induction of differentiation and anti-leukemic activity *in vivo*. HL-60 cells were pretreated *in vitro* with 270 nM of CM-444 or 300 nM of CM-1758 for 96 hours. After verifying CD11b induction by flow cytometry, equal amount of cells were injected subcutaneously in Rag2^{-/-} γc^{-/-} mice, and tumor volumes were measured (n = 8). Statistical significance was calculated by a two-tailed Student's t-test. **p ≤ 0.01.

Reviewer Figure 12. CM-444 and CM-1758 induction of differentiation and anti-leukemia activity *in vivo*. Schematic diagram of *in vivo* CM-444 and CM-1758 treatment procedure and tumor volume curve of MV4-11 subcutaneous xenograft model in Rag2^{-/-} γc^{-/-} mice (n = 8). Statistical significance was calculated by a two-tailed Student's t-test. n.s. = non-significant; *p ≤ 0.05.

6) For Fig.S4D and S4E the authors should provide the original histone blots.

As suggested by the reviewer, we have included in the revised version of the manuscript the original western blots corresponding to these figures (Reviewer Figure 13 which are Figure S4D and S4E in the revised version of the manuscript).

Reviewer Figure 13. Relative H3Ac/H3 total levels after *in vivo* CM-444 and CM-1758 treatment in s.c. ML-2. D–E) Relative H3Ac/H3 total levels after *in vivo* CM-444 and CM-1758 treatment in s.c. ML-2 (D) and MV4-11 (E) mouse model. Statistical significance was calculated by a two-tailed Student's t-

test. n.s. = non-significant; *p ≤ 0.05; **p ≤ 0.01; ****p ≤ 0.0001. The original western blots are shown below.

7) In their proteomic analysis, the authors reveal 29 proteins that were commonly deregulated upon treatment with CM-444 and CM-1758 or with the reference HDACi Panobinostat and Vorinostat. The authors should show these proteins and comment on their function and relevance. Did the authors observe overlap also between RNA-seq and proteome data in terms of up and down regulated genes/proteins?

As suggested by the reviewer, we have carried out a thorough study of the 29 proteins deregulated by the four HDACi. A heatmap with the logFC of these proteins is shown in Reviewer Figure 14.

Reviewer Figure 14. Heatmap view of proteins commonly deregulated with CM-444, CM-1758, Panobinostat and Vorinostat in ML-2. log FC data are represented.

In order to analyze in more detail the relevance of these 29 proteins, we performed a Gene Ontology (GO) analysis and found proteins related to cell cycle (GO:0051301) and mitotic spindle assembly checkpoint (GO:0007094). Additional analysis of the protein functions is summarized in Reviewer Table 2. Consistent with the GO results, we found several proteins related to cell cycle (INIP, SYCP1, BUB1, SSNA1, CDC20, HAUS6) and cell death (DIABLO or UBL5). Several metabolic proteins were included among these 29 proteins (GLUL, ETNK1, PLCG1) as well as proteins implicated in ubiquitination process (UBEL5, RNF126, UBE2F) or transcriptional regulation (SATB1, NSD3, ZNF740, CCDC59). All these results together could be a reflection of the general effect of HDACi in cell cycle and cell proliferation and in the regulation of transcription and gene expression mediated by HDAC.

Reviewer Table 2. Information about the function of the 29 proteins commonly regulated by CM-444, CM-1758, Panobinostat and Vorinostat. Rows highlighted in blue indicate metabolic proteins; rows highlighted in yellow show proteins related to cell cycle; rows highlighted in pink indicate proteins involved in cell death.

Protein	Function
GLUL1	Glutamine synthetase.
SLC4A2	Anion exchange protein 2; Plasma membrane anion exchange protein of wide distribution.
SATB1	DNA-binding protein SATB1; Matrix protein that binds to DNA and recruits chromatin-remodeling factor to regulate chromatin structure and gene expression.
DIABLO	Diablo IAP-binding mitochondrial protein. Related to apoptosis.
NSD3	Histone methyltransferase. Preferentially dimethylates 'Lys- 4' and 'Lys-27' of histone H3 forming H3K2me2 and H3K27me2.
SYCP1	Synaptonemal complex protein 1; Major component of the transverse filaments of synaptonemal complexes, formed between homologous chromosomes during meiotic prophase.
INIP	Subunit of single-stranded DNA binding complexes involved in G2/M checkpoint control.
ZNF740	Zinc finger protein 740; May be involved in transcriptional regulation. Enables sequence-specific double-stranded DNA binding activity.

BLOC1S4	Biogenesis of lysosome-related organelles complex 1 subunit 4; Component of the BLOC-1 complex, a complex that is required for normal biogenesis of lysosome-related organelles.
GIPC1	PDZ domain-containing protein GIPC1; May be involved in G protein-linked signaling. Scaffolding protein that regulates all surface receptor expression and trafficking.
SLC35E1	Solute carrier family 35 member E1; Putative transporter.
BUB1	Ser/Thr kinase implicated in mitosis checkpoint.
ETNK1	Ethanolamine kinase. Highly specific for ethanolamine phosphorylation. May be a rate-controlling step in phosphatidylethanolamine biosynthesis.
COG8	Conserved oligomeric Golgi complex subunit 8; Required for normal Golgi function.
PLCG1	1-phosphatidylinositol 4,5-bisphosphate phosphodiesterase gamma-1; Mediates the production of the second messenger molecules diacylglycerol (DAG) and inositol 1,4,5-trisphosphate (IP3). Plays an important role in the regulation of intracellular signaling cascades.
SSNA1	Microtubule-binding protein which stabilizes dynamic microtubules.
UBL5	Protein similar to ubiquitin and related to cell death.
CCDC59	Thyroid transcription factor 1-associated protein 26; Component of the transcription complexes of the pulmonary surfactant-associated protein-B (SFTPB) and -C (SFTPC). Enhances homeobox protein Nkx-2.1-activated SFTPB and SFTPC promoter activities.
PHKG2	Phosphorylase b kinase gamma catalytic chain; regulates glycogeneolysis.
RNF126	E3 ubiquitin-protein ligase RNF126; E3 ubiquitin-protein ligase that mediates ubiquitination of target proteins.
ANXA1	Annexin A1; Plays important roles in the innate immune response as effector of glucocorticoid-mediated responses and regulator of the inflammatory process.
FAM111A	Protein FAM111A; Chromatin-associated protein required for PCNA loading on replication sites. Promotes S-phase entry and DNA synthesis.
NECAP1	Adaptin ear-binding coat-associated protein 1; Involved in endocytosis.
PPWD1	Peptidylprolyl isomerase domain and WD repeat-containing protein 1; PPIase that catalyzes the cis-trans isomerization of proline imidic peptide bonds in oligopeptides and may therefore assist protein folding.
NCK2	Cytoplasmic protein NCK2; Adapter protein which associates with tyrosine-phosphorylated growth factor receptors or their cellular substrates.
CDC20	Cell division cycle protein 20 homolog; Activator of APC/C, regulatory protein interacting with several other proteins at multiple points in the cell cycle.
UBE2F	NEDD8-conjugating enzyme UBE2F; Accepts the ubiquitin-like protein NEDD8 from the UBA3-NAE1 E1 complex and catalyzes its covalent attachment to other proteins.
HAUS6	HAUS augmin-like complex subunit 6; Contributes to mitotic spindle assembly, maintenance of centrosome integrity and completion of cytokinesis as part of the HAUS augmin-like complex.
LGALS8	Galectin-8; Beta-galactoside-binding lectin that acts as a sensor of membrane damage caused by infection and restricts the proliferation of infecting pathogens by targeting them for autophagy.

With regard to the correlation between RNA-seq and proteome data, we have compared both data finding low correlation between them. Specifically, only 2 and 43 proteins/genes are shared between proteome and RNA-seq data for CM-444 and CM-1758 treatments, respectively (Reviewer Figure 15). The analysis was performed 12 hours after treatment with HDACi, which may explain the discordance between RNA and protein data. While 12 hours may be enough time to observe changes at the acetylation or RNA level, changes in protein levels may require more prolonged times.

Reviewer Figure 15. Correlation between proteome and RNA-seq data. Venn diagrams of the genes / proteins deregulated with A) CM-444 and B) CM-1758 in the RNA-seq and proteome data.

However, it is worth noting that even if we detected a low correlation, the genes/proteins that correlate show a consistent behavior both being upregulated or downregulated after CM-444 or CM-1758 treatment (Reviewer Figure 16).

Reviewer Figure 16. Correlation between proteome and RNA-seq data. Venn diagrams of the common genes / proteins deregulated with A) CM-444 and B) CM-1758 in the RNA-seq and proteome data.

8) Fig. 4C shows the number of deregulated acetyl-K sites. Based on the numbers, the percentage of up-regulated sites for CM-444 should be 49% and not 79% as stated by the authors.

We apologize for this error, which has been corrected in the revised version of the manuscript. Together with this mistake we have found another one in the same sentence, where we exchanged the words Panobinostat and Vorinostat: “Acetyl-K sites in the treated cells was greater than the fraction of downregulated sites, with 76.6%, 61.2%, and 53.2% of the

acetylation sites upregulated by CM-444, CM-1758, and Vorinostat, respectively. Panobinostat was an exception, with only 37.3% of the Acetyl-K sites upregulated.” This mistake has also been corrected in the revised version of the manuscript.

Results section, page 9: “The fraction of upregulated and downregulated Acetyl-K sites in the treated cells was similar, with 49.7%, 61.2%, and 53.2% of the acetylation sites upregulated by CM-444, CM-1758, and Panobinostat, respectively. Vorinostat was an exception, with only 37.3% of the Acetyl-K sites upregulated (Figure 4C).”

9) Fig.4E shows the overlap of regulated sites between the 4 treatments. More detail on the overlapping and unique sites/affected proteins should be shown and discussed.

As suggested by the reviewer we have studied in more detail the 54 Acetyl-K sites regulated by the four HDACi. As shown in Reviewer Figure 17, from these 54 Acetyl-K sites, 41 corresponded to unique K residues in different proteins while 13 of them corresponded to distinct acetylations in the same protein (CDC27, CDK16, H3F3A, KAT6A, NUCKS1 and STAG2).

Reviewer Figure 17. Heatmap view of Acetyl-K sites commonly deregulated with CM-444, CM-1758, Panobinostat and Vorinostat in ML-2. log FC data are represented.

Proteins with acetyl-K residues deregulated by HDACi belong to GO terms related to transcription regulation (EP300, FOSL, H2AZ1, MYC, KAT6A, MED6, MEF2D, NUCKS1, SBNO1), chromatin remodeling (MYC, CHRAC1, KMT2B, NPM2), nucleosome assembly (H2BC19P, H2BC5, H3-3A, CHRAC1, KAT6A), cell division (STAG2, CDC27, CDCA7, SMC5) and acetylation (EP300, KAT6A, SMC5). KEGG and Reactome pathways showed these proteins to be involved in cell cycle and transcriptional deregulation in cancer (KEGG) and cell cycle, gene expression (transcription) and transcriptional regulation of granulopoiesis (Reactome). Besides, several G₂M checkpoints are among these proteins (H2AZ1, H2AZ2, MYC, CDC27, PBK, and LMNB1). These results are consistent with the described activity of HDACi regulating cell cycle, growth arrest, altered signaling and aberrant mitosis.

If we look into the unique acetyl K-sites sites/affected proteins, we found 202, 80, 232 and 121 acetyl K-sites regulated specifically by CM-444, CM-1758, Panobinostat and Vorinostat, respectively. In the manuscript we have already discussed about the specific K sites deregulated specifically by our compounds CM-444 and CM-1758. We found that these Acetyl K regulated proteins were involved in nucleic acid metabolism processes, gene expression, histone modifications and DNA repair, being many of them participating in the enhancer-promoter chromatin regulatory complex such as BRDs or members of the MOZ/MORF complex among others.

In the case of Panobinostat, the large amount of metabolic proteins is striking (HIBCH, ATIC, ACLY, NFS1, NME2, ACADM, ACADS, ACADVL, ACOX1, ALDH1B1, ALDOA, ALDOC, ASNS, BCAT2, DLAT, DBT, ECHS1, EHMT2, FASN, FH, GLUD1, EPRS1, GAPDH, HNMT, HADHB, HADH, HSD17B4, IDH3A, IDH1, KMT2D, MDH2, MTHFD1L, MT-ATP6, PRDX6, GART, PCCB, PDHX, SHMT2, SCP2, TALDO1, TPI1 and UQCRC2). The KEGG pathways showed enrichment of carbon metabolism, fatty acid metabolism, biosynthesis of amino acids, valine, leucine and isoleucine degradation, fatty acid degradation, citrate cycle, glycolysis / gluconeogenesis or pyruvate metabolism pathways, among others. It should be noted that other processes such as necroptosis are also enriched. The GO analysis corroborated

these results, showing a strong relationship with metabolism such as lipid metabolic process (ACLY, ACADM, ACADS, ACADVL, ACOX1, BCAT2), fatty acid beta-oxidation (ACADM, ACADS, ACOX1, CRAT, ECI1, ECH1, ECHS1, HADHB, HADH, HSD17B4, SCP2) or tricarboxylic acid cycle (ACLY, DLAT, FH, IDH3A, IDH1, MDH2) but also with positive regulation of apoptosis process (TNNBL1, CCAR2, HSPD1, HMGB1, ING3, ING5, PRKDC, RRP1B, PRS6), cell division (ARL8B, RAN, BOD1L1, CDC23, CDC42, DYNC1H1, HNRNPU, PRS3, SM2, SMC5, UBE2I) or positive regulation of transcription from RNA polymerase II promoter (EP300, MYBBP1A, MYB, NBE2, YBX1, ACTR3, AP3B1, HNRNPU, HMGA1, HMGB1, HCFC1, KMT2D, MED14, NCL, NPM1, PARP1, PRKDC, RNASEL, RRP1B) among others.

With Vorinostat we observed something similar to Panobinostat. Again, the large number of proteins related to metabolism is particularly striking (OXCT13, TP5F1A, ATP5F1D, NDUFAB1, NME2, ACADVL, ACSF3, AHCY, AGPS, CAT, CMPK1, DBT, ENO1, ESD, FASN, GOT2, GSTO1, HADHA, IDH1, IDH2, IVD, LDHA, KMT2C, MTHFD1, OGDH, PAICS, PDHB, SCP2, SUCLG1, TPI1). As expected, the classification of proteins according to the KEGG pathways analysis showed enrichment in metabolic pathways such as carbon metabolism, citrate cycle, fatty acid metabolism, valine, leucine and isoleucine degradation, biosynthesis of amino acids, propanoate metabolism or glycolysis / gluconeogenesis, among others. GO analysis revealed the relationship of these proteins with fatty acid metabolism process (ACSF3, FASN, HADHA, SNCA), glycolytic process (ENO1, LDHA, OGDH, TPI1), 2-oxoglutarate metabolic process (GOT2, IDH1, IDH2, GGD1), glutathione metabolic process (CLIC1, EEF1G, GSTO1, IDH1, SOD2) or tricarboxylic acid cycle (IDH1, IDH2, OGDH, PDH3, SUCLG1). As observed with Panobinostat, the proteins with acetyl K sites regulated by Vorinostat were also related to apoptosis processes (NME2, ARHGDI1, CAT, HSPA1B, RPS3A, SOD2, SNCA). Finally, another aspect to highlight is their involvement in chromatin remodeling and organization (EP300, ANP32E, KMT2C, NUCKS1, RNF20, UBN1, SMARCC2, SMARCA2, ACTB, KMT2C, RIOX2, RNF20).

Altogether, these results demonstrate that there is a mechanism of action shared by all HDACi as well as specific pathways affected by the different compounds. All HDACi have in common the capacity of modulating the acetylation of histone proteins as well as non-histone proteins related to cell cycle and gene expression and transcription regulation. Vorinostat and Panobinostat specifically regulate the acetylation of proteins related to metabolism and apoptotic processes. On the other hand, CM-444 and CM-1758 regulate acetyl K sites present in proteins that participate in the enhancer-promoter chromatin regulatory complex such as BRDs.

These results have been summarized in Reviewer Figure 18, which has been included in the revised version of the manuscript as Figure S7E. The results have been included in the results section (page 10) of the revised version of the manuscript.

Reviewer Figure 18. Gene Ontology analysis of the K-sites regulated specifically by CM-444 and CM-1758, Panobinostat and Vorinostat. 1: protein stabilization; 2: regulation of translation; 3: histone acetylation; 4: histone H3 acetylation; 5: positive regulation of histone H3-K36 trimethylation; 6: RNA splicing; 7: mRNA processing; 8: regulation of transcription, DNA-templated; 9: positive regulation of transcription, DNA-templated; 10: transcription, DNA-templated; 11: negative regulation of transcription, DNA-templated; 12: positive regulation of transcription elongation from RNA polymerase II promoter; 13: regulation of transcription from RNA polymerase II promoter; 14: positive regulation of transcription from RNA polymerase II promoter; 15: nucleosome assembly; 16: DNA repair; 17: chromatin remodeling; 18: chromatin organization; 19: fatty acid beta-oxidation; 20: lipid metabolic process; 21: fatty acid metabolic process; 22: fatty acid biosynthetic process; 23: organic acid metabolic process; 24: glycolytic process; 25: gluconeogenesis; 26: glutathione metabolic process; 27: tricarboxylic acid cycle; 28: 2-oxoglutarate metabolic process; 29: positive regulation of apoptotic process; 30: cell cycle; 31: regulation of G0 to G1 transition; 32: regulation of mitotic metaphase/anaphase transition; 33: regulation of G1/S transition of mitotic cell cycle.

Results section, page 10: “On the contrary, it is noteworthy that Vorinostat and Panobinostat specifically regulate the acetylation of proteins related to metabolism and apoptotic processes (Figure S7E). A detailed analysis of the 54 Acetyl-K sites commonly regulated by the four HDACi showed their implication in cell cycle, cell acetylation, transcription regulation and chromatin remodeling (Figure S7E). These results are consistent with the described activity of HDACi regulating cell cycle, growth arrest and altered signaling. Altogether these results demonstrate that there is a mechanism of action shared by all DACi as well as specific pathways affected by the different compounds”.

10) Commonly, histone acetylomics is performed via derivatization, since trypsin digest of histone proteins would yield very small fragments that are difficult to retain on HPLC columns and used for MS. Could the authors please comment on their methodology? The authors should also provide a list of mapped histone peptides including the acetylation sites.

Derivatization of histones with acylating agents such as propionic anhydride neutralizes lysine charges and generates longer tryptic peptides (Garcia B.A., et al, Nat Protoc, 2007). While this strategy is highly valuable when the goal is to specifically analyze histone modifications, the interest of our study focused instead on obtaining a global profile of Lys acetylated proteins (histones and non-histones). This was particularly relevant in our work considering that histone acetyl-transferases and de-acetylases modify numerous proteins beyond histones (Choudhary C., et al, Nat Rev Mol Cell Biol, 2014).

Our strategy is based on the immuno-purification of Lys-acetylated peptides from complex whole cell extracts using a pan-specific antibody followed by the LC-MS/MS analysis. This enables the identification of acetylation sites in hundreds of proteins (Choudhary C., et al, Science, 2009). Importantly, owing to the selectivity of the antibodies and the sensitivity of current MS instrumentation, this strategy also enables the identification of all major histone acetylation sites (Choudhary C., et al, Science, 2009). Indeed, in our data sets, we were able to map a very significant number of histone acetylation sites. These included well-known epigenetic marks such as:

- H3K9Ac (associated with active transcription, enriched at the promoter and enhancer regions; involved in DNA replication and repair)
- H3K14Ac (active transcription)
- H4K5Ac (gene expression and chromatin structure; maintenance of centromeric chromatin)
- H4K8Ac (gene expression, activation of genes involved in cell cycle progression)
- H4K12Ac (gene expression; regulation of stress-responsive genes)

Remarkably, we identified multiple histone acetylations in all core histones (including variants) as well as the linker histone H1. In response to the reviewer's suggestion, we have elaborated a new table (Reviewer Table 3) which presents all the acetylation sites of histone proteins identified in our data sets. It is important to note that the residues mentioned in this table refer to the unprocessed histone protein, which includes the N-termini Met residue.

Since all acetylome data have been deposited in the ProteomeXchange Consortium via the PRIDE partner repository (PXD038202), we have not included these results in the revised version of the manuscript.

11) The statement: “Interestingly, we observed that most of the Acetyl-K sites regulated after treatment with each of the four HDACi belong to non-histone proteins (>90%) instead of histone proteins” is unclear. Is this assumption based on the number of sites detected? Presumably histone proteins represent only a small fraction of overall proteins, thus this conclusion is not valid.

We agree with the reviewer that the number of non-histone proteins is larger so we have rephrased this sentence to avoid confusion.

Results section, page 9: “Interestingly, many of the Acetyl-K sites regulated after treatment with each of the four HDACi belong to non-histone proteins (Figure 4H)”.

12) Acetylomics of histones showed less separation in the PCA plot provided in Fig.4I and the authors argue “..that CM-444 or CM-1758 modulate specifically non-histone protein acetylation patterns in AML cells, suggesting that this differential effect could be key to discover the molecular mechanism underlying the increased ability to induce myeloid differentiation shown by CM-444 and CM-1758”. This conclusion is somewhat misleading, since presumably, all

inhibitors induce hyperacetylation of histones to a similar level but show differences in non-protein acetylation. Please rephrase.

As suggested by the reviewer we have rephrased this sentence.

Results section, page 9: “These results suggest that CM-444 or CM-1758 could modulate specifically non-histone protein acetylation patterns in AML cells, in addition to histone protein acetylation. This differential effect could be key to discover the molecular mechanism underlying the increased ability to induce myeloid differentiation shown by CM-444 and CM-1758”.

13) Focusing on non-histone proteins in their further analysis, the authors identify 87 proteins that are specifically regulated following treatment with CM-444 and CM-1758. They suggest that the BRD proteins might be essential for the differentiation of AML cells following treatment. Their conclusion is based on using BRD inhibitors in combination with their compounds, which represses differentiation. Based on their illustration, BRD proteins are also targeted by other HDACi. It is unclear whether their compounds target different lysines. Again, the authors need to show the specific sites.

We agree with the reviewer that the information regarding the specific lysines is lacking and not explained in detail. As pointed out by the reviewer, we detected several BRD proteins with K sites differentially acetylated also by other HDACi, specifically by Panobinostat (BRD1 K407, BRD K519 or BRD4 K317). However, there are other K sites in BRD1 and BRD4 which are only differentially acetylated by both CM-444 and CM-1758 (BRD1 K516 and BRD4 K1177) or by one of them (BRD1 K413, BRD1 K418, BRD2 K31, BRD4 K1177 and BRD8 K174). This is summarized in Reviewer Figure 19.

Reviewer Figure 19. Acetyl-K sites of BRDs proteins deregulated by HDACi.

We have modified the revised version the manuscript, specifying the specific K-sites of BRDs, as well as in the rest of non-histone proteins participating in the enhancer–promoter chromatin regulatory complex, which were deregulated after treatments in Reviewer Figure 20. Reviewer Figure 20 is included as Figure 5C in the revised version of the manuscript.

Reviewer Figure 20. Representation of the protein complexes highly acetylated by CM-444 and CM-1758 and other HDACi. The specific K-sites regulated in each protein are specified.

14) In order to confirm the essential function of BRD proteins in the differentiation process, the authors should perform further mechanistical studies including site directed mutagenesis. What is the function of BRD acetylation? Changes in their binding affinity to acetyl-lysine? Changes in overall stability or localization? Are there differences in binding on the respective promoters/enhancers of genes induced upon differentiation?

This is indeed an interesting issue raised by the reviewer. Thus, we thought in carrying out a site directed mutagenesis of the acetyl-K sites of BRD4 regulated by CM-444 and CM-1758 in order to try to elucidate the role of these acetylations in the differentiation induced by our compounds. Unfortunately, the plasmid containing BRD4 (from Addgene) has not been provided in all this time by the supplier and we were not able to carry out this experiment.

Nevertheless, to address the mechanisms of action of CM-444 and CM-1758 we performed CUT&RUN experiments against BRD4, H3K27Ac and H3K9Ac after treatment for 12h with CM-444 in HL-60 cells. We detected 3451, 15101 and 13454 peaks for BRD4, H3K27Ac and H3K9Ac respectively in the control samples (HL-60 before treatment) that were located predominantly in promoter regions (Reviewer Figure 1). After CM-444 treatment, the number of peaks detected were 19570, 48991 and 7568. Interestingly, we observed a swift in the location of the peaks that move to active intronic and distal intergenic regions instead of promoter regions for BRD4, H3K27Ac and H3K9Ac. Moreover, CM-444 treatment drove BRD4, H3K27Ac and H3K9Ac to genes related to myeloid differentiation such as *CB11b*, *AB11*, *GATA2* and *GF11* (Reviewer Figure 1C). These results suggest that both histone and non-histone protein acetylation are important for the induction of myeloid differentiation by our epigenetic compounds CM-444 and CM-1758. The acetylation induced by our compounds in BRD4, as well as H3K27Ac and H3K9Ac histone marks, induced a swift towards regulatory regions of genes involved in differentiation. This is consistent with the effect of BET inhibitors Molibresib and JQ1 in AML cell lines, where we observed that inhibition of BRD abrogates the differentiation of AML cells induced by CM-444 and CM-1758 (Reviewer Figure 1).

To further differentiate the mechanism of action of our compounds with the reference DACi we performed CUT&RUN experiments in HL-60 cells using Panobinostat. Treatment with Panobinostat was associated with 16030, 26959 and 32775 peaks for BRD4, H3K27Ac and

H3K9Ac respectively. As shown in Reviewer Figure 1 there was some overlap in the peaks between Panobinostat and our compounds. Interestingly, we observed some changes in the location of peaks in BRD4 and H3K9Ac with an increase in active regions similar to what was observed after treatment with CM-444 (although to a significantly lesser extent) while there were hardly any differences in the localization of the peaks in H3K27Ac after Panobinostat treatment compared to the control (Reviewer Figure 1B). These changes are exemplified in the differences observed in genes involved in AML cells differentiation (Reviewer Figure 1C).

These results have been included in the revised version of the manuscript (Results section, page 10) and the Reviewer Figure 1 has been included as Figure 6.

Results section, page 10: *“To delve into the role that BRDs could play in the differentiation process induced by our compounds but not by other DACi, we performed CUT&RUN experiments with BRD4 and the specific histone marks H3K9Ac and H3K27Ac in HL-60 after treatment with CM-444 or Panobinostat for 12 hours. The numbers of peaks of the different marks are shown in Figure 6A. Interestingly, CM-444 induced changes in the location of peaks associated with BRD4, H3K27Ac and H3K9Ac. While BRD4, H3K27Ac and H3K9Ac peaks were predominantly located in promoters in untreated cells, after treatment with CM-444 the majority of the peaks were located in intronic and distal intergenic regions (Figure 6B). Moreover, CM-444 treatment drove BRD4, H3K9Ac and H3K27Ac to genes related to myeloid differentiation and we observed that BRD4 peaks after CM-444 treatment were also associated with active chromatin regions as demonstrated by the increase in H3K27Ac mark (Figure 6C). Regarding the global changes carried out by Panobinostat, it should be highlighted that they were much lower than with CM-444 (Figure 6B). Regarding H3K27Ac, there were hardly any differences in the localization of the peaks after Panobinostat treatment compared to the control (Figure 6B). This is reflected in the different peaks observed in genes involved in AML cells differentiation (Figure 6C), where only minor changes were observed in comparison with untreated cells. All these results demonstrated that our compounds modulate the localization of BRDs, as well as acetyl histone marks, in a different way than Panobinostat. This could explain the different potential to induce differentiation of AML of our compounds in comparison with other DACi”.*

Reviewer Figure 1: Role of BRDs in the differentiation induction in AML with CM-444 and CM-1758. **A**) BRD4 (left), H3K27Ac (middle) and H3K9Ac (right) peak numbers in HL-60 cells treated with CM-444 or Panobinostat for 12 hours. **B**) CUT&RUN peaks distribution of BRD4 (left), H3K27Ac (middle) and H3K9Ac (right) in HL-60 cells treated with CM-444 or Panobinostat for 12 hours. **C**) Examples of myeloid genes involved in differentiation of AML cells. **D–E**) Cell differentiation assay measuring CD11b by flow cytometry after treating **D**) HL-60 and **E**) ML-2 cells daily with 25% GI_{50} of CM-444, CM-1758, Molibresib, JQ1, and the combination of CM-444 or CM-1758 with Molibresib or JQ1 for 48 h. Error bars indicate the S.D. of three replicates. Statistical significance was calculated by a two-tailed Student's *t*-test. n.s. = non-significant; * $p \leq 0.05$. **F–G**) q-PCR of *GATA2*, *PU.1*, *SCL*, and *CEBPA* after treating **F**) HL-60 and **G**) ML-2 cells daily with 25% GI_{50} of CM-444, CM-1758,

Molibresib, JQ1 and the combination of CM-444 or CM-1758 with Molibresib or JQ1 for 48 h. Error bars indicate the S.D. of three replicates. Statistical significance was calculated by a two-tailed Student's *t*-test. n.s. = non-significant; * $p \leq 0.05$.

15) Did the authors detect changes in MYC acetylation after treatment?

This is indeed an interesting issue raised by the reviewer. We have checked the acetylome results and a down-regulation of MYC acetylation in the residue K148 was detected after treatment not only CM-444 and CM-1758, but also with the commercial HDACi Panobinostat and Vorinostat. The fact that this change is produced with all compounds is consistent with the downregulation of *MYC* observed in AML cells treated with CM-444, CM-1758, Panobinostat and Vorinostat (Reviewer Figure 21). Actually, MYC acetylation in the residue K148 by the acetyltransferase EP300 has been related with MYC stabilization (Lynch J.T. *et al.*, *Cell Death Dis*, 2013; Faiola F, *Mol Cell Biol*, 2005). In this sense, the inhibition of MYC K148 acetylation after treatment with DACi could be directly involved in the global downregulation of MYC levels.

Reviewer Figure 21. MYC expression measure by q-PCR after treatment ML-2 cells daily with 25% GI₅₀ of CM-444, CM-1758, Panobinostat and Vorinostat for 48h.

16) HDAC inhibitors induce DNA damage. Was this observed also for their compounds? The authors should analyze this e.g by gH2AX immunofluorescence staining.

Following the reviewer's suggestion, we have performed gH2Ax immunofluorescence in AML cell lines treated daily with our compounds CM-444 and CM-1758 for 48 hours. As shown in Reviewer Figure 22, the levels of γ -H2AX were not increased after treatment with our compounds in any of the four AML cell lines tested (HL-60, ML-2, MOLM-13 and MV4-11). As positive control, HL-60 cells irradiated with an acute dose of 30 J/m² UV for 2 minutes were used. As expected, one hour after UV irradiation, an increased in γ -H2AX was clearly observed (see Reviewer Figure 22). In order to clarify these results, we quantified the number of foci per cell, as shown in the right panel of the figure. With these results, we can conclude that CM-444 or CM-1758 do not induced DNA damage in AML cell lines, at least at the doses and times tested.

Reviewer Figure 22. CM-444 and CM-1758 did not induce DNA damage in AML cell lines. γ -H2Ax immunofluorescence after treatment with our compounds CM-444 and CM-1758 for 48 hours in **A)** HL-60, **B)** ML-2, **C)** MOLM-13 and **D)** MV4-11 AML cell lines (left panel). Nuclear DNA was counterstained with DAPI. In the right panel the quantitation of γ -H2Ax staining pattern is shown.

Minor comments:

Fig. 1A: some of the circles are cut off.

We apologize for this error, which has been corrected in the revised version of the manuscript.

Page 8, last paragraph, typo: “Kmeans clustering split the network of the 85 specific Acetyl-K regulated proteins ...”. It should be 87.

We apologize for this error, which has been corrected in the revised version of the manuscript.

Reviewer #3 (Remarks to the Author):

The authors propose two molecules for a “Novel epigenetic based differentiation therapy for Acute Myeloid Leukemia”. These two molecules feature the same warhead as Quisinostat.

Overall the manuscript gathers a wide-range of assays from docking to in-vivo evaluation. The authors describe these molecules as pan-HDAC inhibitors, which in contrast to the approved ones induce differentiation (CD11B marker) without excessive apoptosis (annexin-V). Whereas the pre-treatment of cells before injection in the murine model is convincing, the efficacy of the molecules for AML without pre-treating the cells is not as obvious. Interestingly, BRD inhibitors prevented the differentiation induced by the two inhibitors. A large part of the manuscript is devoted to acetylomics to try and understand the molecular cause of the observed differences. Whereas the approach and the molecules are interesting, I deem this manuscript unsuitable for publication for the following two major reasons:

1) The acetylomics study is conducted with a single replicate of cellular treatment per drug (two technical replicates of the IP of the same pellet and 2 MS injections of these IPs) rendering all the interpretation dubious.

As rightly pointed out by the reviewer, the experiment aimed to identify changes in the acetylome was performed as a biological single replicate. Although in general as scientist we are accustomed to include multiple replicates including several samples in our experimental design, the situation with acetylome studies is to some extent different for a number of reasons including the technical complexity. To ensure a successful detection of acetylated proteins we need a minimum of 600 million of cells by condition. Therefore, 3000 million of cells were needed to carry out the acetylomics study presented in the manuscript. The generation and management of this amount of cells cannot be underscored and was the reason for the current design including one biological replicate and 2 IPs for each extract. Each IP was analyzed twice by LC-MS/MS. Despite using a biological single replicate, this technique is a robust one as has been previously demonstrated (*Martinez-Val A. et. al., J Proteome Res, 2017*).

Based on the issue raised by the reviewer, an additional acetylome experiment was performed using HL-60 cells treated with CM-444 and CM-1758 to validate the acetylome results included in the manuscript. As shown in Reviewer Figure 23 (panel A and B), both HL-60 and ML-2 untreated control cells clustered together and separately from cells treated with CM-444 and CM-1758 which also clustered together, independently of the cell line. In addition a degree of overlapping of acetyl-K sites regulated by CM-444 or CM-1758 between HL-60 and ML-2 was observed (Reviewer Figure 23C and D), providing a good correlation between results derived from both cell lines. This reflects the robustness of our acetylome analysis and supports the results obtained, despite being a single replicate.

Reviewer Figure 23. Acetylome study in HL-60 and ML-2 cells treated with CM-444 and CM-1758. A) PCA of acetylome data from HL-60 and ML-2 cells after treatment with CM-444 or CM-1758 compared with untreated cells. B) Unsupervised hierarchical cluster of identified Acetyl-K sites from ML-2 and HL-60 cells after treatment with CM-444 or CM-1758 compared with untreated cells. C) Venn diagram of the total number of regulated Acetyl-K sites in HL-60 and ML-2 cells after treatment with CM-444 or CM-1758 compared with untreated cells. D) Comparison of Acetyl-K sites regulated by CM-444 (upper panel) or CM-1758 (lower panel) between HL-60 and ML-2 cell lines.

2) The lack of consideration for medicinal chemistry is disconcerting. Whereas the two proposed molecules are obvious Quisinostat analogues, this molecule only features in one assay. It should be a control in most, with a fair determination of its optimal concentration. The 2 molecules are very close analogues, CM-444 being initially presented as a DNMTs/HDACs inhibitor while CM-1758 is presented as a HDACs inhibitor (fig1A). Yet no SAR discussion about this difference is to be found. The panHDAC nature of these molecules is also to be questioned, where the assays stem from Eurofins. Recent papers indeed cast doubt on the usually used enzymatic assays for HDACs and this would at least need commenting. Because of the similarity to Quisinostat, the assays of the two new molecules should be complemented by the same assay for Quisinostat and compared to recent publications that outline major issues notably for HDAC11 and class IIa HDACs. E.g: <https://doi.org/10.1021/acsomega.9b02808> <https://doi.org/10.1038/s41589-022-01015-5> <https://doi.org/10.3390/ijms24054720>

The reviewer concerns can be divided in 3 main points: a) Lack of SAR discussion; b) Use of unappropriated HDACi as controls; and c) Unclear Pan-HDAC nature of CM-444 and CM-1758 compounds.

a) Lack of SAR discussion: As described in our manuscript, the study was based on the analysis of 41 proprietary epigenetic inhibitors, including CM-444 and CM-1758, previously described and published which included various chemical series. The full description of the synthesis and SAR exploration of these compounds, including CM-444 and CM-1758, is summarized in Rabal O. et al., *J Med Chem*, 2021 (reference 27 in the manuscript), being the compound 12a and 13e CM-444 and CM-1758, respectively. More information regarding these compounds is also reported in our patent (Agirre X. et al.,

Novel compounds for use in cancer, WO2018229139A1, 13 June 2018). We apologize for not providing this information in a clear way in the original version of our manuscript. The revised version of the manuscript has been modified to include the information:

Results section, page 5: “Design, synthesis and SAR exploration of both compounds are described in detail by Rabal O. and colleagues²⁷ and in our patent WO2018229139A1”.

- b) Unappropriated HDACi used as controls: the reason to use Vorinostat and Panobinostat as reference HDACi to compare with CM-444 and CM-1758 is that FDA and EMA have approved both compounds. This is a fundamental point as our objective is the development of new epigenetic drugs and the comparison with current clinically applied drugs is warranted. However, as described in the manuscript the potential for myeloid differentiation of additional HDACi such as Entinostat, Quisinostat and Tubastatin was extensively tested in 4 different AML cell lines. These experiments showed higher capacity of inducing myeloid cell differentiation of our compounds with respect the commercially available HDACi (Reviewer Figure 24 and Reviewer Figure 25). Based on these results we decided to focus on the approved HDACi for the mechanistical studies, which would be more meaningful from the clinical stand point. Reviewer Figure 24 and 25 are included in the revised version of the manuscript as Figure 4A and Figure S5A. Considering these arguments we believe that the comparison of CM-444 and CM-1758 with Vorinostat and Panobinostat is appropriate.

Reviewer Figure 24. Cell differentiation assay in ML-2 cell line. CD11b and Sytox Green were measured by flow cytometry in an ML-2 cell line treated with 25% GI₅₀ of CM-444 (260 nM), CM-1758 (210 nM), and the commercial HDACi Panobinostat (12.9 nM), Vorinostat (1.1 μM), entinostat (2.3 μM), quisinostat (18 nM), and tubastatin (2.5 μM) for 48 h.

Reviewer Figure 25. Cell differentiation assay in HL-60, MOLM-13 and MV4-11 cell lines. CD11b and Sytox Green were measured by flow cytometry in HL-60, MOLM-13 and MV4-11 cells treated with 25% GI₅₀ of CM-444, CM-1758, and the commercial HDACi, Panobinostat, Vorinostat, entinostat, quisinostat, or tubastatin for 48 h.

- c) Unclear Pan-HDAC nature of CM-444 and CM-1758 compounds: to analyze the HDAC activity of our compounds, we used fluorescence intensity enzymatic assays for HDACs, which are widely accepted by the scientific community, such as those performed by Eurofins or BPS Bioscience. There are several recent studies where the same BPS or Eurofins's assays were used: Bouchet S. et. al., ACS Med Chem Lett, 2019; Beshore D.C. et. al., ACS Med Chem Lett, 2021; Chang T.Y. et. al., Biomed Pharmacother, 2021;

Horndahl, J. et al, PLoS One, 2022; Cuadrado-Tejedor M. et al, Neuropsychopharmacol, 2017; She A. et al., Cell Chemical Biology, 2017; Lee J.H. et al., PNAS, 2015; Serebryanny L.A. et al., Scientific Reports, 2016. Nevertheless and following the reviewer's suggestion, we have determined the HDAC1, HDAC2, HDAC3 and HDAC6 IC₅₀ values of Quisinostat, Vorinostat and Panobinostat utilizing the same assay used with CM-444 and CM-1758. As shown in the Reviewer Table 4, values for the five compounds are similar, not finding any surprising or unexpected data.

Reviewer Table 4. HDAC1, HDAC2, HDAC3 and HDAC6 IC₅₀ values for Quisinostat, CM-444 and CM-1758.

	Quisinostat	CM-444	CM-1758	Vorinostat	Panobinostat
HDAC1	1.17 nM	6.55 nM	4.3 nM	6.61 nM	1.38 nM
HDAC2	11.55 nM	51 nM	31 nM	46.77 nM	N.D.
HDAC3	6.21 nM	18 nM	14 nM	8.13 nM	N.D.
HDAC6	217 nM	531 nM	257 nM	5.89 nM	12.59 nM

In addition, and following the suggestion from the reviewer we review the suggested publications and contacted the groups that had developed alternative methods for HDAC activity determination. While the group of Guillaume Médard's was unable to performed quantitative chemical proteomics assay using immobilized HDACi and mass spectrometry due to personal reasons, the group of Mike Schutkowski and Cyril Barinka, who has developed an alternative HDAC11 assay, agree to perform the HDAC11 alternative assay. As shown in Reviewer Table 5, beside our compounds CM-444 and CM-1758, Quisinostat, Vorinostat and Panobinostat were also tested. The IC₅₀ for CM-444 and CM-1758 in these studies showed some minor differences in comparison with the results obtained with Eurofins's assay even though the IC₅₀ continue to be in the nanomolar range. These results clearly support the pan-HDAC inhibitory activity of CM-444 and CM-1758.

Reviewer Table 5. HDAC11 IC₅₀ values for CM-444, CM-1758, Quisinostat, Vorinostat and Panobinostat using a continuous activity assay described by Mike Schutkowski and Cyril Barinka's groups.

	Continuous Activity Assay for HDAC11	Eurofins assay
CM-444	567.5 ± 133.05 nM	140 nM
CM-1758	599.8 ± 133.90 nM	150 nM
Quisinostat	123.9 ± 9.32 nM	N.D.
Vorinostat	> 20 μM	N.D.
Panobinostat	438.53 ± 32.8 nM	N.D.

As we did not perform Eurofin assays with commercial HDACi we can not compare the results. However, we observed an HDAC11 IC₅₀ > 20 μM activity for Vorinostat, being the IC₅₀ values for Quisinostat and Panobinostat in nanomolar range too. Further exploration of this finding is beyond the scope of our manuscript. According to the reviewer comment, the HDAC11 IC₅₀ values determined by the new assay have been included in the revised version of the manuscript as can be seen in Reviewer Figure 26 (Results section, page 5, Materials and Methods section, page 29 and in the Supplemental information, page 17 (Table S2). Reviewer Figure 26 has been included in the revised version of the manuscript as Figure 1B.

Reviewer Figure 26. Percentage of inhibition of CM-444 and CM-1758 at 10 μ M against a panel of 95 epigenetic targets. HDACs, DNMTs, and UTX IC₅₀ values are indicated.

Results section, page 5: "Specifically, we found that CM-444 and CM-1758 had IC₅₀ values against HDAC1 (HDAC family-I) of 6.55 and 4.3 nM, against HDAC7 (HDAC family-IIA) of 120 nM, against HDAC10 (HDAC family-IIB) of 15 and 29 nM and against HDAC11 activity of 567.5 and 599.8 nM (HDAC family-IV), respectively".

Reviewers' Comments:

Reviewer #1:

Remarks to the Author:

I commend the authors for their thorough responses. They have addressed my previous comments and I am pleased to be able to recommend acceptance of their revised manuscript.

Reviewer #2:

Remarks to the Author:

The authors put a lot of effort to revise their manuscript and addressed all my previous comments in large detail. Overall, I am satisfied with their revised version, however a couple of concerns remain.

1) The quality of the RNA-seq data appears unsatisfactorily. The PCA plots (Figure S2C) show the variance of the individual samples. Especially in the HL-60 cells, the CM-444 treatment shows a large variance, with one sample clustering closer to the CM-1758 treated samples, and two samples clustering closer to the controls. This is problematic and also reflected by the lower number of differentially regulated genes in CM-444 treated cells. Similarly, control samples in both cell lines show a large heterogeneity, with one outlier in each experiment. Why are only two samples shown for ML-2 cells treated with CM-444? Further, the authors note, that FC of +/- 0,4 was selected as filtering criteria. To me, this is not a biologically relevant change in expression and I would suggest to filter at least for 1,5- or 2-fold changes. Presumably, no significant genes will then remain and I would thus suggest to omit this data from the manuscript. Further, hierarchical clustering (Figure S2D) should be applied to the individual samples, rather than differentially expressed genes.

2) For the cut&run experiments it would be interesting to show, whether distal intergenic and intronic peaks after CM-444 treatment (Figure 6B) overlap with known enhancers or other regulatory elements. Additionally, the browser tracks (Figure 6C) should indicate the genomic coordinates (e.g size of the displayed regions). The peaks appear somewhat noisy and some of the data are not convincing (e.g for GATA2 and GF11).

Minor comments:

Line 466: it should be Figure 6E,D

Line 472: it should be Figure 6F,G

Reviewer #3:

Remarks to the Author:

I thank the authors for their additional work, which however does not fully address my concerns.

The clear reference to the medicinal chemistry paper is certainly helping, and the additionally testings on HDAC11 is a worthy addition. Class IIa inhibition remains to be taken with caution, though.

The additional comparison to Quisinostat is more than welcome in the CD11 % assay showing that the differentiation is higher at 25% GI50.

However most mechanistic conclusion of the paper is still based on an acetylomics study performed on a single treatment experiment in the cell line. The addition of another cell line acetylomics experiment is asking the question whether the same MoA is at play in a different cell line, not the question of the validity of the results.

In my view an experiment featuring triplication of control vs claimed DNMTi/DACi CM44 or pure DACi CM1758 is to be done, when the experiment is so central to a study. Should we observe a change between CM44 and CM1758 (the authors present CM44 as a dual inhibitor but fail to show us where

this is impactful. No change in methylation is for instance reported)? Would there even be one compared to Quisinostat? If not, the proposed mechanism would be void. This structure-activity relationships discussion towards the differences in acetylation is still missing. The robustness of the acetylomics workflow in the hands of the authors can only be assessed in the light of a statistical testing and not by referring to other papers.

I appreciate the amount of work that might represent. But in the absence of such replication, no conclusion should be drawn from this acetylomics experiment.

Response to the Reviewers' comments

We thank the reviewers for their thorough revision and suggestions on our revised manuscript as well as for their positive feedback. We have added new information to address the reviewer's comments, which we sincerely believe contribute to improve the quality of our manuscript.

Below you can find the detailed answers to all the issues raised by the reviewer #2 and #3.

Reviewer #1 (Remarks to the Author):

I commend the authors for their thorough responses. They have addressed my previous comments and I am pleased to be able to recommend acceptance of their revised manuscript.

We sincerely appreciate the reviewer's general comment.

Reviewer #2 (Remarks to the Author):

The authors put a lot of effort to revise their manuscript and addressed all my previous comments in large detail. Overall, I am satisfied with their revised version, however a couple of concerns remain.

We are sincerely grateful for the positive comments regarding our revised version of the manuscript, as well as for the reviewer suggestions.

Below, we provide a detailed response to his/her main concerns.

1) The quality of the RNA-seq data appears unsatisfactorily. The PCA plots (Figure S2C) show the variance of the individual samples. Especially in the HL-60 cells, the CM-444 treatment shows a large variance, with one sample clustering closer to the CM-1758 treated samples, and two samples clustering closer to the controls. This is problematic and also reflected by the lower number of differentially regulated genes in CM-444 treated cells. Similarly, control samples in both cell lines show a large heterogeneity, with one outlier in each experiment. Why are only two samples shown for ML-2 cells treated with CM-444? Further, the authors note, that FC of $\pm 0,4$ was selected as filtering criteria. To me, this is not a biologically relevant change in expression and I would suggest to filter at least for 1,5- or 2-fold changes. Presumably, no significant genes will then remain and I would thus suggest to omit this data from the manuscript. Further, hierarchical clustering (Figure S2D) should be applied to the individual samples, rather than differentially expressed genes.

We sincerely appreciate the comment from the reviewer. Indeed as described below we wrongly explain the results. In our answers below we have corrected these mistakes and actually argue that we would be inclined to maintain the results in the revised version of the manuscript. However, if the reviewer believes we should remove them we will be delighted to do it.

Regarding the quality of the RNA-seq data, all of our samples passed the quality controls except for one (ML-2 CM-444 replicate 1), which was excluded from the analysis due to its low read count. This explains why only two samples for ML-2 treated with CM-444 are observed in the PCA, as pointed out by the reviewer. Therefore, we believe that the remaining RNA-seq data are suitable for further analysis.

The variation we observed between samples is due to the fact that for the experiment using RNA-seq, we performed three strictly biological replicates. While there is variability between biological replicates of the two AML cell lines used, technically, we do not have any specific criteria for eliminating any of the samples in the study. Furthermore, in each of the biological replicates, in two AML cell lines, we observed a robust difference between the treated and untreated cells. Nevertheless, we are confident that our results demonstrate clear and significant differences between untreated and treated AML cells.

We would like also to clarify that the filtering criteria used for this RNA-seq experiment was $\log_2FC \pm 0.4$ rather than just FC, as we mistakenly indicated in the previous response to the reviewers. We apologize for this error. However, we agree with the reviewer that even though it is \log_2FC , it represents a relaxed criteria. Consequently, we have reanalyzed the data using an increased \log_2FC filter of ± 1 . Upon reanalysis, in the case of HL-60 cell line, we found 136 deregulated genes (3 down-regulated and 133 up-regulated) and 1278 deregulated genes (409 down-regulated and 869 up-regulated) in response to CM-444 or CM-1758 treatment,

respectively. In the ML-2 cell line, the number of deregulated genes by CM-444 or CM-1758 were 460 (44 down-regulated and 416 up-regulated) and 2520 (1183 down-regulated and 1337 up-regulated), respectively (**Reviewer Figure 1**). In our opinion, these results are important to show in the context of the study carried out, and therefore, we have maintained them in the manuscript.

Reviewer Figure 1: RNA-seq data of HL-60 and ML-2 treated with CM-444 or CM-1758. Venn diagram of differentially expressed genes after CM-444 or CM-1758 treatment in HL-60 and ML-2 cell lines.

With respect to the last comment, we would like to highlight that the hierarchical clustering, included in the previous version of the manuscript, showed the \log_2FC of all genes after treatment with CM-444 or CM-1758 compared with untreated cells in HL-60 and ML-2 cell lines, and not just the differentially expressed genes, as erroneously stated in the figure legend. We apologize for this mistake and since these results are similar to those shown in the PCA analysis (**Figure S2B**), we have decided to remove the heatmap figure in the new version of our manuscript.

In the revised version of the manuscript, all these modifications have been included in the results section (page 6). Reviewer Figure 1 has been included as Figure S2C.

Results section, page 6: “Principal component analysis (PCA) of RNA-seq data showed the differences between treatments (Figure S2B). After treatment, 136 (3 down-regulated and 133 up-regulated) and 1278 (409 down-regulated and 869 up-regulated) genes were deregulated by CM-444 and CM-1758, respectively in HL-60 cells. In the case of ML-2 cell line, the number of deregulated genes by CM-444 and CM-1758 were 460 (44 down-regulated and 416 up-regulated) and 2520 (1183 down-regulated and 1337 up-regulated), respectively (Figure S2C).”

2) For the cut&run experiments it would be interesting to show, whether distal intergenic and intronic peaks after CM-444 treatment (Figure 6B) overlap with known enhancers or other regulatory elements. Additionally, the browser tracks (Figure 6C) should indicate the genomic coordinates (e.g size of the displayed regions). The peaks appear somewhat noisy and some of the data are not convincing (e.g for GATA2 and GF11).

This is indeed an interesting observation raised by the reviewer. We have examined the distribution of chromatin state in the three conditions (Control, CM-444 treatment and Panobinostat treatment) by overlapping the identified regions of each sample of the study with the chromatin states of AML samples. Specifically, 12 chromatin states, based in the combination of 6 histone marks, for 38 AML samples were used (Yi G, *et al*, *Cell Rep.*, 2019). Consistent with our previous findings, this analysis showed that Cut&Run peaks following CM-444 and Panobinostat treatment exhibited a decrease in active promoters and transcription for BRD4 and the histone marks H3K27ac and H3K9ac (**Reviewer Figure 2**). However, these changes were less pronounced in the case of Panobinostat.

Reviewer Figure 2: Distribution of the different chromatin states in untreated cells and cells treated with CM-444 and Panobinostat for BRD4 and the histone marks H3K27ac and H3K9ac from Cut&Run peaks.

These results have been included in the revised version of the manuscript (Results section, page 10) and the Reviewer Figure 2 has been included as Figure S8.

Results section, page 10: “The analysis of the distribution of the chromatin states was consistent with all these findings (Figure S8).

As suggested by the reviewer, we have indicated the genomic coordinates and we have removed the *GATA2* and *GFII* data from the Figure 6C.

Minor comments:

Line 466: it should be Figure 6E,D

We apologize for this error, which has been corrected in the revised version of the manuscript.

Line 472: it should be Figure 6F,G

We apologize for this error, which has been corrected in the revised version of the manuscript.

Reviewer #3 (Remarks to the Author):

I thank the authors for their additional work, which however does not fully address my concerns.

The clear reference to the medicinal chemistry paper is certainly helping, and the additionally testings on HDAC11 is a worthy addition. Class IIa inhibition remains to be taken with caution, though.

The additional comparison to Quisinostat is more than welcome in the CD11 % assay showing that the differentiation is higher at 25% GI50.

However most mechanistic conclusion of the paper is still based on an acetylomics study performed on a single treatment experiment in the cell line. The addition of another cell line acetylomics experiment is asking the question whether the same MoA is at play in a different cell line, not the question of the validity of the results. In my view an experiment featuring triplication of control vs claimed DNMTi/DACi CM44 or pure DACi CM1758 is to be done, when the experiment is so central to a study. Should we observe a change between CM44 and CM1758 (the authors present CM44 as a dual inhibitor but fail to show us where this is impactful. No change in methylation is for instance reported)? Would there even be one compared to Quisinostat? If not, the proposed mechanism would be void. This structure-activity relationships discussion towards the differences in acetylation is still missing. The robustness of the acetylomics workflow in the hands of the authors can only be assessed in the light of a statistical testing and not by referring to other papers.

I appreciate the amount of work that might represent. But in the absence of such replication, no conclusion should be drawn from this acetylomics experiment.

As suggested by the reviewer, we have replicated the acetylome experiment. We believe it is crucial to highlight that the conclusions derived from this experiment remain consistent with those of the previous version.

Firstly, we would like to emphasize that the equipment utilized in the previous experiment is no longer available, and we have employed more sensitive equipment for this new experiment. Moreover, we have implemented a new technical approach for both the proteome and acetylome analyses. These adjustments have contributed to enhancing the robustness of our experiment. This new technology has been used to perform a new experiment, including four new replicates for the Control (Untreated cells) and cells treated with CM-444 or CM-1758, and three replicates for cells treated with Panobinostat or Vorinostat (**Reviewer Figure 3**). This thoughtful design ensures the reliability and accuracy of our findings. Consequently, we believe that our experiment is now more robust, notably improving the quality of our work but at the same time maintaining the conclusions of the study.

Reviewer Figure 3: Proteome and acetyloyme experimental design. ML-2 cells were treated with 25% GI_{50} of CM-444 (260 nM) or CM-1758 (210 nM) and with reference HDACi Panobinostat (12.9 nM) or Vorinostat (1.1 μ M) for 12 h. Then, cells were lysed and digested and the subsequent peptides were labeled and fractionated. 5 % of the fractions was kept for proteome analysis by LC-MS/MS. For the complete acetyloyme study, 95% of the obtained fractions were combined into five pools and three consecutive immunoprecipitations were performed for each pool, and the resulting samples were subsequently analyzed by LC-MS/MS.

We believe it is inappropriate to combine data from the previous experiment with the new one. Therefore, we have replaced all data and figures with the new data. With respect to proteome, 8342 proteins were quantified, being 9.5%, 8.8%, 5.8% and 2.7% significantly regulated after treatment with CM-444, CM-1758, Panobinostat or Vorinostat, respectively (**Reviewer Figure 4A**). The levels of 192 proteins were regulated by treatment with the four HDACi used. Additionally, it is important to highlight that CM-444 and CM-1758 shared approximately 80% of the regulated proteins. Vorinostat and Panobinostat showed a shared regulation in over 90% of the proteins, while Panobinostat shared around 40% of the regulated proteins with both CM-444 and CM-1758 (**Reviewer Figure 4B**).

A**B**
Reviewer Figure 4: Proteome analysis after CM-444 and CM-1758 treatment of AML cells. A) Total number of proteins quantified and the fraction proteins regulated by individual HDACi. The bar chart shows the percentage of upregulated proteins ($\log FC > 0.3$, $p < 0.01$, shown in red) and downregulated proteins ($\log FC < 0.3$, $p > 0.01$, shown in blue). **B)** Venn diagram of deregulated proteins in ML-2 cells after treatment with CM-444, CM-1758, Panobinostat, or Vorinostat compared with untreated cells.

With respect to acetylome analysis, 3618 Acetyl-K sites were quantified. The percentage of Acetyl-K sites regulated after CM-444, CM-1758, Panobinostat or Vorinostat treatment were 22.8%, 20.3%, 17.7%, and 14.1%, respectively. The fraction of upregulated and downregulated Acetyl-K sites in the treated cells was similar, with 47.9%, 46.3%, 49.6% and 54.4% of the acetylation sites upregulated by CM-444, CM-1758, Panobinostat or Vorinostat, respectively (**Reviewer Figure 5A**). Similar to what we observed previously, most proteins were found acetylated on a single K residue, whereas around 7-10% of the identified proteins were highly acetylated (modified at ≥ 3 Acetyl-K sites) (**Reviewer Figure 5B**). The four compounds shared 378 of the 1021 regulated Acetyl-K sites. However, each HDACi modulated a specific acetylation pattern, being the pattern of modulated Acetyl-K sites very similar between the treatment with CM-444 and CM-1758, with approximately 80% of the Acetyl-K sites commonly regulated (**Reviewer Figure 5C**). Principal component analysis (PCA) of acetylome data verified these observations, showing a clear separation of untreated cells from HDACi-treated cells. In addition, cells treated with CM-444 and CM-1758 clustered together and separately from Panobinostat- and above all, from Vorinostat-treated cells (**Reviewer Figure 5D**).

Reviewer Figure 5: Acetyloyme analysis revealed different acetylation profiles between CM-444/CM-1758 and the reference HDACi. **A)** The total number of Acetyl-K sites quantified and the fraction of Acetyl-K sites regulated by each HDACi are shown. The bar chart shows the percentage of upregulated sites ($\log_2 FC > 1$, $p < 0.05$, shown in red) and downregulated sites ($\log_2 FC < 1$, $p > 0.05$, shown in blue). **B)** Distribution of acetylated sites in the acetylated proteins. **C)** Venn diagram of the total number of regulated Acetyl-K sites in ML-2 cells after treatment with CM-444, CM-1758, Panobinostat, or Vorinostat compared with untreated cells. **D)** PCA of acetyloyme data from ML-2 cells after treatment with CM-444, CM-1758, Panobinostat, or Vorinostat compared with untreated cells.

Additionally, we have reanalyzed the acetylated sequences identified after HDACi treatment, examining the 7 residues flanking the acetylated lysines in both up- and downregulated Acetyl-K sites. The conclusion drawn from this analysis corresponds with that of the previous experiment, as we did not observe differences between our compounds and the commercial HDACi Panobinostat or Vorinostat (**Reviewer Figure 6**).

Reviewer Figure 6: Analysis of the acetylation motifs. Sequence motif surrounding acetylated lysines after treatment of AML cells with CM-444, CM-1758, Panobinostat, or Vorinostat. The logo was created using the icelogo software package. A cutoff value of $p < 0.01$ was used.

Everything described so far is the descriptive analysis of the new results from both the proteome and the acetylome. We believe it is crucial to highlight that despite the differences between the two experiments, the conclusions derived from this new experiment remain consistent with those of the previous version. This is highly significant because with the robustness of the new experiment, there is no doubt that the conclusions drawn are accurate and reliable. In this manner, we have verified the following observations and conclusions:

1. More than 90% of the Acetyl-K residues were detected in histone proteins. However, most of these sites were not modified after treatment with any of the HDACi. On the other hand, the majority of the Acetyl-K sites regulated after treatment with each of the four HDACi were found in non-histone proteins (**Reviewer Figure 7**). Hence, the modulation of non-histone protein acetylation patterns in AML cells by CM-444 and CM-1758, in addition to histone protein acetylation, could be crucial in uncovering the molecular mechanism behind the enhanced ability of CM-444 and CM-1758 to induce myeloid differentiation. This justifies our decision to further study the regulated non-histone proteins.

Reviewer Figure 7: Representation of Acetyl-K sites regulated after HDACi in histone and non-histone proteins. The data are shown as percentages, and the number of Acetyl-K sites regulated by each HDACi is indicated in the correspondent bar.

- After analyzing the Acetyl-K sites differentially regulated by CM-444 and CM-1758 but not by Panobinostat and/or Vorinostat, we found one hundred four non-histone Acetyl-K sites specific to our compounds (**Reviewer Figure 8**).

Reviewer Figure 8: Acetyl-K sites specifically deregulated by CM-444 and CM-1758.

- Similarly to the previous experiment, k-means clustering of the 104 specific Acetyl-K regulated proteins using STRING revealed three clusters (**Reviewer Figure 9**).

Reviewer Figure 9: STRING protein-protein interaction analysis on the 104 Acetyl-K sites differentially regulated specifically by CM-444 and CM-1758. Three different functional clusters were detected.

One of these clusters, labeled Cluster 2, was again related to nucleic acid metabolism processes and gene expression, predominantly linked to nucleic acid binding (Reviewer Figure 10A and B). The larger cluster (Cluster 3), was associated with histone modifications, DNA repair and notably, myeloid differentiation. Indeed, several transcription factor related to myeloid differentiation, such as CEBPA, EP300 and PML, were specifically acetylated by our compounds. Moreover, the molecular function was associated with histone, chromatin, transcription factors, and histone acetyltransferase activity (Reviewer Figure 10C and D).

Reviewer Figure 10: Gene ontology (GO) analysis of the three functionally different clusters generated from the Acetyl-K sites specifically deregulated by CM-444 and CM-1758. A) Biological process GO results of cluster 2. B) Molecular function GO results of cluster 2. C) Biological process GO results of cluster 3. D) Molecular function GO results of cluster 3.

It is interesting to note that, similarly to the previous experiment, many of these proteins of Cluster 3 are proteins that participate in the enhancer–promoter chromatin regulatory complex, and some have been shown to have important roles in AML, such as proteins containing bromodomains (BRDs) and those in the cohesin complex, mediator complex, or MOZ/MORF complex, among others (**Reviewer Figure 11**). These results once again provide us with the basis to develop our hypothesis regarding the potential crucial role that BRDs could play in the differentiation mechanism of our epigenetic compounds, thus justifying the final part of our study.

Reviewer Figure 11: Representation of the protein complexes highly acetylated by CM-444 and CM-1758 and other HDACi. The specific K-sites regulated in each protein are specified.

In conclusion, we have replicated the acetylome experiment, obtaining much more robust results. Furthermore, we have validated all the conclusions drawn from the previous acetylome experiment, without altering the message of our study. Finally, the new mass spectrometry proteomics data have been deposited in the ProteomeXchange Consortium via the PRIDE partner repository with the data set identifier PXD050623.

In the revised version of the manuscript, changes have been included in the results section (pages 8-11) and in the materials and methods section (pages 38-41). Reviewer Figure 3 has been included as Figure 4B, Reviewer Figure 4 as Figure S5B-C, Reviewer Figure 5 as Figure 4C-F, Reviewer Figure 6 as Figure S6, Reviewer Figure 7 as Figure 4G, Reviewer Figure 8 as Figure 5A, Reviewer Figure 9 as Figure 5B, Reviewer Figure 10 as Figure S7 and Reviewer Figure 11 as Figure 5C.

Results section, page 8: “Firstly, after cells were lysed and digested and the subsequent peptides were labeled and fractionated, 5 % of the fractions was kept for proteome analysis by liquid chromatography - tandem mass spectrometry (LC-MS/MS) (Figure 4B). We were able to quantify a total of 8342 proteins after treatment with any of the HDACi. Among them, 9.5%, 8.8%, 5.8% and 2.7% were significantly regulated after treatment with CM-444, CM-1758, Panobinostat or Vorinostat, respectively (Figure S5B). Only the levels of 192 proteins were regulated by treatment with the four HDACi used (Figure S5C). Additionally, it is important to highlight that CM-444 and CM-1758 shared approximately 80% of the regulated proteins. Vorinostat and Panobinostat showed a shared regulation in over 90% of the proteins, while Panobinostat shared around 40% of the regulated proteins with both CM-444 and CM-1758.”

Results section, pages 8-9: “For the complete acetylome study, 95% of the obtained fractions were combined into five pools for acetyl-lysine (Acetyl-K) enrichment using an anti-acetyl-lysine antibody. Particularly, three consecutive immune-precipitations (IPs) were performed for each pool, and the resulting samples were subsequently analyzed by LC-MS/MS (Figure 4B). Specifically, we quantified 3618 Acetyl-K sites for each of the experiments carried out with each of the four HDACi used (Figure 4C). Most proteins were found acetylated on a single K residue, whereas around 7-10% of the identified proteins were highly acetylated (modified at ≥ 3 Acetyl-K sites) (Figure 4D). The percentage of Acetyl-K sites regulated after CM-444, CM-1758, Panobinostat or Vorinostat treatment were 22.8%, 20.3%, 17.7%, and 14.1%, respectively. The fraction of upregulated and downregulated Acetyl-K sites in the treated cells was similar, with 47.9%, 46.3%, 49.6% and 54.4% of the acetylation sites upregulated by CM-444, CM-1758, Panobinostat or Vorinostat, respectively (Figure 4C). Despite all four compounds shared 378 of the 1021 regulated Acetyl-K sites, each HDACi modulated a specific acetylation pattern. Interestingly, the pattern of modulated Acetyl-K sites was very similar between the treatment with CM-444 and CM-1758, with approximately 80% of the Acetyl-K sites commonly regulated (Figure 4E). Principal component analysis (PCA) of acetylome data verified these observations, showing a clear separation of untreated cells from HDACi-treated cells. In addition, cells treated with CM-444 and CM-1758 clustered together and separately from Panobinostat- and above all, from Vorinostat-treated cells (Figures 4F).”

Results section, page 9: “Interestingly, more than 90% of the Acetyl-K residues were detected in histone proteins. However, most of these sites were not modified after treatment with any of the HDACi. On the other hand, the majority of the Acetyl-K sites regulated after treatment with each of the four HDACi were found in non-histone proteins (Figure 4G).”

Results section, pages 9-10: “One hundred four non-histone Acetyl-k proteins were specific to our epigenetic compounds CM-444 and CM-1758, as shown in Figure 5A. We next generated a protein-protein interaction network using STRING with the application of K-means clustering on the main networks. K-means clustering split the network of the 104 specific Acetyl-K regulated proteins into three clusters: Cluster 1 was the smallest subnetwork of hubs (7 proteins, blue color); Cluster 2 was a subnetwork with more hubs (27 proteins, green color), and cluster 3 was the largest subnetwork (56 proteins, red color) (Figure 5B). Cluster 1 showed no significant biological processes or molecular functions, as the number of edges was very limited. Cluster 2 was enriched with nucleic acid metabolism processes and gene expression (Figure S7A) and was mainly associated with nucleic acid binding (Figure S7B). Finally, several enriched biological processes in cluster 3 were related to histone modifications, DNA repair and notably, myeloid differentiation (Figure S7C). In fact, several transcription factor related to myeloid differentiation were specifically acetylated by our compounds such as CEBPA, EP300 and PML. The molecular function was associated mainly with histone, chromatin, transcription factors, and histone acetyltransferase activity (Figure S7D). Of note, a large number of the cluster-3 proteins were related to DNA repair, bromodomains, or were members of the MOZ/MORF, mediator, SWI/SNF chromatin remodeling or histone acetyltransferase complexes. A remarkable finding was that most of the cluster-3 proteins whose acetylation was specifically modulated by CM-444 and CM-1758 treatment were proteins that participate in the enhancer-promoter chromatin regulatory complex, and some have been shown to have important roles in AML, such as proteins containing bromodomains 29 and those in the cohesin complex 30, mediator complex 31, or MOZ/MORF complex 32, among others (Figure 5C).”

Regarding the nature of our compounds, we biochemically describe CM-444 as a dual HDAC-DNMTs inhibitor, while CM-1758 exhibits only activity against HDACs. However, at the concentrations used in all our experiments, they are well below the IC_{50} for the different DNMTs. Therefore, under these circumstances, we consider that CM-444 behaves as an HDACi. In fact, we have demonstrated that, both short and long-term treatments, do not produce changes in global DNA methylation levels with any of our compounds. All of this is detailed in the Reviewer Figure 12.

Reviewer Figure 12: Treatment of AML cells with CM-444 and CM-1758, selective pan-HDACi at low non-cytotoxic doses. A) DNA methylation of LINE-1 analyzed by pyrosequencing after daily

treatment in HL-60 cell line with 270 nM CM-444 or 300 nM CM-1758 for 48 h. The DNA methylation percentage is indicated inside the circles. As a DNA methylated control, a universally methylated DNA was used. The data shown are the mean of two independent experiments. **B)** Dot blot was used to detect global 5-methylcytosine levels after CM-444 and CM-1758 daily treatment for 48 hours in an HL-60 cell line (270 nM and 300 nM, respectively). Methylene blue staining was used as a loading control. **C)** ML-2, MV4-11, and MOLM-13 cell lines were treated daily for 48 hours with 260, 160, and 280 nM of CM-444 and 210, 80, and 140 nM of CM-1758, respectively. DNA methylation analysis by pyrosequencing of LINE-1 was performed after treatment. Universally methylated DNA was used as DNA methylated control. The DNA methylation percentage is indicated inside the circles. The data shown the mean of two independent experiments. **D)** Dot blot was used to detect 5-methylcytosine levels after treatment. Methylene blue staining was used as a loading control. The experiment was repeated twice with similar results. 5 mC: 5-methylcytosine; MB: methylene blue. **E)** DNA methylation of LINE-1 analyzed by pyrosequencing after daily treatment with CM-444 or CM-1758 in HL-60 and ML-2 cell lines for 10 days or MOLM-13 and MV4-11 cells for 5 days. The DNA methylation percentage is indicated inside the circles. Treatment with decitabine was used as a positive control of DNA demethylation. As a DNA methylated control, a universally methylated DNA was used. The data shown are the mean of two independent experiments.

These results are included in the revised version of the manuscript in the results section (pages 5-6) and the Reviewer Figure 12 has been included as Figure 1E-F and Figure S1B-D.

Results section, pages 5-6: “Considering that the doses used for differentiation assays in this study were in the low nM range, we considered both compounds to be novel pan-HDACi, at least at the doses tested. We verified this by measuring histone 3 acetylation (H3Ac) and H3K27me3 levels by western blot and the 5 methylcytosine (5mC) levels by dot blot and performing LINE-1 pyrosequencing after treatment of HL-60 cells with CM-444 and CM-1758. We only detected a significant increase in H3Ac, with no changes in H3K27me3 or 5mC or LINE-1 DNA methylation levels (Figures 1D-F). We validated these results in three different AML cell lines, which showed that our lead compounds only induced an increase in H3Ac (Figure S1A-C). To definitely demonstrate that our compounds do not induce changes in DNA methylation, we treated AML cell at long-term (10 days for HL-60 and ML-2 and 5 days for MOLM-13 and MV4-11) with CM-444 and CM-1758 and then analyzed DNA methylation levels of LINE-1 (Figure S1D). In summary, these results confirmed that CM-444 and CM-1758 were novel and potent pan-HDACi compounds with a high capacity to promote myeloid differentiation in AML cell lines at low non-cytotoxic doses.”

Reviewers' Comments:

Reviewer #2:

Remarks to the Author:

The authors addressed my comments satisfactorily and I recommend acceptance of the revised manuscript.

I agree to keep the RNA-seq data as presented in their latest version.

Minor comments:

FigS2A: the colors in the legend and graph are not matching for CM-1758

Page 6, 3rd paragraph: in the sentence "Principal component analysis (PCA) and unsupervised hierarchical clustering analysis of RNA-seq data showed the differences between treatments (Figure S2B)." delete "unsupervised hierarchical clustering" or add "data not shown" since the figure was omitted in the latest version of the manuscript.

Reviewer #3:

Remarks to the Author:

I would like to thank the authors for the confirmation of their results by robust additional experiments and wish them widespread impact after publication of their findings. May their new compound help patients in the near future!

Response to the Reviewers' comments

We thank the reviewers for their thorough revision and suggestions on our revised manuscript as well as for their positive feedback.

Below you can find the detailed answers to all the issues raised by the reviewer #2.

Reviewer #2 (Remarks to the Author):

The authors addressed my comments satisfactorily and I recommend acceptance of the revised manuscript.

I agree to keep the RNA-seq data as presented in their latest version.

We would like to thank the reviewer for their positive feedback and for their recommendation for acceptance.

Minor comments:

FigS2A: the colors in the legend and graph are not matching for CM-1758

We apologize for this error, which has been corrected in the revised version of the manuscript (**Reviewer Figure 1**).

Reviewer Figure 1: cell differentiation and apoptosis assays in MOLM-13 and MV4-11 cells by CM-444 and CM-1758. CD11b and annexin-V were measured by flow cytometry at 2, 4, 6, and 8 days after daily treatment of MOLM-13 and MV4-11 cell lines with CM-444 (280 and 160 nM, respectively) and CM-1758 (140 and 80 nM, respectively). Data are presented as mean values +/- S.D. of three replicates.

In the revised version of the manuscript the **Reviewer Figure 1** has been included as **Figure S2A**.

Page 6, 3rd paragraph: in the sentence “Principal component analysis (PCA) and unsupervised hierarchical clustering analysis of RNA-seq data showed the differences between treatments (Figure S2B).” delete “unsupervised hierarchical clustering” or add “data not shown” since the figure was omitted in the latest version of the manuscript.

We apologize for this error, which has been corrected in the revised version of the manuscript.

Results section, page 6: “Principal component analysis (PCA) of RNA-seq data showed the differences between treatments (Figure S2B)”.

Reviewer #3 (Remarks to the Author):

I would like to thank the authors for the confirmation of their results by robust additional experiments and wish them widespread impact after publication of their findings. May their new compound help patients in the near future!

We would like to express our sincere gratitude to the reviewer for their kind comments. We appreciate once again their valuable review and encouraging words.